# $k$-Sliced Mutual Information:
# A Quantitative Study of Scalability with Dimension

**Ziv Goldfeld**
Cornell University
goldfeld@cornell.edu

**Kristjan Greenewald**
MIT-IBM Watson AI Lab
IBM Research
kristjan.h.greenewald@ibm.com

**Theshani Nuradha**
Cornell University
pt388@cornell.edu

**Galen Reeves**
Duke University
galen.reeves@duke.edu

## Abstract

Sliced mutual information (SMI) is defined as an average of mutual information (MI) terms between one-dimensional random projections of the random variables. It serves as a surrogate measure of dependence to classic MI that preserves many of its properties but is more scalable to high dimensions. However, a quantitative characterization of how SMI itself and estimation rates thereof depend on the ambient dimension, which is crucial to the understanding of scalability, remain obscure. This work provides a multifaceted account of the dependence of SMI on dimension, under a broader framework termed $k$-SMI, which considers projections to $k$-dimensional subspaces. Using a new result on the continuity of differential entropy in the 2-Wasserstein metric, we derive sharp bounds on the error of Monte Carlo (MC)-based estimates of $k$-SMI, with explicit dependence on $k$ and the ambient dimension, revealing their interplay with the number of samples. We then combine the MC integrator with the neural estimation framework to provide an end-to-end $k$-SMI estimator, for which optimal convergence rates are established. We also explore asymptotics of the population $k$-SMI as dimension grows, providing Gaussian approximation results with a residual that decays under appropriate moment bounds. All our results trivially apply to SMI by setting $k = 1$. Our theory is validated with numerical experiments and is applied to sliced InfoGAN, which altogether provide a comprehensive quantitative account of the scalability question of $k$-SMI, including SMI as a special case when $k = 1$.

## 1 Introduction

Mutual information (MI) is a fundamental measure of dependence between random variables [1, 2], with a myriad of applications in information theory, statistics, and more recently machine learning [3–14]. Its appeal stems from the favorable structural properties it possesses, such as meaningful units (bits or nats), identification of independence, entropy decompositions, and convenient variational forms. However, modern learning applications require estimating MI between high-dimensional variables based on data, which is known to be notoriously hard with exponential in dimension sample complexity [15, 16]. To alleviate this impasse, sliced MI (SMI) was recently introduced by a subset of the authors as a surrogate dependence measure that preserves much of the classic structure while being more scalable for computation and estimations in high dimensions [17].

Inspired by slicing techniques for statistical divergences [18–21], SMI is defined as an average of MI terms between one-dimensional projections of the high-dimensional variables. Beyond showing that

36th Conference on Neural Information Processing Systems (NeurIPS 2022).

SMI inherits many properties of its classic counterpart, [17] demonstrated that it can be estimated with (optimal) parametric error rates in all dimensions by combining a MI estimator between scalar variables with a MC integrator. However, the bounds from [17] rely on high-level assumptions that may be hard to verify in practice and hide dimension-dependent constants whose characterization is crucial for understanding scalability in dimension. Furthermore, when projecting high-dimensional variables it is natural to ask what information can be extracted from more than just the real line, say, a subspace of dimension $k \geq 1$, but this extension was not considered in [17]. This work defines $k$-SMI (which employs projections to $k$-dimensional subspaces), and provides a comprehensive quantitative study of its dependence on dimension, encompassing the MC error, formal guarantees for neural estimators, and asymptotics of the population $k$-SMI as dimension increases. All our results trivially apply for the original SMI case (when $k = 1$), thereby closing the aforementioned gaps in analysis from [17].

## 1.1 Contributions

The objective of this work is provide a thorough quantitative study of the dependence of SMI on dimension. We do so under the slightly broader framework of $k$-SMI, which we define between random variables $X$ and $Y$ with values in $\mathbb{R}^{d_x}$ and $\mathbb{R}^{d_y}$ as

$$\mathsf{SI}_k(X;Y) := \int_{\mathrm{St}(k,d_x)} \int_{\mathrm{St}(k,d_y)} \mathsf{I}(\mathrm{A}^\intercal X; \mathrm{B}^\intercal Y) d\sigma_{k,d_x}(\mathrm{A}) d\sigma_{k,d_y}(\mathrm{B}), \tag{1}$$

where $\mathrm{St}(k,d)$ is the Stiefel manifold of $d \times k$ matrices with orthonormal columns and $\sigma_{k,d}$ is its uniform measure. $k$-SMI coincides with SMI when $k = 1$, but to further support it as a natural extension, we show that structural properties of SMI derived in [17] still hold for any $1 \leq k \leq \min\{d_x, d_y\}$. We then move to study formal guarantees for $k$-SMI estimation, targeting explicit dependence on $(k, d_x, d_y)$. A key technical tool we employ is a new continuity result of differential entropy with respect to (w.r.t.) the 2-Wasserstein distance $\mathsf{W}_2$, which we derive using the HWI inequality from [22, 23]. Our continuity claim strengthens the one from [24] in two ways: (i) it replaces the $(c_1, c_2)$-regularity condition therein with the weaker requirement of finite Fisher information, and (ii) it sharpens the constant multiplying $\mathsf{W}_2$ to be optimal. As a corollary, we show that the differential entropy of a projected variable, say $\mathsf{h}(\mathrm{A}^\intercal X)$, is Lipschitz continuous w.r.t. the Frobenius norm on the $\mathrm{St}(k,d)$.

Lipschitzness is pivotal for obtaining dimension-dependent bounds on MC-based estimates of $k$-SMI. We bound the MC error in terms of the variance of $\mathsf{I}(\mathrm{A}^\intercal X; \mathrm{B}^T Y)$ when $(\mathrm{A}, \mathrm{B})$ are uniform over their respective Stiefel manifolds. Lipschitz continuity of differential entropy implies Lipschitzness of this projected MI, which enables controlling its variance via a concentration argument over $\mathrm{St}(k,d)$. The resulting bound scales as $O\big(\sqrt{k(1/d_x + 1/d_y)/m}\big)$, where $m$ is the number of MC samples and the constant is explicitly expressed via basic characteristics of the $(X, Y)$ distribution (its covariance and Fisher information matrices). This result, which also applies to standard SMI, sharpens the bounds from [17], characterizes the dependence on dimension, and holds under primitive assumptions on the joint distribution. Furthermore, the bound reveals that higher dimension can shrink the error in some cases—a surprising observation which is also verified numerically on synthetic examples.

In addition to MC integration, the $k$-SMI estimator employs a generic MI estimator between $k$-dimensional variables. We instantiate this estimator via the neural estimation framework based on the Donsker-Varadhan (DV) variational form [25] (see also [26-28]). The neural estimator is realized by an $\ell$-neuron shallow ReLU network and the effective convergence rate of the resulting $k$-SMI estimate is explored. We lift the convergence rates derived in [29] for neural estimators of $f$-divergences to the $k$-SMI problem. The resulting rate scales as $O\big(k^{1/2}(\ell^{-1/2} + m^{-1/2} + kn^{-1/2})\big)$, where $\ell$ is the number of neurons, $m$ is the number of MC samples, and $n$ is the number of $(X, Y)$ samples. Equating $\ell$, $m$, and $n$ results in the (optimal) parametric rate. Our result also shows that neural estimation of $k$-SMI requires milder smoothness assumptions on the population distributions. Namely, we relax the smoothness level $\lfloor (d_x + d_y)/2 \rfloor + 3$ imposed in [29] to $k + 3$, i.e., adapting to the projection dimension rather than the ambient one. This is a significant relaxation since we often have $d_x, d_y \gg k$.

To further understand the effect of the ambient dimension, we explore how $\mathsf{SI}_k(X;Y)$ behaves as $d_x, d_y \to \infty$. To that end, we first provide a full characterization of $\mathsf{SI}_k(X, Y)$ between jointly Gaussian variables, revealing that it scales as $k^2/(d_x d_y)$ times the squared Frobenius norm of the cross-

covariance matrix. We then show that general $k$-SMI can be decomposed into a Gaussian part plus a residual term that quantifies the average distance (over projections) from Gaussianity. The latter is intimately related to the conditional central limit theorem (CLT) phenomenon [30–32], and we use those ideas to identify approximate isotropy conditions under which the residual vanishes as $d_x, d_y \to \infty$. Lastly, we conduct an empirical study that validates our theory and explores applications to independence testing and sliced infoGAN. Specifically, we revisit the infoGAN generative model [6] and replace the classic MI used therein with SMI. Training the model, we find that it successfully learns disentangled representations despite the low-dimensional projections, suggesting that SMI can replace classic MI even in applications with complex underlying structure.

## 2 Background and Preliminaries

### 2.1 Notation and Definitions

**Notation.** For $d \geq 1$, $\|\cdot\|$ is the Euclidean norm in $\mathbb{R}^d$, $\langle \cdot, \cdot \rangle$ is the inner product, while $\|\cdot\|_1$ is the $\ell^1$ norm. We use $\|\cdot\|_{\mathrm{op}}$ and $\|\cdot\|_{\mathrm{F}}$ for the operator and Frobenius norms of matrices, respectively. Matrix inequalities are understood in the sense of (partial) semi-definite ordering, i.e., we write $\mathrm{A} \succeq \mathrm{B}$ when $\mathrm{A} - \mathrm{B}$ is positive semi-definite. The Stiefel manifold of $d \times k$ matrices with orthonormal columns is denoted by $\mathrm{St}(k, d)$. For a $d \times k$ matrix $\mathrm{A}$, we use $\mathfrak{p}^{\mathrm{A}} : \mathbb{R}^d \to \mathbb{R}^k$ for the orthogonal projection onto the row space of $\mathrm{A}$.

Let $\mathcal{P}(\mathbb{R}^d)$ denote the space of Borel probability measures on $\mathbb{R}^d$, and set $\mathcal{P}_2(\mathbb{R}^d) := \{\mu \in \mathcal{P}(\mathbb{R}^d) : \int \|x\|^2 d\mu(x) < \infty\}$ as the subset of distributions with finite 2nd absolute moment. For $\mu, \nu \in \mathcal{P}(\mathbb{R}^d)$, we use $\mu \otimes \nu$ to denote a product measure, while $\mathrm{spt}(\mu)$ designates the support of $\mu$. We use $\mathrm{Leb}$ for the Lebesgue measure on $\mathbb{R}^d$, and denote the subset of probability measures that are absolutely continuous w.r.t. $\mathrm{Leb}$ by $\mathcal{P}_{\mathsf{ac}}(\mathbb{R}^d)$. For a measurable map $f$, the pushforward of $\mu$ under $f$ is denoted by $f_\sharp \mu = \mu \circ f^{-1}$, i.e., if $X \sim \mu$ then $f(X) \sim f_\sharp \mu$. For $a, b \in \mathbb{R}$, we use the notation $a \wedge b = \min\{a, b\}$ and $a \vee b = \max\{a, b\}$. We write $a \lesssim_x b$ when $a \leq C_x b$ for a constant $C_x$ that depends only on $x$ ($a \lesssim b$ means the constant is absolute).

For a multi-index $\alpha = (\alpha_1, \ldots, \alpha_d) \in \mathbb{Z}_{\geq 0}^d$, the partial derivative operator of order $\|\alpha\|_1$ is denoted by $D^\alpha = \frac{\partial^{\alpha_1}}{\partial^{\alpha_1} x_1} \cdots \frac{\partial^{\alpha_d}}{\partial^{\alpha_d} x_d}$. For an open set $\mathcal{U} \subseteq \mathbb{R}^d$ and integer $s \geq 0$, the class of functions whose partial derivatives up to order $s$ all exist and are continuous on $\mathcal{U}$ is denoted by $\mathsf{C}^s(\mathcal{U})$, and we define the subclass $\mathsf{C}_b^s(\mathcal{U}) := \{f \in \mathsf{C}^s(\mathcal{U}) : \max_{\alpha : \|\alpha\|_1 \leq s} \|D^\alpha f\|_{\infty, \mathcal{U}} \leq b\}$. The restriction of $f : \mathbb{R}^d \to \mathbb{R}$ to $\mathcal{X} \subseteq \mathbb{R}^d$ is denoted by $f|_{\mathcal{X}}$. For compact $\mathcal{X}$, slightly abusing notation, we set $\|\mathcal{X}\| := \sup_{x \in \mathcal{X}} \|x\|$.

**Divergences and information measures.** Let $\mu, \nu \in \mathcal{P}(\mathbb{R}^d)$ satisfy $\mu \ll \nu$, i.e., $\mu$ is absolutely continuous w.r.t. $\nu$. The relative entropy and the relative Fisher information are defined, respectively, as $\mathsf{D}(\mu \| \nu) := \int_{\mathbb{R}^d} \log(d\mu/d\nu) d\mu$ and $\mathsf{J}(\mu \| \nu) := \int_{\mathbb{R}^d} \|\nabla \log(d\mu/d\nu)\|^2 d\mu$. The 2-Wasserstein distance between $\mu, \nu \in \mathcal{P}_2(\mathbb{R}^d)$ is $\mathsf{W}_2(\mu, \nu) := \inf_{\pi \in \Pi(\mu, \nu)} \left( \int_{\mathbb{R}^d \times \mathbb{R}^d} \|x - y\|^2 d\pi(x, y) \right)^{1/2}$, where $\Pi(\mu, \nu)$ is the set of couplings of $\mu$ and $\nu$. All three measures are divergences, i.e., non-negative and nullify if and only if (iff) $\mu = \nu$. In fact, $\mathsf{W}_2$ is a metric on $\mathcal{P}_2(\mathbb{R}^d)$, which metrizes weak convergence plus convergence of 2nd moments.

MI and differential entropy are defined from the relative entropy as follows. Consider a pair of random variables $(X, Y) \sim \mu_{XY} \in \mathcal{P}(\mathbb{R}^{d_x} \times \mathbb{R}^{d_y})$ and denote the corresponding marginal distributions by $\mu_X$ and $\mu_Y$. The MI between $X$ and $Y$ is given by $\mathsf{I}(X; Y) := \mathsf{D}(\mu_{XY} \| \mu_X \otimes \mu_Y)$ and serves as a measure of dependence between those random variables. The differential entropy of $X$ is defined as $\mathsf{h}(X) = \mathsf{h}(\mu_X) := -\mathsf{D}(\mu_X \| \mathrm{Leb})$. MI between (jointly) continuous variables and differential entropy are related via $\mathsf{I}(X; Y) = \mathsf{h}(X) + \mathsf{h}(Y) - \mathsf{h}(X, Y)$; decompositions in terms of conditional entropies are also available [1]. The Fisher information of $X \sim \mu$ is $\mathsf{J}(\mu) := \mathsf{J}(\mu \| \mathrm{Leb})$. Denoting the density of $\mu$ by $f_\mu$, the Fisher information matrix of $\mu$ is $\mathsf{J}_{\mathrm{F}}(\mu) := \mathbb{E}\left[ (\nabla \log f_\mu)(\nabla \log f_\mu)^\intercal \right]$, and we have $\mathrm{tr}\left( \mathsf{J}_{\mathrm{F}}(\mu) \right) = \mathsf{J}(\mu)$.

## 2.2 Lipschitz Continuity of Projected Differential Entropy

A key technical tool we use is a new continuity result of differential entropy w.r.t. the 2-Wasserstein distance. It strengthens an earlier version of this result from [24], and may be of independent interest.

**Lemma 1** (Wasserstein continuity). *Let $\mu, \nu \in \mathcal{P}_2(\mathbb{R}^d)$ satisfy $\mu \ll \mathrm{Leb}$ and $\mathsf{h}(\mu), \mathsf{J}(\nu) < \infty$. Then*

$$\mathsf{h}(\mu) - \mathsf{h}(\nu) \leq \sqrt{\mathsf{J}(\nu)} \mathsf{W}_2(\mu, \nu),$$

*and the constant above is optimal in the sense that* $\sup_{\substack{\mu \neq \nu: \\ \mathsf{h}(\mu), \mathsf{J}(\nu) < \infty}} \frac{\mathsf{h}(\mu) - \mathsf{h}(\nu)}{\sqrt{\mathsf{J}(\nu)} \mathsf{W}_2(\mu, \nu)} = 1.$

The proof of the lemma, given in Appendix A.1, follows by invoking the HWI inequality for the difference of relative entropies [22, 23] with an isotropic Gaussian reference measure $\gamma_\sigma = \mathcal{N}(0, \sigma^2 \mathrm{I}_d)$, re-expressing the relative entropy difference in terms of differential entropies, and taking the limit as $\sigma \to 0$.

**Remark 1** (Comparison to [24]). *Continuity of differential entropy w.r.t. the $\mathsf{W}_2$ was previously derived in [24, Proposition 1], but via a different argument, under stronger conditions, and without an optimal constant. The inequality from [24] assumed $(c_1, c_2)$-regularity of the density of $\nu$ (i.e., that $\|\nabla \log f_\nu(x)\| \leq c_1 \|x\| + c_2$, for all $x \in \mathbb{R}^d$), which is stronger than $\mathsf{J}(\nu) < \infty$ when $\nu \in \mathcal{P}_2(\mathbb{R}^d)$.*

A rather direct implication of Lemma 1 is the following Lipschitz continuity of projected entropy (also proven in Appendix A.1), which plays a key role in the subsequent analysis of $k$-SMI estimation.

**Proposition 1** (Lipschitzness of projected entropy). *Let $\mu \in \mathcal{P}_2(\mathbb{R}^d)$ have covariance matrix $\Sigma_\mu$ and $\mathsf{J}(\mu) < \infty$. For any $\mathrm{A}, \mathrm{B} \in \mathrm{St}(k, d)$, we have $\left| \mathsf{h}(\mathfrak{p}_\sharp^A \mu) - \mathsf{h}(\mathfrak{p}_\sharp^B \mu) \right| \leq \sqrt{k \|\mathsf{J}_\mathrm{F}(\mu)\|_{\mathrm{op}} \|\Sigma_\mu\|_{\mathrm{op}}} \|A - B\|_\mathrm{F}.$*

## 3 $k$–Sliced Mutual Information

SMI was defined in [17] as an average of MI terms between one-dimensional projections of the considered random variables. As higher dimensional projections preserve more information about the original $(X, Y)$, we extend this definition to $k$-dimensional projections.

**Definition 1** ($k$-sliced mutual information). *For $1 \leq k \leq d_x \wedge d_y$, the $k$-SMI between $(X, Y) \sim \mu_{XY} \in \mathcal{P}(\mathbb{R}^{d_x} \times \mathbb{R}^{d_y})$ is defined in (1), where $\sigma_{k,d}$ is the uniform distribution on $\mathrm{St}(d, k)$.*

$k$-SMI can be equivalently expressed in term of conditional (classic) MI as $\mathsf{SI}_k(X; Y) = \mathsf{I}(\mathrm{A}^\intercal X; \mathrm{B}^\intercal Y | \mathrm{A}, \mathrm{B})$, where $(\mathrm{A}, \mathrm{B}) \sim \sigma_{k,d_x} \otimes \sigma_{k,d_y}$, i.e., $(\mathrm{A}, \mathrm{B})$ are independent and uniform over the respective Stiefel manifolds. $k$-SMI reduces to the SMI from [17] when $k = 1$. Below we show that $\mathsf{SI}_k$ preserves the structural properties of SMI, as derived in [17, Section 3].

**Remark 2** (Related definitions). *$k$-SMI entropy decompositions and chain rule require defining $k$-sliced entropy and conditional $k$-SMI. For $(X, Y, Z) \sim \mu_{XYZ} \in \mathcal{P}(\mathbb{R}^{d_x} \times \mathbb{R}^{d_y} \times \mathbb{R}^{d_z})$ and $(\mathrm{A}, \mathrm{B}, \mathrm{C}) \sim \sigma_{k,d_x} \otimes \sigma_{k,d_y} \otimes \sigma_{k,d_z}$, the $k$-sliced entropy of $X$ is $\mathsf{sh}_k(X) := \mathsf{h}(\mathrm{A}^\intercal X | \mathrm{A})$, while the conditional version given $Y$ is given by $\mathsf{sh}_k(X|Y) := \mathsf{h}(\mathrm{A}^\intercal X | \mathrm{A}, \mathrm{B}, \mathrm{B}^\intercal Y)$. The condition $k$-SMI between $X$ and $Y$ given $Z$ is $\mathsf{SI}_k(X; Y|Z) := \mathsf{I}(\mathrm{A}^\intercal X; \mathrm{B}^\intercal Y | \mathrm{A}, \mathrm{B}, \mathrm{C}, \mathrm{C}^\intercal Z)$.*

### 3.1 Structural Properties

We verify that $k$-SMI preserves structural properties previously established in [17] for SMI.

**Proposition 2** ($k$-SMI properties). *For any $1 \leq k \leq d_x \wedge d_y$, the following properties hold:*

1. ***Identification of independence:*** $\mathsf{SI}_k(X; Y) \geq 0$ *with equality iff $X$ and $Y$ are independent.*

2. ***Bounds:*** *For integers $k_1 < k_2$:* $\mathsf{SI}_{k_1}(X; Y) \leq \mathsf{SI}_{k_2}(X; Y) \leq \sup_{\substack{\mathrm{A} \in \mathrm{St}(k_2, d_x) \\ \mathrm{B} \in \mathrm{St}(k_2, d_y)}} \mathsf{I}(\mathrm{A}^\intercal X; \mathrm{B}^\intercal Y) \leq \mathsf{I}(X; Y).$

3. ***Relative entropy and variational form:*** *Let $(\tilde{X}, \tilde{Y}) \sim \mu_X \otimes \mu_Y$ and $(\mathrm{A}, \mathrm{B}) \sim \sigma_{k,d_x} \otimes \sigma_{k,d_y}$, then*

$$\mathsf{SI}_k(X; Y) = \mathsf{D}_{\mathsf{KL}}\big((\mathfrak{p}^A, \mathfrak{p}^B)_\sharp \mu_{XY} \big\| (\mathfrak{p}^A, \mathfrak{p}^B)_\sharp \mu_X \otimes \mu_Y \big| \mathrm{A}, \mathrm{B}\big)$$

$$= \sup_{f: \mathrm{St}(k, d_x) \times \mathrm{St}(k, d_y) \times \mathbb{R}^{2k} \to \mathbb{R}} \mathbb{E}\big[f(\mathrm{A}, \mathrm{B}, \mathrm{A}^\intercal X, \mathrm{B}^\intercal Y)\big] - \log\left(\mathbb{E}\left[e^{f(\mathrm{A}, \mathrm{B}, \mathrm{A}^\intercal \tilde{X}, \mathrm{B}^\intercal \tilde{Y})}\right]\right),$$

*where the supremum is over all measurable functions for which both expectations are finite.*

4. **Entropy decomposition:** $\mathsf{SI}_k(X;Y) = \mathsf{sh}_k(X) - \mathsf{sh}_k(X|Y) = \mathsf{sh}_k(Y) - \mathsf{sh}_k(Y|X) = \mathsf{sh}_k(X) + \mathsf{sh}_k(Y) - \mathsf{sh}_k(X,Y)$, *provided that all the relevant (joint / marginal / conditional) densities exist.*

5. **Chain rule:** *For any $X_1, \dots, X_n, Y, Z$, we have* $\mathsf{SI}_k(X_1, \dots, X_n; Y) = \mathsf{SI}_k(X_1; Y) + \sum_{i=2}^{n} \mathsf{SI}_k(X_i; Y | X_1, \dots, X_{i-1})$. *In particular,* $\mathsf{SI}_k(X, Y; Z) = \mathsf{SI}_k(X; Z) + \mathsf{SI}_k(Y; Z | X)$.

6. **Tensorization:** *For mutually independent* $\{(X_i, Y_i)\}_{i=1}^n$, $\mathsf{SI}_k\left(\{X_i\}_{i=1}^n; \{Y_i\}_{i=1}^n\right) = \sum_{i=1}^{n} \mathsf{SI}_k(X_i; Y_i)$.

The proposition is proven in Appendix A.2 via a direct extension of the $k = 1$ argument from [17].

## 4 Estimation and Asymptotics of *k*-SMI in High Dimensions

As shown in [17], SMI can be estimated from high-dimensional data by combining a MI estimator between scalar random variables and a MC integration step. However, the bounds from [17] do not explicitly capture dependence on the ambient dimension, which is crucial for understanding scalability of the approach. We now extend the estimator from [17] to $k$-SMI and provide formal guarantees with explicit dependence on $k$, $d_x$, and $d_y$, thus closing the said gap.

To estimate $k$-SMI, let $\{(X_i, Y_i)\}_{i=1}^n$ be i.i.d. from $\mu_{XY} \in \mathcal{P}(\mathbb{R}^{d_x} \times \mathbb{R}^{d_y})$ and proceed as follows:

1. Draw $\{(A_j, B_j)\}_{j=1}^m$ i.i.d. from $\sigma_{k,d_x} \otimes \sigma_{k,d_y}$ (i.e., each pair is uniform on $\mathrm{St}(k, d_x) \times \mathrm{St}(k, d_y)$).[1]

2. Compute $\left\{(A_j^\mathsf{T} X_i, B_j^\mathsf{T} Y_i)\right\}_{j,i=1}^{m,n}$, which, for fixed $(A_j, B_j)$, are samples from $(\mathfrak{p}^A, \mathfrak{p}^B)_\sharp \mu_{XY}$.

3. For each $j = 1, \dots, m$, a MI estimator between $k$-dimensional random vectors is applied to the $n$ samples corresponding to $(A_j, B_j)$ to obtain an estimate $\hat{\mathsf{I}}\left((A_j^\mathsf{T} X)^n, (B_j^\mathsf{T} Y)^n\right)$ of $\mathsf{I}(A_j^\mathsf{T} X; B_j^\mathsf{T} Y)$, where $(A_j^\mathsf{T} X)^n := (A_j^\mathsf{T} X_1, \dots, A_j^\mathsf{T} X_n)$ and $(B_j^\mathsf{T} Y)^n$ is defined similarly.

4. Take a MC average of the above estimates, resulting in the $k$-SMI estimator:

$$\widehat{\mathsf{SI}}_k^{m,n} := \frac{1}{m} \sum_{j=1}^{m} \hat{\mathsf{I}}\left((A_j^\mathsf{T} X)^n, (B_j^\mathsf{T} Y)^n\right). \tag{2}$$

We provide formal guarantees for the quality of the $\widehat{\mathsf{SI}}_k^{m,n}$ estimator given a generic $k$-dimensional MI estimator $\hat{\mathsf{I}}(\cdot, \cdot)$ in Step 3. Afterwards, we instantiate the latter as a neural MI estimator and provide explicit convergence rates. To get further insight into the dependence on dimension, we study asymptotics of Gaussian $k$-SMI as $d_x, d_y \to \infty$ and corresponding Gaussian approximation arguments.

### 4.1 Error Bounds with Explicit Dimension Dependence

Our analysis decomposes the overall error of $\widehat{\mathsf{SI}}_k^{m,n}$ into the MC error plus the error of the $k$-dimensional MI estimator $\hat{\mathsf{I}}(\cdot, \cdot)$. We first consider an arbitrary estimator $\hat{\mathsf{I}}(\cdot, \cdot)$ whose error is (implicitly) upper bounded by $\delta_k(n)$ and focus on analyzing the MC error, targeting explicit dependence on $k$, $d_x$, and $d_y$. As in [17], the statement relies on the following assumption on the $k$-dimensional estimator $\hat{\mathsf{I}}(\cdot; \cdot)$.

**Assumption 1.** $(X, Y) \sim \mu_{XY} \in \mathcal{P}(\mathbb{R}^{d_x} \times \mathbb{R}^{d_y})$ *is such that* $\mathsf{I}(A^\mathsf{T} X; B^\mathsf{T} Y)$ *can be estimated by* $\hat{\mathsf{I}}\left((A^\mathsf{T} X)^n, (B^\mathsf{T} Y)^n\right)$ *with error at most* $\delta_k(n)$, *uniformly over* $(A, B) \in \mathrm{St}(k, d_x) \times \mathrm{St}(k, d_y)$.

**Theorem 1** (*k*-SMI estimation error). *Let* $\mu_{XY} \in \mathcal{P}_2(\mathbb{R}^{d_x} \times \mathbb{R}^{d_y})$ *satisfy Assumption 1, have marginal covariance matrices* $\Sigma_X$ *and* $\Sigma_Y$, *and* $\mathsf{J}(\mu_{XY}) < \infty$. *Then the estimator from* (2) *has error bounded by*

$$\mathbb{E}\left[\left|\mathsf{SI}_k(X; Y) - \widehat{\mathsf{SI}}_k^{m,n}\right|\right] \leq C(\mu_{XY}) \sqrt{\frac{k(d_x + d_y)}{d_x d_y}} m^{-\frac{1}{2}} + \delta_k(n), \tag{3}$$

---

[1] A simple approach for sampling the uniform distribution on $\mathrm{St}(k, d)$ is to draw $kd$ random samples from $\mathcal{N}(0, 1)$, arrange them into an $d \times k$ matrix $\Lambda$, and compute $\Lambda(\Lambda^\mathsf{T} \Lambda)^{-1/2}$ (cf. [33, Theorem 2.2.1]). A slightly more efficient approach is to first apply a QR decomposition to $\Lambda$ and then follow the aforementioned sampling method only to the Q matrix. Note that for $k = O(1)$, both computation times are linear in $d$ (QR decomposition via the Schwarz-Rutishauser algorithm is $O(dk^2)$) [34].

*where $C(\mu_{XY}) = 21\sqrt{\|J_F(\mu_{XY})\|_{\mathrm{op}}\big(\|\Sigma_X\|_{\mathrm{op}} \vee \|\Sigma_Y\|_{\mathrm{op}}\big)}$.*

The proof of Theorem 1 (in Appendix A.3) bounds the MC error by $\big(\mathrm{Var}\big(i_{XY}(\mathrm{A},\mathrm{B})\big)/m\big)^{\frac{1}{2}}$, where $i_{XY}(\mathrm{A},\mathrm{B}) := \mathsf{I}(\mathrm{A}^\intercal X; \mathrm{B}^\intercal Y)$ and $(\mathrm{A},\mathrm{B}) \sim \sigma_{k,d_x} \otimes \sigma_{k,d_y}$. We then use the continuity result from Proposition 1 along with the entropy decomposition of $k$-SMI (Proposition 2, Claim 4) to show that $i_{XY}$ is Lipschitz continuous (w.r.t. the Frobenius norm) on $\mathrm{St}(k,d_x) \times \mathrm{St}(k,d_y)$. Concentration of Lipschitz functions on the Stiefel manifold and the Efron-Stein inequality then imply the above bound. This result clarifies the dependence of the MC error on $k$, $d_x$, and $d_y$, and reveals scaling rates of the parameters with $m$ for which (high-dimensional) convergence holds true.

**Remark 3** (Comparison to [17])**.** *Theorem 1 from [17] treats the $k = 1$ case under stronger high-level assumptions and without identifying the dependence on dimension. Namely, assuming the uniform bound $\|i_{XY}\|_{L^\infty} \leq M$, they control the variance by $M^2/4$ to obtain the $m^{-1/2}$ rate, although $M$ generally depends on $(d_x, d_y)$. Herein, we rely on the finer observation that $i_{XY}$ is Lipschitz and use concentration results to get a dimension-dependent bound in terms of basic characteristics of $(X, Y)$.*

**Remark 4** (Blessing of dimensionality)**.** *The constant in the MC error may decay as dimension grows. For instance, if $X$ and $Y$ are both $d$-dimensional with identity covariance matrices, then $\|\Sigma_X\|_{\mathrm{op}}, \|\Sigma_Y\|_{\mathrm{op}}$ are $O_d(1)$. For such $(X, Y)$, the MC bound decays to 0 as $d \to \infty$, assuming that $\|J_F(\mu_{XY})\|_{\mathrm{op}}$ grows at most sublinearly with $d$. Also note that $C(\mu_{XY})$ has the same invariances as the $k$-SMI: it is invariant to translations and scalings of the form $(X, Y) \mapsto (sX, sY)$ for $s \neq 0$.*

## 4.2   Neural Estimation

We now instantiate the $k$-dimensional MI estimator via the neural estimation framework of [29, 35], and obtain an explicit bound on $\delta_k(n)$ in terms of $m$, $n$, $k$, and the size of the neural network.

Neural estimation of MI relies on the DV variational form

$$\mathsf{I}(U; V) = \sup_{f:\mathbb{R}^{d_u} \times \mathbb{R}^{d_v} \to \mathbb{R}} \mathbb{E}[f(U, V)] - \log\left(e^{\mathbb{E}[f(\tilde{U}, \tilde{V})]}\right),$$

where $(U, V) \sim \mu_{UV}$, $(\tilde{U}, \tilde{V}) \sim \mu_U \otimes \mu_V$, and $f$ is a measurable function for which the expectations above are finite. Define the class of $\ell$-neuron ReLU network as

$$\mathcal{G}_{d_u,d_v}^\ell(a) := \left\{ g : \mathbb{R}^{d_u+d_v} \to \mathbb{R} : \begin{array}{l} g(z) = \sum_{i=1}^\ell \beta_i \phi\left(\langle w_i, z \rangle + b_i\right) + \langle w_0, z \rangle + b_0, \\[2mm] \max_{1 \leq i \leq \ell} \|w_i\|_1 \vee |b_i| \leq 1, \ \max_{1 \leq i \leq \ell} |\beta_i| \leq \dfrac{a}{2\ell}, \ |b_0|, \|w_0\|_1 \leq a \end{array} \right\},$$

where $\phi(z) = z \vee 0$ is the ReLU activation; set the shorthand $\mathcal{G}_{d_u,d_v}^\ell = \mathcal{G}_{d_u,d_v}^\ell(\log\log \ell \vee 1)$. Given i.i.d. data $(U_1, V_1), \ldots, (U_n, V_n)$ from $\mu_{UV}$, the neural estimator parameterizes the DV potential $f$ by the class $\mathcal{G}_{d_u,d_v}^\ell$ and approximates expectations by sample means,[2] resulting in the estimate

$$\hat{\mathsf{I}}_{d_u,d_v}^\ell(U^n, V^n) := \sup_{g \in \mathcal{G}_{d_u,d_v}^\ell} \frac{1}{n} \sum_{i=1}^n g(U_i, V_i) - \log\left(\frac{1}{n} \sum_{i=1}^n e^{g(U_i, V_{\sigma(i)})}\right).$$

For $k$-SMI neural estimation, we set

$$\widehat{\mathsf{SI}}_{k,\mathsf{NE}}^{\ell,m,n} := \frac{1}{m} \sum_{j=1}^m \hat{\mathsf{I}}_{k,k}^\ell\big((\mathrm{A}_j^\intercal X)^n, (\mathrm{B}_j^\intercal Y)^n\big),$$

i.e., we use $\hat{\mathsf{I}}_{k,k}^\ell$ as the $k$-dimensional MI estimator in (2). This estimator is readily implemented by parallelizing $m$ $\ell$-neuron ReLU nets with inputs in $\mathbb{R}^{2k}$ and scalar outputs. We provide explicit convergence rates for it over an appropriate distribution class, drawing upon the results of [29] for neural estimation of $f$-divergences (see also [35]). For compact $\mathcal{X} \subset \mathbb{R}^{d_x}$ and $\mathcal{Y} \subset \mathbb{R}^{d_y}$, let $\mathcal{P}_{\mathsf{ac}}(\mathcal{X} \times \mathcal{Y}) := \{\mu_{XY} \in \mathcal{P}_{\mathsf{ac}}(\mathbb{R}^{d_x} \times \mathbb{R}^{d_y}) : \mathrm{spt}(\mu_{XY}) \subseteq \mathcal{X} \times \mathcal{Y}\}$, and denote the density of $\mu_{XY}$ by $f_{XY}$. The distribution class of interest is

$$\mathcal{F}_{d_x,d_y}^k(M, b) := \left\{ \mu_{XY} \in \mathcal{P}_{\mathsf{ac}}(\mathcal{X} \times \mathcal{Y}) : \begin{array}{l} \exists\, r \in \mathsf{C}_b^{k+3}(\mathcal{U}) \text{ for some open set } \mathcal{U} \supset \mathcal{X} \times \mathcal{Y} \\ \text{s.t. } \log f_{XY} = r|_{\mathcal{X} \times \mathcal{Y}}, \ \mathsf{I}(X;Y) \leq M \end{array} \right\},$$

---

[2]Negative samples, i.e., from $\mu_X \otimes \mu_Y$, can be obtained from the positive one via $(U_1, V_{\sigma(1)}), \ldots, (U_n, V_{\sigma(n)})$, where $\sigma \in S_n$ is a permutation such that $\sigma(i) \neq i$, for all $i = 1, \ldots, n$.

which, in particular, contains distributions whose densities are bounded from above and below on $\mathcal{X} \times \mathcal{Y}$ with a smooth extension to an open set covering $\mathcal{X} \times \mathcal{Y}$. This includes uniform distributions, truncated Gaussians, truncated Cauchy distributions, etc.

We next provide convergence rates for the $k$-SMI estimator from (2), uniformly over $\mathcal{F}^k_{d_x,d_y}(M,b)$.

**Theorem 2** (Neural estimation error). *For any $M, b \geq 0$, we have*

$$\sup_{\mu_{X,Y} \in \mathcal{F}^k_{d_x,d_y}(M,b)} \mathbb{E}\left[\left|\mathsf{SI}_k(X;Y) - \widehat{\mathsf{SI}}^{\ell,m,n}_{k,\mathsf{NE}}\right|\right] \lesssim_{M,b,k,d_x,d_y,\|\mathcal{X}\times\mathcal{Y}\|} k^{\frac{1}{2}}\left(m^{-\frac{1}{2}} + \ell^{-\frac{1}{2}} + kn^{-\frac{1}{2}}\right).$$

*The dependence on $d_x, d_y$ above is only through the MC bound (3) (explicit) and $\|\mathcal{X} \times \mathcal{Y}\|$ (implicit).*

Theorem 2 is proven in Appendix A.4 by combining the MC bound from Theorem 1 with the neural estimation error bound from [29, Proposition 2]. To apply that bound for each $\mathsf{I}(\mathrm{A}^\intercal X; \mathrm{B}^\intercal Y)$, where $(\mathrm{A}, \mathrm{B}) \in \mathrm{St}(k, d_x) \times \mathrm{St}(k, d_y)$, we show that the existence of an extension $r$ of $\log f_{XY}$ with $k + 3$ continuous and uniformly bounded derivatives implies that the density of $(\mathrm{A}^\intercal X, \mathrm{B}^\intercal Y)$ also has such an extension.

**Remark 5** (Parametric rate and optimality). *Taking $\ell \asymp m \asymp n$, the resulting rate in Theorem 2 is parametric, and hence minimax optimal. This result implicitly assumes that $M$ is known when picking the neural net parameters. This assumption can be relaxed to mere existence of (an unknown) $M$, resulting in an extra $\mathrm{polylog}(\ell)$ factor multiplying the $n^{-1/2}$ term.*

**Remark 6** (Comparison to [29]). *Neural estimation of classic MI under the framework of [29] requires the density to have Hölder smoothness $s \geq \lfloor (d_x + d_y)/2 \rfloor + 3$. For $\mathsf{SI}_k(X;Y)$, smoothness of $k + 3$ is sufficient (even though the ambient dimension is the same), which mean it can be estimated over a larger class of distributions. This is another virtue of slicing in addition to fast convergence rates. For SMI (i.e., $k = 1$) as in [17], a constant smoothness level suffices irrespective of $(d_x, d_y)$.*

### 4.3 Characterization of and Approximation by Gaussian $k$-SMI

To gain further insight into the dependence of $k$-SMI on dimension, we fully characterize it in the Gaussian case. Afterwards, we show that general $k$-SMI decomposes into a Gaussian part plus a residual, and discuss conditions for the latter to decay as $d \to \infty$. As before, $\Sigma_X$ is the covariance matrix of $X$ (similarly, for $Y$), while $\mathrm{C}_{XY} := \mathbb{E}\left[(X - \mathbb{E}[X])(Y - \mathbb{E}[Y])^\intercal\right]$ is the cross-covariance.

**Theorem 3** (Gaussian $k$-SMI). *Let $(X, Y) \sim \gamma_{XY} = \mathcal{N}(0, \Sigma_{XY})$ be jointly Gaussian random variables. Suppose that $\|\Sigma_X\|_{\mathrm{op}}\|\Sigma_X^{-1}\|_{\mathrm{op}}, \|\Sigma_Y\|_{\mathrm{op}}\|\Sigma_Y^{-1}\|_{\mathrm{op}} \leq \kappa$ and $\|\Sigma_X^{-1/2}\mathrm{C}_{XY}\Sigma_Y^{-1}\|_{\mathrm{op}} \leq \rho$ for some $\kappa \geq 1$ and $\rho < 1$. Then, for any fixed $k$, we have*

$$\mathsf{SI}_k(X;Y) = \frac{k^2\|\mathrm{C}_{XY}\|_\mathrm{F}^2}{2\mathrm{tr}(\Sigma_X)\mathrm{tr}(\Sigma_Y)}\big(1 + o(1)\big),$$

*as $d_x, d_y \to \infty$, where $o(1)$ denotes a quantity that converges to zero in the limit.*

Theorem 3 is proven in Appendix A.5. It states that if $\Sigma_X$ and $\Sigma_Y$ have bounded condition numbers and the correlation, as quantified by $\|\Sigma_X^{-1/2}\mathrm{C}_{XY}\Sigma_Y^{-1}\|_{\mathrm{op}}$, is less than 1, then the Gaussian $k$-SMI is asymptotically equivalent to the squared Frobenius norm $\mathrm{C}_{XY}$, normalized by the traces of the marginal covariances. Since $\|\mathrm{C}_{XY}\|_F^2 \leq (d_x \wedge d_y)\rho^2 \|\Sigma_X\|_{\mathrm{op}}\|\Sigma_Y\|_{\mathrm{op}}$ and $\mathrm{tr}(\Sigma_X)\mathrm{tr}(\Sigma_Y) \geq d_x d_y\|\Sigma_X^{-1}\|_{\mathrm{op}}\|\Sigma_Y^{-1}\|_{\mathrm{op}}$, we see that the $\mathsf{SI}_k(X;Y)$ typically decreases with dimension as $d_x^{-1} \wedge d_y^{-1}$. This rate is inline with the shrinkage with dimension of the MC bound from (3), which renders that bound meaningful even when $k$-SMI is itself decaying, e.g., under the framework of Theorem 3.

**$k$-SMI decomposition and Gaussian approximation.** Given the above result and the recent interest in Gaussian approximations of sliced Wasserstein distances [36, 37], we present a decomposition of $k$-SMI into a Gaussian part plus a residual. For $(X, Y) \sim \mu_{XY} \in \mathcal{P}(\mathbb{R}^{d_x} \times \mathbb{R}^{d_y})$, let $(X^*, Y^*) \sim \gamma_{XY} := \mathcal{N}(0, \Sigma_{XY})$ be jointly Gaussian with the same covariance as $(X, Y)$. The $k$-SMI satisfies

$$\mathsf{SI}_k(X;Y) = \mathsf{SI}_k(X^*;Y^*) + \mathbb{E}\big[\delta_{XY}(A, B)\big], \tag{4}$$

where, for each $(\mathrm{A}, \mathrm{B}) \in \mathrm{St}(k, d_x) \times \mathrm{St}(k, d_y)$

$$\delta_{XY}(\mathrm{A}, \mathrm{B}) := \mathsf{D}\big((\mathfrak{p}^\mathrm{A}, \mathfrak{p}^\mathrm{B})_\sharp\mu_{XY}\big\|(\mathfrak{p}^\mathrm{A}, \mathfrak{p}^\mathrm{B})_\sharp\gamma_{XY}\big) - \mathsf{D}\big((\mathfrak{p}^\mathrm{A}, \mathfrak{p}^\mathrm{B})_\sharp\mu_X \otimes \mu_Y\big\|(\mathfrak{p}^\mathrm{A}, \mathfrak{p}^\mathrm{B})_\sharp\gamma_X \otimes \gamma_Y\big).$$

This decomposition is proven in Appendix A.7. Theorem 3 fully accounts for the first summand, which begs the questions of whether it is the leading term in the decomposition, and under what conditions? This question is intimately related to the conditional CLT of low-dimensional projections under relative entropy [32]. This is a challenging and active research topic [30–32], for which sharp convergence rates remain unknown. As a first step towards a complete answer, in Appendix B we bound this residual term and identify mild isotropy conditions on the marginal distributions of $X$ and $Y$ that are sufficient for the residual term to vanish as $d_x, d_y \to \infty$.

## 5  Experiments

**MC error and Gaussian $k$-SMI rates.** Under the Gaussian setting described next, we illustrate the dependence on $k, d_x, d_y$ of (i) the population $k$-SMI expression in Theorem 3, and (ii) the associated MC estimation error from Theorem 1. Let $Z_1, Z_2 \sim \mathcal{N}(0, \mathrm{I}_d)$ and $V \sim \mathcal{N}(0, \mathrm{I}_2)$ be independent, and set $X = \mathrm{P}_1 V + Z_1$ and $Y = \mathrm{P}_2 V + Z_2$, where $\mathrm{P}_1, \mathrm{P}_2 \in \mathbb{R}^{d \times 2}$ are projection matrices (with i.i.d. normal entries). We draw $m = 10^3$ pairs of projection matrices $\{(\mathrm{A}_j, \mathrm{B}_j)\}_{j=1}^m$, and use the classic $k$-NN MI estimator of [38] with $n = 16 \times 10^3$ samples of $(X, Y)$ to approximate the MI along each projection pair, i.e., for each $j = 1, \dots, 10^3$, we compute

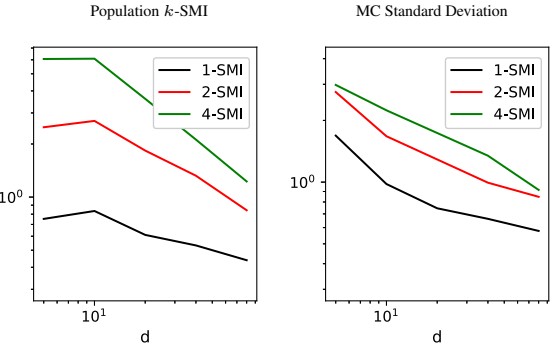

Figure 1: Decay with dimension the population $k$-SMI (left) and the associated MC standard deviation (right).

$\mathsf{I}\big((\mathrm{A}_j^\mathsf{T} X)^n, (\mathrm{B}_j^\mathsf{T} Y)^n\big)$. Note that the mean of $\mathsf{I}\big((\mathrm{A}_j^\mathsf{T} X)^n, (\mathrm{B}_j^\mathsf{T} Y)^n\big)$ is the population $k$-SMI (which, in this Gaussian example, is given by Theorem 3), while its standard deviation is the constant in front of the $m^{-1/2}$ term in (3) of Theorem 1. Figure 1 plots the said mean and standard deviation of the projected MI terms $\mathsf{I}\big((\mathrm{A}_j^\mathsf{T} X)^n, (\mathrm{B}_j^\mathsf{T} Y)^n\big)$. The rates of decay in both cases follow those predicted by Theorems 3 and 1, respectively. This implies that $m$ need not be rapidly scaled up, even as the population $k$-SMI shrinks with increasing dimension.

**Independence testing.** It was shown in [17] that SMI can be used for independence testing between high-dimensional variables, when classic MI is too costly to estimate. We revisit this experiment with $k$-SMI to demonstrate similar scalability and understand the effect of $k$. The test estimates $k$-SMI based on $n$ samples from $\mu_{XY}$ and then thresholds the value to declare dependence/independence.

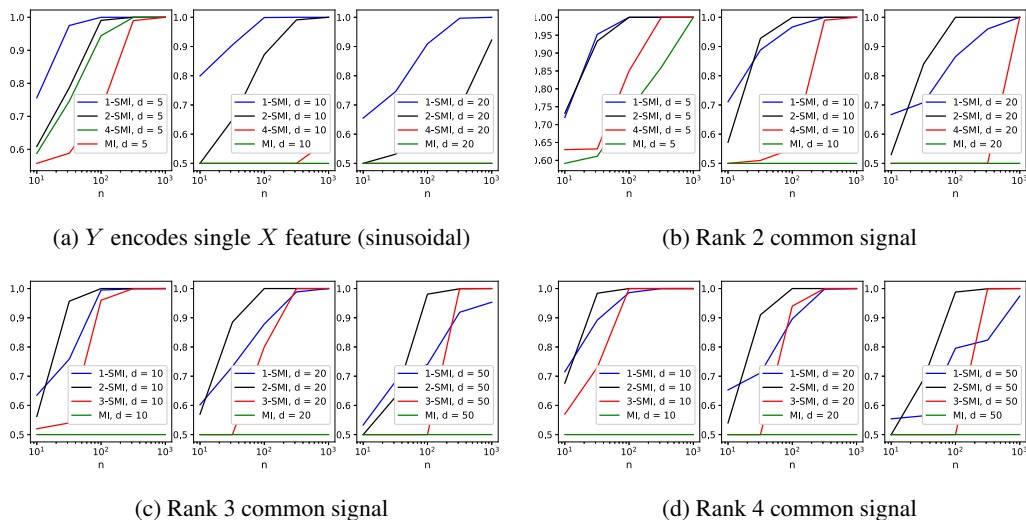

Figure 2: Independence testing with $k$-SMI: AUC ROC versus sample size $n$ for different $k$ and $d$.

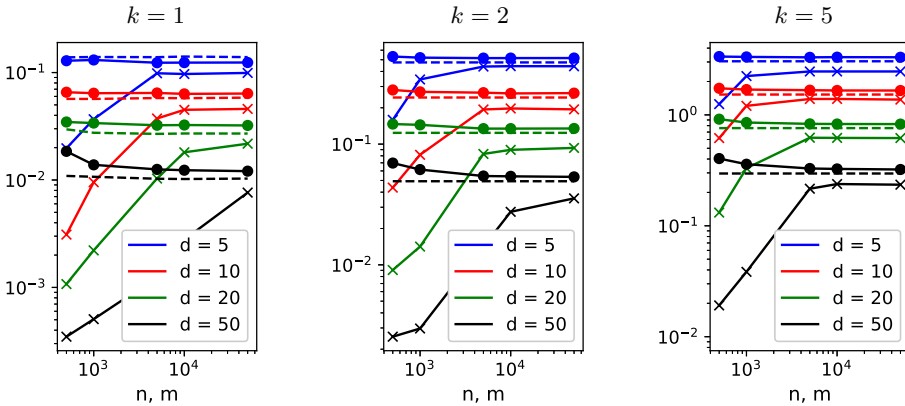

Figure 3: Neural estimation rates: Dashed line shows the ground truth, circle line is the value of the parallel neural estimator from Section 4.2, and the cross line is the SMI neural estimator from [17]. The parallel neural estimator converges at a faster rate for all considered $k$ and $d$.

Two types of models for $(X, Y)$ are considered: (i) $X, Z \sim \mathcal{N}(0, \mathrm{I}_d)$ are independent and $Y = \frac{1}{\sqrt{2}}\left(\frac{1}{\sqrt{d}}\sin(\mathbf{1}^\mathsf{T}X)\mathbf{1} + Z\right)$ (i.e., $X$ and $Y$ share one sinusoidal feature), and (ii) the rank 2 common signal model from the previous paragraph, as well as its extension to ranks 3 and 4. Figure 2 at the bottom of the previous page shows the area under the curve (AUC) of the receiver operating characteristic (ROC) as a function of $n$ for each of those models. Figure 2(a) shows the results for Model (i), while Figures 2(b)-(c) corresponds to Model (ii) with ranks 2, 3, and 4, respectively. The estimator $\widehat{\mathsf{SI}}_k^{m,n}$ from (2) is realized with $m = 1000$ and $\hat{\mathsf{I}}(\cdot, \cdot)$ as the Kozachenko–Leonenko estimator [38]; the AUC ROC curves are computed from 100 random trials. For Figures 4(a) and 4(b), we vary the ambient dimension as $d = 5, 10, 20$, while the projection dimension is $k = 1, 2, 4, d$; note that $k = 1$ corresponds to the SMI from [17] and $k = d$ to classic MI. In Figures 4(c) and 4(d) we consider, respectively, a common signal of rank 3 and 4. The ambient dimension is varied as $d = 10, 20, 50$, while the projection dimension is $k = 1, 2, 3, d$. Evidently, $k$-SMI-based tests perform well even when $d$ is large, while tests using classic MI fail. 1-SMI has a clear advantage in the model from Figure 4(a), where the common signal is 1-dimensional, but this is no longer the case for the models from Figures 4(b)-(d), where the shared structure is of higher dimension. Indeed, in Figure 4(b) we see that 2-SMI generally presents the best performance as it can better capture the underlying structure. For Figures 4(c) and 4(d), 3-SMI slightly outperforms 2-SMI for larger sample sizes, particularly in higher dimension. This highlights the potential gain of using higher $k$ values (to retain more information about the original signal, albeit at the cost of higher sample complexity) and the importance of adapting them to the intrinsic dimensionality of the model.

**Neural estimation.** Figure 3 (on the next page) illustrates the convergence of the $k$-SMI neural estimator[3] from Section 4.2 as $n = m$ increase together, for $X = Y \sim \mathcal{N}(0, \mathrm{I}_d)$. For comparison, we include the original neural estimator of [17], which uses a single neural net to approximate a shared DV potential.[4] While both neural estimators eventually converge to the ground truth, our parallel implementation converges much faster. Again note the clear decay of the true $k$-SMI as $d$ increases.

**Sliced InfoGAN.** We demonstrate a simple application of $k$-SMI to modern machine learning. Recall the InfoGAN [6]—a GAN variant that learns disentangled latent factors by maximizing a neural estimator of the MI between those factors and the generated samples. Figure 4(left) shows InfoGAN results for MNIST,[5] where 3 latent codes $(C_1, C_2, C_3)$ were used for disentanglement, with $C_1$ being a 10-state discrete variable and $(C_2, C_3)$ being continuous variables with values in $[-2, 2]$. The shown images are generated by the trained InfoGAN, where each row of corresponds to

---

[3]$m$ parallel 3-layer ReLU NNs were used, each with $30 \cdot k$ hidden units in each layer.

[4]A 3-layer ReLU NN was used with $20 \cdot d$ hidden units in each layer.

[5]Used experiment and code from https://github.com/Natsu6767/InfoGAN-PyTorch.

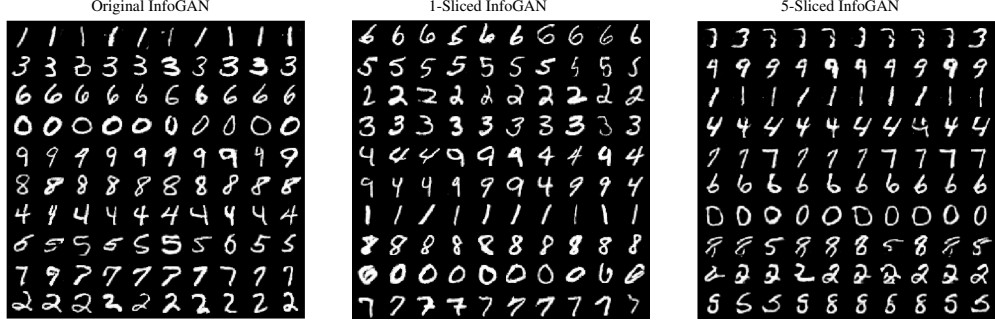

Figure 4: MNIST images generated via InfoGAN using neural estimators of MI (left), 1-SMI (middle), and 5-SMI (right). The latent codes $C_1$ (encodes digits) is varied across rows, while columns correspond to (random) $C_2, C_3$ values. In all three cases, the latent codes are successfully disentangled.

a different values the discrete $C_1$, while columns corresponds to random $C_2, C_3$ values. Despite being completely unsupervised, $C_1$ has been successfully disentangled to encode the digits 0-9. Figure 4(middle) shows the resulting generated images when the neural estimator for MI is replaced with a neural 1-SMI estimator with $m = 10^3$, and Figure 4(right) for 5-SMI. Evidently, 1-SMI and 5-SMI successfully disentangle the latent factors, despite seeing only $10^3$ 1- (respectively 5-) dimensional projections of this very high-dimensional data.

## 6 Summary and Concluding Remarks

This paper introduced $k$-SMI as a measure of statistical dependence defined by averaging MI terms between $k$-dimensional projections of the considered random variables. Our objective was to quantify and provide a rigorous justification for the perceived scalability of sliced information measures. We have done so by studying MC-based estimators of $k$-SMI, neural estimation methods, and asymptotics of $\mathsf{SI}_k(X; Y)$ under the Gaussian setting. Throughout, results with explicit dependence on $k, d_x, d_y$ were provided, revealing different gains associated with slicing, from the anticipated scalability to relaxed smoothness assumptions needed for neural estimation. Numerical experiments supporting our theory were provided, as well as a more advanced application to sliced infoGAN, showing that $k$-SMI can successfully replace classic MI even in applications with more intricate underlying structure.

Future research directions, both theoretical and applied, are abundant. In particular, we seek to derive sharp rates of decay of the residual term in (4), thereby establishing the Gaussian $k$-SMI as the leading term in that decomposition. Extensions of our results to the case when the projection dimensions for $X$ and $Y$ are different, i.e., $k_1 \neq k_2$, may allow further flexibility and are also of interest. We also plan to explore non-linear dimensionality reduction maps, as in the generalized sliced Wasserstein distance setting [39], as well as non-uniform distributions over parameterizations of the projection functions (cf. [40]). The max-SMI, where instead of averaging over $(A, B)$ we maximize over them, is another interesting avenue. On the application side, there are various machine learning models that utilize MI [6–8, 10]; revisiting those with $k$-SMI is an appealing endeavor due to the expected gains from slicing and the formal guarantees our theory can provide for those systems.

### Acknowledgments and Disclosure of Funding

Z. Goldfeld is partially supported by NSF grants CCF-1947801, CCF-2046018, and DMS-2210368, and the 2020 IBM Academic Award. G. Reeves is partially supported by NSF grant CCF-1750362.

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
