# Supplementary Materials for:
# $k$-Sliced Mutual Information: A Quantitative Study of Scalability with Dimension

## A    Proofs of Results in the Main Text

### A.1    Proofs for Section 2.2

The HWI inequality of Otto and Villani [22] is a functional inequality relating the entropy (H), quadratic transportation cost (W), and Fisher information (I), all defined w.r.t. a suitable reference measure that has bounded curvature. In deriving the classic result, a more general version of the HWI was established in [22]—one that is particularly well suited to the application in this paper, where we consider the differences between two entropy terms. See also [23, Proposition 1.5] for a recent derivation of the generalized inequality via a different argument based on an entropic interpolation of Wasserstein geodesics.

The result reads as follows: let $\gamma_\sigma = \mathcal{N}(0, \sigma^2 \mathrm{I}_d)$ denote the isotropic Gaussian measure on $\mathbb{R}^d$ with variance $\sigma^2$ and consider $\mu, \nu \in \mathcal{P}_2(\mathbb{R}^d)$ with $\mathsf{D}(\nu \| \gamma_\sigma) < \infty$ for some $\sigma > 0$, then

$$\mathsf{D}(\mu \| \gamma_\sigma) - \mathsf{D}(\nu \| \gamma_\sigma) \leq \mathsf{W}_2(\mu, \nu) \sqrt{\mathsf{J}(\mu \| \gamma_\sigma)} - \frac{1}{2\sigma^2} \mathsf{W}_2^2(\mu, \nu). \tag{5}$$

This HWI inequality is used to prove Lemma 1, from which the Lipschitz continuity in Proposition 1 readily follows.

*Proof of Lemma 1.*  If $\mu$ has finite Fisher information then we have the well-known identities

$$\mathsf{D}(\mu \| \gamma_\sigma) = \frac{n}{2} \log(2\pi\sigma^2) + \frac{1}{2\sigma^2} \mathbb{E}_\mu \|X\|^2 - \mathsf{h}(\mu)$$

$$\mathsf{J}(\mu \| \gamma_\sigma) = \mathsf{J}(\mu) - \frac{2n}{\sigma^2} + \frac{1}{\sigma^4} \mathbb{E}_\mu \|X\|^2.$$

Making the change of variables $\lambda = \sigma^{-2}$ and swapping the roles of $\mu$ and $\nu$ leads to the following bound on the difference in differential entropy:

$$\mathsf{h}(\mu) - \mathsf{h}(\nu) \leq \left(\mathsf{J}(\nu) - 2n\lambda + \lambda^2 \mathbb{E}_\nu \|X\|^2\right)^{\frac{1}{2}} \mathsf{W}_2(\mu, \nu) - \frac{\lambda}{2} \mathsf{W}_2^2(\mu, \nu) + \frac{\lambda}{2} \left(\mathbb{E}_\mu \|X\|^2 - \mathbb{E}_\nu \|X\|^2\right).$$

As the left-hand side (LHS) does not depend on $\lambda$, by taking the $\lambda \to 0^+$ we obtain

$$\mathsf{h}(\mu) - \mathsf{h}(\nu) \leq \sqrt{\mathsf{J}(\nu)} \mathsf{W}_2(\mu, \nu) \tag{6}$$

as desired. To see that the constant cannot be improved, evaluate the above bound for $\mu = \gamma_a$ and $\nu = \gamma_b$, and consider the limiting case of $a/b \to 1^+$:

$$\lim_{\frac{a}{b} \to 1^+} \frac{\mathsf{h}(\gamma_a) - \mathsf{h}(\gamma_b)}{\sqrt{\mathsf{J}(\gamma_b)} \mathsf{W}(\gamma_a, \gamma_b)} = \lim_{\frac{a}{b} \to 1^+} \frac{\log\left(\frac{a}{b}\right)}{\left(\frac{a}{b} - 1\right)} = 1.$$

$\square$

*Proof of Proposition 1.*  Because differential entropy is translation invariant we may assume without loss of generality that $\mu$ has zero mean. From the definition of the 2-Wasserstein distance, we obtain

$$\mathsf{W}_2(\mathfrak{p}_\sharp^A \mu, \mathfrak{p}_\sharp^B \mu) \leq \|\Sigma_\mu^{1/2}(A - B)\|_{\mathrm{F}}. \tag{7}$$

Combining this with (6) gives the first result.

To obtain a uniform bound on the Lipschitz constant, we use the fact that $J_F(\mathfrak{p}^B_\sharp \mu) \preceq B^\intercal J_F(\mu) B$ for any matrix B with orthogonal columns (cf. [41, Equation (67)]). Thus, both $\|J_F(\mathfrak{p}^A_\sharp \mu)\|_{op}$ and $\|J_F(\mathfrak{p}^B_\sharp \mu)\|_{op}$ are bounded from above by the sum of the $k$ largest eigenvalues of $J_F(\mu)$, and so

$$\|J_F(\mathfrak{p}^A_\sharp \mu)\|_{op} \vee \|J_F(\mathfrak{p}^B_\sharp \mu)\|_{op} \leq k\|J_F(\mu)\|_{op}.$$

Combining this with $\|\Sigma_\mu^{1/2}(A - B)\|_F \leq \|\Sigma_\mu\|_{op}\|A - B\|_F$ in (7) completes the proof. $\qquad\square$

## A.2 Proof of Proposition 2

**Proof of 1.** Non-negativity follows because $k$-SMI is an average of classic MI terms, which are non-negative. Nullification of $k$-SMI between independent $(X, Y)$ is also straightforward, since in this case $(A^\intercal X, B^\intercal Y)$ are independent for all $(A, B) \in St(k, d_x) \times St(k, d_y)$, which implies $I(A^\intercal X; B^\intercal Y) = 0$, i.e., the integrand in the $k$-SMI definition is identically zero. For the opposite implication, as will be shown below, we have $SI(X; Y) \leq SI_k(X; Y)$, for any $1 \leq k \leq d_x \wedge d_y$. Hence, if $SI_k(X; Y) = 0$ then $SI(X; Y) = 0$ and by Proposition 1 form [17] we have that $(X, Y)$ are independent.

**Proof of 2.** Throughout this proof we use our standard matrix notation (non-italic letter, such as A) to designate random matrices; for fixed matrices we add a tilde, e.g., $\tilde{A}$. Fix $1 \leq k_1 < k_2 \leq d_x \wedge d_y$ and let $(A_1, B_1) \sim \sigma_{k_1, d_x} \otimes \sigma_{k_1, d_y}$ and $(A_2, B_2) \sim \sigma_{k_2, d_x} \otimes \sigma_{k_2, d_y}$. For each $\tilde{A}_2 \in St(k_2, d)$, represent it as $\tilde{A}_2 = [\tilde{A}_{21} \ \tilde{A}_{22}]$, where $\tilde{A}_{21} \in St(k_1, d)$ and $\tilde{A}_{22} \in St(k_2 - k_1, d)$, and similarly for $\tilde{B}_2$. We now have

$$SI_{k_2}(X; Y)$$
$$= I(A_2^\intercal X; B_2^\intercal Y | A_2, B_2)$$
$$= I(A_{21}^\intercal X; B_{21}^\intercal Y | A_2, B_2) + I(A_{22}^\intercal X; B_2^\intercal Y | A_2, B_2, A_{21}^\intercal X) + I(A_{21}^\intercal X; B_{22}^\intercal Y | A_2, B_2, B_{21}^\intercal Y)$$
$$\geq I(A_1^\intercal X; B_1^\intercal Y | A_1, B_1)$$
$$= SI_{k_1}(X; Y),$$

where the inequality uses the non-negativity of (conditional) MI and the fact that

$$I(A_{21}^\intercal X; B_{21}^\intercal Y | A_2, B_2) = I(A_{21}^\intercal X; B_{21}^\intercal Y | A_{21}, B_{21}) = I(A_1^\intercal X; B_1^\intercal Y | A_1, B_1).$$

Indeed, the latter holds since $(A_{22}, B_{22})$ are marginalized out in the conditioning and because $(A_{21}, B_{21}) \stackrel{d}{=} A_1, B_1 \sim \sigma_{k_1, d_x} \otimes \sigma_{k_1, d_y}$.

Lastly, supremizing the integrand in the $k$-SMI definition over all pair of matrices from the Stiefel manifold, we further obtain

$$SI_{k_2}(X; Y) = I(A_2^\intercal X; B_2^\intercal Y | A_2, B_2) \leq \sup_{(\tilde{A}, \tilde{B}) \in St(k_2, d_x) \times St(k_2, d_y)} I(\tilde{A}^\intercal X; \tilde{B}^\intercal Y),$$

which concludes the proof.

**Proof of 3.** This follows because conditional mutual information can be expressed as

$$I(X; Y | Z) = \mathbb{E}_{\mu_Z}\left[ D_{KL}\left(\mu_{X,Y|Z}(\cdot|Z) \big\| \mu_{X|Z}(\cdot|Z) \otimes \mu_{Y|Z}(\cdot|Z)\right)\right],$$

and because the joint distribution of $(A^\intercal X, B^\intercal Y)$ given $\{A = \tilde{A}, B = \tilde{B}\}$, for fixed $(\tilde{A}, \tilde{B}) \in St(k, d_x) \times St(k, d_y)$, is $(\mathfrak{p}^{\tilde{A}}, \mathfrak{p}^{\tilde{B}})_\sharp \mu_{X,Y}$, while the corresponding conditional marginals are $\mathfrak{p}^{\tilde{A}}_\sharp \mu_X$ and $\mathfrak{p}^{\tilde{B}}_\sharp \mu_Y$, respectively. Hence,

$$SI_k(X; Y) = D\left((\mathfrak{p}^A, \mathfrak{p}^B)_\sharp \mu_{XY} \big\| (\mathfrak{p}^A, \mathfrak{p}^B)_\sharp \mu_X \otimes \mu_Y \big| A, B\right) = D\left(\mu_{A,B,A^\intercal X, B^\intercal Y} \big\| \mu_{A,B,A^\intercal \tilde{X}, B^\intercal \tilde{Y}}\right)$$

where the second step follows from the relative entropy chain rule, with

$$(A, B, A^\intercal X, B^\intercal Y) \sim \mu_{A,B,A^\intercal X, B^\intercal Y} = \sigma_{k, d_x} \otimes \sigma_{k, d_y} (\mathfrak{p}^A, \mathfrak{p}^B)_\sharp \mu_{XY}$$
$$(A, B, A^\intercal \tilde{X}, B^\intercal \tilde{Y}) \sim \mu_{A,B,A^\intercal \tilde{X}, B^\intercal \tilde{Y}} = \sigma_{k, d_x} \otimes \sigma_{k, d_y} (\mathfrak{p}^A, \mathfrak{p}^B)_\sharp \mu_X \otimes \mu_Y.$$

The variational form follows by applying the DV representation of relative entropy to the latter expression for $SI_k(X; Y)$ (see Section 4.2).

**Proof of 4.** Recall the definition of marginal and conditional $k$-sliced entropies: $\mathsf{sh}_k(X) := \mathsf{h}(\mathrm{A}^\intercal X|\mathrm{A})$ and $\mathsf{sh}_k(X|Y) := \mathsf{h}(\mathrm{A}^\intercal X|\mathrm{A}, \mathrm{B}, \mathrm{B}^\intercal Y)$. Given the representation of $k$-SMI as a conditional mutual information, we now have

$$\mathsf{SI}_k(X;Y) = \mathsf{I}(\mathrm{A}^\intercal X; \mathrm{B}^\intercal Y|\mathrm{A}, \mathrm{B}) = \mathsf{h}(\mathrm{A}^\intercal X|\mathrm{A}) - \mathsf{h}(\mathrm{A}^\intercal X|\mathrm{A}, \mathrm{B}, \mathrm{B}^\intercal Y),$$

where we have used independence of $\mathrm{B}$ and $(\mathrm{A}, \mathrm{A}^\intercal X)$ in the first conditional entropy term. The other decompositions follow in a similar fashion.

**Proof of 5.** We only prove the small chain rule; generalizing to $n$ variables is straightforward. Consider:

$$\begin{aligned}
\mathsf{SI}_k(X, Y; Z) &= \mathsf{I}(\mathrm{A}^\intercal X, \mathrm{B}^\intercal Y; \mathrm{C}^\intercal Z|\mathrm{A}, \mathrm{B}, \mathrm{C}) \\
&= \mathsf{I}(\mathrm{A}^\intercal X; \mathrm{C}^\intercal Z|\mathrm{A}, \mathrm{B}, \mathrm{C}) + \mathsf{I}(\mathrm{B}^\intercal Y; \mathrm{C}^\intercal Z|\mathrm{A}, \mathrm{B}, \mathrm{C}, \mathrm{A}^\intercal X),
\end{aligned}$$

where the last equality is the regular chain rule. Since $(X, Z, \mathrm{A}, \mathrm{C})$ are independent of $\mathrm{B}$, we have

$$\mathsf{I}(\mathrm{A}^\intercal X; \mathrm{C}^\intercal Z|\mathrm{A}, \mathrm{B}, \mathrm{C}) = \mathsf{I}(\mathrm{A}^\intercal X; \mathrm{C}^\intercal Z|\mathrm{A}, \mathrm{C}) = \mathsf{SI}_k(X; Z),$$

while $\mathsf{I}(\mathrm{B}^\intercal Y; \mathrm{C}^\intercal Z|\mathrm{A}, \mathrm{B}, \mathrm{C}, \mathrm{A}^\intercal X) = \mathsf{SI}_k(Y; Z|X)$ by definition.

**Proof of 6.** By definition,

$$\mathsf{SI}_k(X_1, \ldots, X_n; Y_1, \ldots, Y_n) = \mathsf{SI}_k(\mathrm{A}_1^\intercal X_1, \ldots, \mathrm{A}_n^\intercal X_n; \mathrm{B}_1^\intercal Y_1, \ldots, \mathrm{B}_n^\intercal Y_n|\mathrm{A}_1, \ldots, \mathrm{A}_n, \mathrm{B}_1, \ldots, \mathrm{B}_n),$$

where the $\mathrm{A}_i$, $\mathrm{B}_i$ are all independent and uniform on the respective Stiefel manifolds. Now, by mutual independence of the $\mathrm{A}_i$, $\mathrm{B}_i$ and $(X_i, Y_i)$ across $i$ and tensorization of MI, we have

$$\begin{aligned}
\mathsf{I}(\mathrm{A}_1^\intercal X_1, \ldots, \mathrm{A}_n^\intercal X_n; \mathrm{B}_1^\intercal Y_1, \ldots, \mathrm{B}_n^\intercal Y_n|\mathrm{A}_1, \ldots, \mathrm{A}_n, \mathrm{B}_1, \ldots, \mathrm{B}_n) &= \sum_{i=1}^n \mathsf{I}(\mathrm{A}_i^\intercal X_i; \mathrm{B}_i^\intercal Y_i|\mathrm{A}_i, \mathrm{B}_i) \\
&= \sum_{i=1}^n \mathsf{SI}_k(X_i; Y_i).
\end{aligned}$$

$\square$

## A.3 Proof of Theorem 1

The proof of Theorem 1 relies on the following technical lemmas concerning the Lipschitzness and variance of the function $i_{XY} : \mathrm{St}(k, d_x) \times \mathrm{St}(k, d_y) \to \mathbb{R}$ defined as $i_{XY}(\mathrm{A}, \mathrm{B}) := \mathsf{I}(\mathrm{A}^\intercal X; \mathrm{B}^\intercal Y)$.

**Lemma 2** (Lipschitzness of projected MI). *For $\mu_{XY} \in \mathcal{P}_2(\mathbb{R}^{d_x} \times \mathbb{R}^{d_y})$ with $\mathsf{J}(\mu_{XY}) < \infty$, the function $i_{XY} : \mathrm{St}(k, d_x) \times \mathrm{St}(k, d_y) \to \mathbb{R}$ is Lipschitz with respect to the Frobenius norm on the Cartesian product of Stiefel manifolds, with Lipschitz constant*

$$L_k(\mu_{XY}) = 3\sqrt{2k\|\mathsf{J}_{\mathrm{F}}(\mu_{XY})\|_{\mathrm{op}}\big(\|\Sigma_X\|_{\mathrm{op}} \vee \|\Sigma_Y\|_{\mathrm{op}}\big)}.$$

*Proof.* Fixing $(\mathrm{A}_1, \mathrm{B}_1), (\mathrm{A}_2, \mathrm{B}_2) \in \mathrm{St}(k, d_x) \times \mathrm{St}(k, d_y)$, we have

$$\begin{aligned}
&\big|i_{XY}(\mathrm{A}_1, \mathrm{B}_1) - i_{XY}(\mathrm{A}_2, \mathrm{B}_2)\big| \\
&\quad \leq \big|\mathsf{h}(\mathrm{A}_1^\intercal X) - \mathsf{h}(\mathrm{A}_2^\intercal X)\big| + \big|\mathsf{h}(\mathrm{B}_1^\intercal Y) - \mathsf{h}(\mathrm{B}_2^\intercal Y)\big| + \big|\mathsf{h}(\mathrm{A}_1^\intercal X, \mathrm{B}_1^\intercal Y) - \mathsf{h}(\mathrm{A}_2^\intercal X, \mathrm{B}_2^\intercal Y)\big|. \quad (8)
\end{aligned}$$

The differences of marginal entropy terms (i.e., the first two terms on the right-hand side (RHS) above) are controlled by $\sqrt{k\|\Sigma_X\|_{\mathrm{op}}\|\mathsf{J}_{\mathrm{F}}(\mu_X)\|_{\mathrm{op}}}\|\mathrm{A}_1 - \mathrm{A}_2\|_{\mathrm{F}}$ and $\sqrt{k\|\Sigma_Y\|_{\mathrm{op}}\|\mathsf{J}_{\mathrm{F}}(\mu_Y)\|_{\mathrm{op}}}\|\mathrm{B}_1 - \mathrm{B}_2\|_{\mathrm{F}}$, respectively, by applying Proposition 1. For the difference of joint entropies, we shall use Lemma 1. To that end, note that

$$\begin{aligned}
\mathsf{W}_2^2\big((\mathfrak{p}^{\mathrm{A}_1}, \mathfrak{p}^{\mathrm{B}_1})_\sharp \mu_{XY}, (\mathfrak{p}^{\mathrm{A}_2}, \mathfrak{p}^{\mathrm{B}_2})_\sharp \mu_{XY}\big) &\leq \mathbb{E}\big[\|(\mathrm{A}_1 - \mathrm{A}_2)^\intercal X\|^2 + \|(\mathrm{B}_1 - \mathrm{B}_2)^\intercal Y\|^2\big] \\
&\leq \big(\|\Sigma_X\|_{\mathrm{op}} \vee \|\Sigma_X\|_{\mathrm{op}}\big)\big(\|\mathrm{A}_1 - \mathrm{A}_2\|_{\mathrm{F}}^2 + \|\mathrm{B}_1 - \mathrm{B}_2\|_{\mathrm{F}}^2\big),
\end{aligned}$$
$$(9)$$

and observe that the Fisher information of the projected joint distribution can be controlled by the operator norm of the corresponding Fisher information matrix. Indeed, the Fisher information data

processing inequality (cf. [41, Equation (67)]) states that for any $(A, B) \in \mathrm{St}(k, d_x) \times \mathrm{St}(k, d_y)$, we have

$$J\big((\mathfrak{p}^A, \mathfrak{p}^B)_\sharp \mu_{XY}\big) = \mathrm{tr}\left(\begin{bmatrix} A^\intercal & 0 \\ 0 & B^\intercal \end{bmatrix} J_F(\mu_{XY}) \begin{bmatrix} A & 0 \\ 0 & B \end{bmatrix}\right) \leq 2k\|J_F(\mu_{XY})\|_{\mathrm{op}}. \qquad (10)$$

Invoking Lemma 1, while using the above along with (9), gives

$$\big|h(A_1^\intercal X, B_1^\intercal Y) - h(A_2^\intercal X, B_2^\intercal Y)\big|$$

$$\leq \Big(2k\|J_F(\mu_{XY})\|_{\mathrm{op}}\big(\|\Sigma_X\|_{\mathrm{op}} \vee \|\Sigma_Y\|_{\mathrm{op}}\big)\Big)^{1/2} \big(\|A_1 - A_2\|_F^2 + \|B_1 - B_2\|_F^2\big)^{1/2}.$$

Together with the marginal entropy bounds the fact that $J_F(\mu_X) \vee J_F(\mu_Y) \preceq J_F(\mu_{XY})$ (which also follows from the data processing inequality), this implies the result. $\qquad \square$

**Lemma 3** (Variance bound). *Let* $(A, B) \sim \sigma_{k,d_x} \otimes \sigma_{k,d_y}$*, then we have the variance bound*

$$\mathrm{Var}\big(i_{XY}(A, B)\big) \leq 24 L_k^2(\mu_{XY}) \big(d_x^{-1} + d_y^{-1}\big),$$

*where* $L_k(\mu_{XY})$ *is defined in Lemma 2.*

*Proof.* Recall that the special orthogonal group $\mathbb{SO}(d) = \{U \in \mathbb{R}^{d \times d} : U^\intercal U = I_d, \det(U) = 1\}$ is the set of $d \times d$ orthogonal matrices with determinant one. The following result is consequence of concentration of measure on compact Riemannian manifolds (see Section 5 in [42]).

**Lemma 4.** *Let* $f \colon \mathbb{SO}(d) \to \mathbb{R}$ *be Lipschitz continuous with respect to the Frobenius norm with Lipschitz constant* $L$*, i.e.,* $|f(U) - f(V)| \leq L\|U - V\|_F$ *for all* $U, V \in \mathbb{SO}(d)$*. If* $d \geq 3$ *and* $U$ *is distributed uniformly on* $\mathbb{SO}(d)$ *then* $f(U)$ *is sub-Gaussian with parameter* $\sigma^2 = 4L^2/(d-2)$*, i.e.,*

$$\log \mathbb{E}\left[e^{\lambda(f(U) - \mathbb{E}f(U))}\right] \leq \frac{\lambda^2 \sigma^2}{2}, \quad \forall \lambda > 0.$$

*In particular, this implies that* $\mathrm{Var}\big(f(U)\big) \leq \sigma^2$.

For our purposes, this result provides concentration bounds with respect to functions defined on the Stiefel manifold. Observe that if $U = [u_1 \ldots u_d]$ is uniform on $\mathbb{SO}(d)$ then the $d \times k$ matrix $A = [u_1 \ldots u_k]$ is uniform on $\mathrm{St}(k, d)$. Thus, if $g$ is a real-valued function on $\mathrm{St}(k, d)$ that is Lipschitz continuous with constant $L$, we can apply the above result to $f(U) = g(U[I_k, 0]^\intercal)$ to conclude that $g(A)$ is sub-Gaussian with parameter $4L^2/(d-2)$, and hence

$$\mathrm{Var}\big(g(A)\big) \leq \frac{4L^2}{d-2}. \qquad (11)$$

Now, to bound the variance of $i_{XY}$, recall that $(A, B) \sim \sigma_{k,d_x} \otimes \sigma_{k,d_y}$ are independent and uniformly distributed random matrices from the corresponding Stiefel manifold. By the Efron-Stein inequality (cf. e.g., [43, Theorem 3.3.7]), the variance satisfies

$$\mathrm{Var}\big(i_{XY}(A, B)\big) \leq \mathbb{E}\big[\mathrm{Var}\big(i_{XY}(A, B)\big|B\big)\big] + \mathbb{E}\big[\mathrm{Var}\big(i_{XY}(A, B)\big|A\big)\big].$$

Since $i_{XY}(\cdot, \cdot)$ is Lipschitz continuous in each of its arguments with the same constant, it follows from Lemma 2 that the terms on the RHS are bounded from above by $(2L_k(\mu_{XY}))^2/(d_x - 2)$ and $(2L_k(\mu_{XY}))^2/(d_y - 2)$, respectively.

As the above requires $d_x, d_y > 2$, we further note that for $A'$ an independent copy of $A$, we have

$$\mathbb{E}\big[\mathrm{Var}\big(i_{XY}(A, B)\big|B\big)\big] = \frac{1}{2}\mathbb{E}\left[\big|i_{XY}(A, B) - i_{XY}(A', B)\big|^2\right]$$

$$\leq \frac{L_k^2(\mu_{XY})}{2}\mathbb{E}\|A_1 - A_2\|_F^2 \leq L_k^2(\mu_{XY}) k d_x.$$

Since $k \leq d_x - 1$, it follows that

$$\mathbb{E}\big[\mathrm{Var}\big(i_{XY}(A, B)\big|B\big)\big] \leq \left(d_x^2 - d_x \wedge \frac{4}{d_x - 2}\right) L_k^2(\mu_{XY}) \leq \frac{12 L_k^2(\mu_{XY})}{d_x},$$

and similarly for $\mathbb{E}\big[\mathrm{Var}\big(i_{XY}(A, B)\big|A\big)\big]$ with $d_y$ replacing $d_x$. The conclusion of Lemma 3 follows. $\qquad \square$

*Proof of Theorem 1.* Since $k$-SMI is invariant to translation (due to bijection invariance of MI), we may assume without loss of generality that $X$ and $Y$ are centered. The error is now decomposed as

$$\mathbb{E}\left[\left|\mathsf{SI}_k(X;Y) - \widehat{\mathsf{SI}}_k^{m,n}\right|\right]$$
$$\leq \mathbb{E}\left[\left|\mathsf{SI}_k(X;Y) - \frac{1}{m}\sum_{i=1}^m i_{XY}(\mathrm{A}_i, \mathrm{B}_i)\right|\right] + \mathbb{E}\left[\left|\frac{1}{m}\sum_{i=1}^m i_{XY}(\mathrm{A}_i, \mathrm{B}_i) - \widehat{\mathsf{SI}}_k^{m,n}\right|\right].$$

The first term on the RHS above corresponds to the MC error. By observing that $\mathsf{SI}_k(X;Y) = \mathbb{E}\left[\frac{1}{m}\sum_{i=1}^m i_{XY}(\mathrm{A}_i, \mathrm{B}_i)\right]$ and using monotonicity of moments, we may upper bound it by $\left(\mathrm{Var}\big(i_{XY}(\mathrm{A}, \mathrm{B})\big)/m\right)^{1/2}$. Lemma 3 then provides a bound on the variance.

The second term above is controlled by the $k$-dimensional MI estimation error $\delta_k(n)$ from Assumption 1, since

$$\mathbb{E}\left[\left|\frac{1}{m}\sum_{i=1}^m i_{XY}(\mathrm{A}_i, \mathrm{B}_i) - \widehat{\mathsf{SI}}_k^{m,n}\right|\right] \leq \sup_{\substack{\mathrm{A}\in\mathrm{St}(k,d_x) \\ \mathrm{B}\in\mathrm{St}(k,d_y)}} \mathbb{E}\left[\left|i_{XY}(\mathrm{A}, \mathrm{B}) - \hat{\mathsf{I}}\big((\mathrm{A}^\intercal X)^n, (\mathrm{B}^\intercal Y)^n\big)\right|\right] \leq \delta_k(n).$$

Combining the two bounds produces the result. $\qquad\square$

## A.4 Proof of Theorem 2

The proof utilizes the result of Theorem 4 from [29] for relative entropy neural estimation along with the sufficient conditions given in Proposition 7 therein (cf. [29, Section 4.1.1] for comments on the applicability of their Theorem 4 to the DV variational form). For completeness, we first restate those results. Denote $\|\mathcal{Z}\|_\infty := \sup_{z\in\mathcal{Z}} \|z\|_\infty$.

**Proposition 3** (Sufficient conditions for relative entropy neural estimation (Theorem 4 and Proposition 2 of [29])). *Fix $d, b, M \geq 0$ and set $s = \lfloor d/2 \rfloor + 3$,. Let $\mathcal{Z} \subset \mathbb{R}^d$ be compact, and $\mu, \nu \in \mathcal{P}_{\mathsf{ac}}(\mathcal{Z})$ have densities $f_\mu, f_\nu$ respectively. Suppose that $\mathsf{D}(\mu\|\nu) \leq M$ and that there exist $r_\mu, r_\nu \in \mathsf{C}_b^s(\mathcal{U})$ for some open set $\mathcal{U} \supset \mathcal{Z}$, such that $\log f_\mu = r_\nu|_\mathcal{Z}$ and $\log f_\mu = r_\nu|_\mathcal{Z}$. Then*

$$\mathbb{E}\left[\left|\mathsf{D}(\mu\|\nu) - \hat{\mathsf{D}}_{\mathcal{G}_{d,d}^\ell}(X^n, Y^n)\right|\right] \lesssim_{M,b,\|\mathcal{Z}\|_\infty} d^{\frac{1}{2}} l^{-\frac{1}{2}} + d^{\frac{3}{2}} n^{-\frac{1}{2}},$$

*where $\hat{\mathsf{D}}_{\mathcal{G}_{d,d}^\ell}(X^n, Y^n) := \sup_{g\in\mathcal{G}_{d,d}^\ell} \frac{1}{n}\sum_{i=1}^n g(X_i, Y_i) - \log\left(\frac{1}{n}\sum_{i=1}^n e^{g(X_i, Y_{\sigma(i)})}\right)$.*

We use the above result to establish the following lemma that accounts for neural estimation of each projected MI term. Given the lemma, the result of Theorem 2 follows by Theorem 1, with the RHS of (12) in place of the $\delta_k(n)$ term therein.

**Lemma 5** (Neural estimation of $i_{XY}(\mathrm{A}, \mathrm{B})$). *Let $\mu_{XY} \in \mathcal{F}_{d_x,d_y}^k(M, b)$. Then uniformly in $(\mathrm{A}, \mathrm{B}) \in \mathrm{St}(k, d_x) \times \mathrm{St}(k, d_y)$, we have the neural estimation bound*

$$\mathbb{E}\left[\left|i_{XY}(\mathrm{A}, \mathrm{B}) - \hat{\mathsf{I}}_{k,k}^\ell\big((\mathrm{A}_j^\intercal X)^n, (\mathrm{B}_j^\intercal Y)^n\big)\right|\right] \lesssim_{M,b,k,\|\mathcal{X}\times\mathcal{Y}\|} k^{\frac{1}{2}}\ell^{-\frac{1}{2}} + k^{\frac{3}{2}}n^{-\frac{1}{2}}. \tag{12}$$

*Proof.* The lemma is proven by showing that densities of $(\mathfrak{p}^\mathrm{A}, \mathfrak{p}^\mathrm{B})_\sharp \mu_{XY}$ and $\mathfrak{p}_\sharp^\mathrm{A}\mu_X \otimes \mathfrak{p}_\sharp^\mathrm{B}\mu_Y$ satisfy the conditions of Proposition 3, whenever $\mu_{XY} \in \mathcal{F}_{d_x,d_y}^k(M, b)$.

Let $f$ be the density of $\mu_{XY}$ and set $f_\theta$, with $\theta := (\theta_1, \theta_2) = (\mathrm{A}, \mathrm{B}) \in \mathrm{St}(k, d_x) \times \mathrm{St}(k, d_y)$ as the density of projection $(\mathfrak{p}^\mathrm{A}, \mathfrak{p}^\mathrm{B})_\sharp \mu_{XY}$ which is supported on $\mathcal{Z}$. Let $\mathrm{A} = [a_1 \dots a_k]$, where $a_i \in \mathbb{S}^{d_x-1}$ with $\langle a_i, a_j\rangle = 0$, $\forall i \neq j$, and denote $\mathcal{W}_x = \{w \in \mathbb{R}^{d_x} : \langle a_i, w\rangle = 0, \forall i = 1, \dots, k\}$. Similarly, for $\mathrm{B} = (b_1 \dots b_k)$, set $\mathcal{W}_y = \{w \in \mathbb{R}^{d_y} : \langle b_i, w\rangle = 0, \forall i = 1, \dots, k\}$.

The density $f_\theta$ is given by

$$f_\theta(z_x, z_y) = \int_{\mathcal{W}_x}\int_{\mathcal{W}_y} f(\mathrm{A}z_x + w_x, \mathrm{B}z_y + w_y)\, dw_x\, dw_y,$$

where we have denoted $z_{x,i} = \langle a_i, x\rangle$ and $z_{y,i} = \langle b_i, y\rangle$, for $i = 1, \dots, k$, and further defined $z_x = [z_{x,1} \dots z_{x,k}]^\intercal$ and $z_y = [z_{y,1} \dots z_{y,k}]^\intercal$

Given $\mu_{XY} \in \mathcal{F}_{d_x,d_y}^k(M,b)$, there exists $r \in \mathsf{C}_b^s(\mathcal{U})$ with $s = k+3$ for some open set $\mathcal{U} \supset \mathcal{X} \times \mathcal{Y}$, such that $\log f = r|_{\mathcal{X} \times \mathcal{Y}}$. Choose $\mathcal{U}' \supset \mathcal{Z}$ such that $\mathcal{U}'$ is the projection of the set $\mathcal{U}$ on to the projection directions specified by $A, B$. Then also set

$$r_1(z_x, z_y) = \log \int_{\mathcal{W}_x} \int_{\mathcal{W}_y} \exp\left(r(A z_x + w_x, B z_y + w_y)\right) dw_x \, dw_y.$$

which implies $r_1|_{\mathcal{Z}} = \log f_\theta$.

To evaluate the derivative, we use the short hand notation $r_1 := \log\left(\int \exp(r)\right)$, omitting the arguments of the functions $r, r_1$. Let $v \in \{z_{x,1}, \ldots, z_{x,k}, z_{y,1}, \ldots z_{y,k}\}$ and $u \in \{x_1, \ldots, x_{d_x}, y_1, \ldots, y_{d_y}\}$, and consider

$$
\begin{aligned}
\frac{\partial^s}{\partial^s v} r_1 &\overset{(a)}{=} \sum_{\mathcal{P}_m^s} \frac{s!}{m_1! \, m_2! \ldots m_s!} \frac{(-1)^{M_s-1} (M_s-1)!}{\left(\int \exp(r)\right)^{M_s}} \prod_{i=1}^s \frac{1}{(i!)^{m_i}} \left(\int \frac{\partial^i}{\partial^i v} \exp(r)\right)^{m_i} \\
&\overset{(b)}{=} \sum_{\mathcal{P}_m^s} \frac{s!}{m_1! \, m_2! \ldots m_s!} \frac{(-1)^{M_s-1}(M_s-1)!}{\left(\int \exp(r)\right)^{M_s}} \\
&\quad \times \prod_{i=1}^s \frac{1}{(i!)^{m_i}} \left(\int \exp(r) \sum_{\mathcal{P}_l^i} \frac{i!}{l_1! \, l_2! \ldots l_i!} \prod_{k=1}^i \frac{1}{(k!)^{l_k}} \left(\frac{\partial^k}{\partial^k v} r\right)^{l_k}\right)^{m_i} \\
&\overset{(c)}{\leq} \sum_{\mathcal{P}_m^s} \frac{s! \, (M_s-1)!}{m_1! \, m_2! \ldots m_s!} \prod_{i=1}^s \frac{1}{(i!)^{m_i}} \left(\frac{\int \exp(r) \sum_{\mathcal{P}_l^i} \frac{i!}{l_1! \, l_2! \ldots l_i!} \prod_{k=1}^i \frac{1}{(k!)^{l_k}} b^{l_k}}{\int \exp(r)}\right)^{m_i} \\
&= \sum_{\mathcal{P}_m^s} \frac{s! \, (M_s-1)!}{m_1! \, m_2! \ldots m_s!} \prod_{i=1}^s \frac{1}{(i!)^{m_i}} \left(\sum_{\mathcal{P}_l^i} \frac{i!}{l_1! \, l_2! \ldots l_i!} \prod_{k=1}^i \frac{1}{(k!)^{l_k}} b^{l_k}\right)^{m_i} \\
&\overset{(d)}{\leq} c_s(b \vee b^s)
\end{aligned}
$$

where:
(a) follows from Faà di Bruno's formula with $M_s = \sum_{i=1}^s m_i$ and $\mathcal{P}_m^s$ as the set of all $s$-tuples of non-negative integers $m_i$ satisfying $\sum_{i=1}^s i m_i = s$;
(b) uses the Faà di Bruno's formula for the function $\exp(r)$, with $\mathcal{P}_l^i$ defined similarly to $\mathcal{P}_m^s$;
(c) holds since $\left|\frac{\partial^k}{\partial v^k} r\right| \leq b$, which comes from the fact that $\left|\frac{\partial^k}{\partial v^k} r\right| \leq \left|\frac{\partial^k}{\partial u^k} r\right| \leq b$ for $k \leq s$; the latter is a consequence of $r$ being $s$-times differentiable with derivatives bounded by $b$ and since $\left|\frac{\partial^k}{\partial v^k} u\right| \leq 1$, which holds because $x = A z_x + w_x, y = B z_y + w_y$ and thus $\frac{\partial}{\partial v} u$ is a constant (i.e., independent of $v$) upper bounded 1;
(d) identifies the dominating term as $b^{\sum_{i=1}^s \sum_{k=1}^i l_k m_i} \leq b \vee b^s$ and uses $c_s$ for a constant that depends only on $s$.

Conclude that $r_1 \in \mathsf{C}_{b^\star}^s(\mathcal{U}')$ with $b^\star = c_s(b \vee b^s)$.

Consider a similar derivation for the product of marginal densities. Let $f_{\theta_1}$ and $f_{\theta_2}$ denote the densities of $\mathfrak{p}_\sharp^A \mu_X$ and $\mathfrak{p}_\sharp^B \mu_Y$, respectively; the corresponding supports are $\mathcal{Z}_1$ and $\mathcal{Z}_2$, for which $\mathcal{Z} = \mathcal{Z}_1 \times \mathcal{Z}_2$. Following steps as above, we can show that $\exists \, r_{\theta_1} \in \mathsf{C}_{b^\star}^s(\mathcal{U}_1'), r_{\theta_2} \in \mathsf{C}_{b^\star}^s(\mathcal{U}_2')$ with $\mathcal{U}_1' \supset \mathcal{Z}_1, \mathcal{U}_2' \supset \mathcal{Z}_2$, such that $\log f_{\theta_1} = r_{\theta_1}|_{\mathcal{Z}_1}$ and $\log f_{\theta_2} = r_{\theta_2}|_{\mathcal{Z}_2}$.

As the density of $\mathfrak{p}_\sharp^A \mu_X \otimes \mathfrak{p}_\sharp^B \mu_Y$ is $f_{\theta_1} f_{\theta_2}$, we choose $r_2(z_x, z_y) = r_{\theta_1}(z_x) + r_{\theta_2}(z_y)$. Accordingly, $\log f_{\theta_1} f_{\theta_2} = r_2|_{\mathcal{Z}}$, and for $\mathcal{U}' = \mathcal{U}_1' \times \mathcal{U}_2' \supset \mathcal{Z}$, we have

$$\|D^\alpha r_2\|_{\infty,\mathcal{U}'} \leq \|D^\alpha r_{\theta_1}\|_{\infty,\mathcal{U}_1'} + \|D^\alpha r_{\theta_2}\|_{\infty,\mathcal{U}_2'} \leq 2b^\star.$$

This implies that $r_2 \in \mathsf{C}_{2b^\star}^s(\mathcal{U}')$, whereby $\mathfrak{p}_\sharp^A \mu_X \otimes \mathfrak{p}_\sharp^B \mu_Y \in \mathcal{F}_{d_x,d_y}^k(M, 2b^\star)$.

Since $\mathcal{Z} \subseteq \mathbb{R}^{2k}$ and $\mu_{XY} \in \mathcal{F}_{d_x,d_y}^k(M,b)$, the above shows that $(\mathfrak{p}^A, \mathfrak{p}^B)_\sharp \mu_{XY}$ and $\mathfrak{p}_\sharp^A \mu_X \otimes \mathfrak{p}_\sharp^B \mu_Y$ satisfy the smoothness requirement of Proposition 3 (the order should be at least $s = k+3$), with

an expansion of smoothness radius to $2c_{k+3}(b \vee b^{k+3})$. For $k = 1$ which corresponds to SMI, the expanded smoothness radius is $2b^\star = 154(b \vee b^4)$.

Lastly, we note that $\|\mathcal{Z}\|_\infty \leq \sup_{(x,y) \in \mathcal{X} \times \mathcal{Y}} \|A^\mathsf{T}x, B^\mathsf{T}y\| \leq \|\mathcal{X} \times \mathcal{Y}\|$, where the last inequality is due to sub-multiplicative property of $\ell^2$-norm and

$$\begin{bmatrix} A^\mathsf{T} & 0 \\ 0 & B^\mathsf{T} \end{bmatrix} \begin{bmatrix} A & 0 \\ 0 & B \end{bmatrix} = I_{2k},$$

which results in the corresponding operator norm being 1. This completes the proof of Lemma 5. $\square$

### A.5 Proof of Theorem 3

We begin by recalling the setting of Theorem 3 as well as some basic properties of mutual information for Gaussian distributions. Let $(X, Y) \sim \gamma_{XY} = \mathcal{N}(0, \Sigma_{XY})$ be jointly Gaussian random variables with positive definite covariance matrix

$$\Sigma_{XY} = \begin{pmatrix} \Sigma_X & C_{XY} \\ C_{XY}^\mathsf{T} & \Sigma_Y \end{pmatrix}$$

The assumption that the covariance is positive definite means that the singular values of the correlation matrix defined by $R := \Sigma_X^{-1/2} C_{XY} \Sigma_Y^{-1/2}$ are strictly less than one. The mutual information between $X$ and $Y$ depends only on the correlation matrix and is given by

$$I(X; Y) = -\frac{1}{2} \log \det(I_{d_x} - RR^\mathsf{T}).$$

Moreover, for a $d_x \times k$ matrix A and $d_y \times k$ matrix B, both with linearly independent columns, the mutual information between the $k$-dimensional Gaussian variables $A^\mathsf{T}X$ and $B^\mathsf{T}Y$ equals to

$$I(A^\mathsf{T}X; B^\mathsf{T}Y) = -\frac{1}{2} \log \det(I_k - \tilde{R}\tilde{R}^\mathsf{T}),$$

where $\tilde{R} = \tilde{A}^\mathsf{T}R\tilde{B}$ is the correlation matrix of the projected distribution and

$$\tilde{A} = \Sigma_X^{1/2} A (A^\mathsf{T} \Sigma_X A)^{-1/2}, \qquad \tilde{B} = \Sigma_Y^{1/2} B (B^\mathsf{T} \Sigma_Y B)^{-1/2} \tag{13}$$

The $k$-SMI is the expectation of this mutual information with respect to $(A, B)$ drawn from the uniform distribution on $\mathrm{St}(k, d_x) \times \mathrm{St}(k, d_y)$

**Remark 7.** *If $\Sigma_X$ and $\Sigma_Y$ are approximately low rank then $\tilde{A}$ and $\tilde{B}$ are concentrated low-dimensional subspaces, which may or may not align with the dominant directions in the correlation matrix R. Therefore, in contrast to the mutual information, the $k$-SMI depends not only on the correlation matrix R but also the marginal distributions of $X$ and $Y$.*

*Proof of Theorem 3.* The proof relies on several technical lemmas whose statements and proofs are deferred to the next section. The $k$-SMI for jointly Gaussian variables can be expressed as

$$\mathsf{SI}_k(X, Y) = -\frac{1}{2} \mathbb{E} \left[ \log \det(I_k - \tilde{R}\tilde{R}^\mathsf{T}) \right] \tag{14}$$

where $\tilde{R} = \tilde{A}^\mathsf{T}R\tilde{B}$ is the projected correlation matrix and $(\tilde{A}, \tilde{B})$ are defined as in (13) as a function of matrices $(A, B)$ drawn from the uniform distribution on $\mathrm{St}(k, d_x) \times \mathrm{St}(k, d_x)$. Note that $\tilde{A}$ and $\tilde{B}$ are both on the Stiefel manifold, and thus $\|\tilde{A}\|_{\mathrm{op}} = \|\tilde{B}\|_{\mathrm{op}} = 1$. Accordingly, the correlation matrix satisfies $\|\tilde{R}\|_{\mathrm{op}} \leq \|R\|_{\mathrm{op}} \leq \rho$ a.s. Applying Lemma 6 (see next section) to the positive definite matrix $\tilde{R}\tilde{R}^\mathsf{T}$ and then taking expectation yields

$$0 \leq \mathsf{SI}_k(X, Y) - \frac{1}{2} \mathbb{E} \|\tilde{R}\|_{\mathrm{F}}^2 \leq \frac{\mathbb{E} \|\tilde{R}\tilde{R}^\mathsf{T}\|_{\mathrm{F}}^2}{2(1 - \rho^2)}$$

To establish the desired result we will characterize the leading order terms in $\mathbb{E}\|\tilde{R}\|_{\mathrm{F}}^2$ and then show that the ratio between $\mathbb{E}\|\tilde{R}\tilde{R}^\mathsf{T}\|_{\mathrm{F}}^2$ and $\mathbb{E}\|\tilde{R}\|_{\mathrm{F}}^2$ converges to zero in the $d_x, d_y \to \infty$ limit.

By the independence of $\tilde{A}$ and $\tilde{B}$, the expected squared Frobenius norm expands as

$$\mathbb{E}\|\tilde{\mathrm{R}}\|_{\mathrm{F}}^2 = \mathbb{E}\operatorname{tr}\left(\tilde{A}\tilde{A}^\intercal \mathrm{R}\tilde{B}\tilde{B}^\intercal \mathrm{R}^\intercal\right) = \operatorname{tr}\left(\mathbb{E}[\tilde{A}\tilde{A}^\intercal]\mathrm{R}\,\mathbb{E}[\tilde{B}\tilde{B}^\intercal]\mathrm{R}^\intercal\right). \tag{15}$$

The matrices $\tilde{A}\tilde{A}^\intercal$ and $\tilde{B}\tilde{B}^\intercal$ are orthogonal projection matrices whose nonozero eigenvalues are equal to one. In the special case where $\Sigma_X$ and $\Sigma_Y$ are isotropic (i.e. proportional to the identity matrix), these matrices are distributed uniformly on the space of projection matrices of rank $k$. In the non-isotropic setting, however, these matrices are biased towards the directions in the covariances with large eigenvalues. An explicit expression for theirs means is provided in Lemma 8, and simplified bounds are given in Lemma 9, which shows that for all $\epsilon > 0$, there exits a number $d = d(\epsilon, \kappa, k)$ such that for all $d_x, d_y \geq d$, we can write

$$\mathbb{E}\,\tilde{A}\tilde{A}^\intercal = \frac{k}{\operatorname{tr}(\Sigma_X)}\Sigma_X(\mathrm{I}_k + \Delta_x), \qquad \mathbb{E}\,\tilde{B}\tilde{B}^\intercal = \frac{k}{\operatorname{tr}(\Sigma_Y)}\Sigma_Y(\mathrm{I}_k + \Delta_y).$$

for matrices $\Delta_x, \Delta_y$ that satisfy $\|\Delta_x\|_{\mathrm{op}}, \|\Delta_y\|_{\mathrm{op}} \leq \epsilon$. Combining these approximations with (15) and recalling that $\Sigma_X^{1/2}\mathrm{R}\Sigma_Y^{1/2} = \mathrm{C}_{XY}$, we conclude that

$$\mathbb{E}\|\tilde{\mathrm{R}}\|_{\mathrm{F}}^2 = \frac{k^2\|\mathrm{C}_{XY}\|_{\mathrm{F}}^2}{\operatorname{tr}(\Sigma_X)\operatorname{tr}(\Sigma_Y)}\left(1 + o(1)\right), \qquad d_x, d_y \to \infty.$$

Finally, we need to show that ratio between $\mathbb{E}\|\tilde{\mathrm{R}}\tilde{\mathrm{R}}^\intercal\|_F^2$ and $\mathbb{E}\|\mathrm{R}\|_{\mathrm{F}}^2$ converges to zero. We begin by considering the lower bound

$$\|\tilde{\mathrm{R}}\|_{\mathrm{F}} \geq \frac{\|A^\intercal \mathrm{C}_{XY}B\|_{\mathrm{F}}}{\|(A^\intercal\Sigma_X A)^{1/2}\|_{\mathrm{op}}\|(B^\intercal\Sigma_Y B)^{1/2}\|_{\mathrm{op}}} \geq \frac{\|A^\intercal \mathrm{C}_{XY}B\|_{\mathrm{F}}}{\|\Sigma_X\|_{\mathrm{op}}^{1/2}\|\Sigma_Y\|_{\mathrm{op}}^{1/2}}$$

as well as the upper bound

$$\|\tilde{\mathrm{R}}\tilde{\mathrm{R}}^\intercal\|_{\mathrm{F}} \leq \|(A^\intercal\Sigma_X A)^{-1}\|_{\mathrm{op}}\|(B^\intercal\Sigma_X B)^{-1}\|_{\mathrm{op}}\|A^\intercal \mathrm{C}_{XY}BB^\intercal \mathrm{C}_{XY}A\|_{\mathrm{F}}$$
$$\leq \|\Sigma_X^{-1}\|_{\mathrm{op}}\|\Sigma_Y^{-1}\|_{\mathrm{op}}\|A^\intercal \mathrm{C}_{XY}BB^\intercal \mathrm{C}_{XY}^\intercal A\|_{\mathrm{F}}.$$

Note that matrices $A$ and $B$ in these bounds are the unbiased projections, which are uniformly distributed. Since $\mathbb{E}AA^\intercal = (k/d_x)\mathrm{I}_{d_x}$ and $\mathbb{E}BB^\intercal = (k/d_y)\mathrm{I}_{d_y}$ one obtains

$$\mathbb{E}\|A^\intercal \mathrm{C}_{XY}BB^\intercal \mathrm{C}_{XY}^\intercal A\|_{\mathrm{F}} = \frac{k^2}{d_x d_y}\|\mathrm{C}_{XY}\|_{\mathrm{F}}^2$$

Meanwhile, successive applications of Lemma 7, first with respect to $AA^\intercal$ and then with respect to $BB^\intercal$, leads to

$$\mathbb{E}\|A^\intercal \mathrm{C}_{XY}BB^\intercal \mathrm{C}_{XY}^\intercal A\|_{\mathrm{F}}^2 \lesssim \frac{k^4}{d_x^2 d_y^2}\left(\|\mathrm{C}_{XY}\mathrm{C}_{XY}^\intercal\|_{\mathrm{F}}^2 + \|\mathrm{C}_{XY}\|_{\mathrm{F}}^4\right) \lesssim \frac{k^4}{d_x^2 d_y^2}\|\mathrm{C}_{XY}\|_{\mathrm{F}}^4$$

Combining these upper and lower bounds and recalling that the condition numbers of $\Sigma_X$ and $\Sigma_Y$ are no greater than $\kappa$, we have

$$\mathbb{E}\|\tilde{\mathrm{R}}\tilde{\mathrm{R}}^\intercal\|_{\mathrm{F}}^2 \lesssim \kappa^4\left(\mathbb{E}\|\tilde{\mathrm{R}}\|_{\mathrm{F}}^2\right)^2.$$

In view of the fact that $\mathbb{E}\|\tilde{\mathrm{R}}\|_{\mathrm{F}}^2$ converges to zero, the proof is complete. $\qquad\square$

## A.6 Auxiliary results for the proof of Theorem 3

**Lemma 6.** *If* $\mathrm{M}$ *is a symmetric positive semidefinite matrix with* $\|\mathrm{M}\|_{\mathrm{op}} < 1$ *then*

$$0 \leq -\log\det(\mathrm{I} - \mathrm{M}) - \operatorname{tr}(\mathrm{M}) \leq \frac{\|\mathrm{M}\|_{\mathrm{F}}^2}{2(1 - \|\mathrm{M}\|_{\mathrm{op}})}.$$

*Proof.* The log determinant is given by $-\log\det(\mathrm{I} - \mathrm{M}) - \operatorname{tr}(\mathrm{M}) = \sum_i -\log(1 - \lambda_i) - \lambda_i$ where $0 \leq \lambda_i \leq \|\mathrm{M}\|_{\mathrm{op}}$ are the eigenvalues of $\mathrm{M}$. Each summand satisfies the double inequality

$$0 \leq -\log(1 - \lambda_i) - \lambda_i = \int_0^{\lambda_i}\frac{x}{1 - x}\,dx \leq \frac{\lambda_i^2}{2(1 - \|\mathrm{M}\|_{\mathrm{op}})}.$$

Summing over both sides and noting that $\|M\|_{\mathrm{F}}^2 = \sum_i \lambda_i^2$ completes the proof. $\qquad\square$

**Lemma 7.** *Let* $P = A^\mathsf{T}A$ *where* $A$ *is distributed uniformly over* $\mathrm{St}(k, d)$. *For any* $d \times d$ *symmetric matrix* $S$, *we have*

$$\mathbb{E}\,\mathrm{tr}(PSPS) = \frac{k(kd + d - 2)}{d(d-1)(d+2)}\mathrm{tr}(S^2) + \frac{k(d-k)}{d(d-1)(d+2)}\mathrm{tr}(S)^2$$

$$\mathbb{E}\,\mathrm{tr}(PS)^2 = \frac{2k(d-k)}{d(d-1)(d+2)}\mathrm{tr}(S^2) + \frac{k(kd + k - 2)}{d(d-1)(d+2)}\mathrm{tr}(S)^2.$$

*Proof.* Because the distribution of $P$ is invariant to orthogonal transformation of its rows and columns (i.e., $P$ is equal in distribution to $UPU^\mathsf{T}$ for any $U \in \mathbb{O}(d)$), the quantities of interest are unchanged if $S$ is replaced by a diagonal matrix containing its eigenvalues $\lambda_1, \ldots, \lambda_n$. In particular, we have

$$\mathrm{tr}(PSPS) \stackrel{d}{=} \mathrm{tr}(P\mathrm{diag}(\lambda)P\mathrm{diag}(\lambda)) = \sum_{i,j=1}^{d} \lambda_i\lambda_j P_{ij}^2$$

$$\mathrm{tr}(PS)^2 \stackrel{d}{=} \mathrm{tr}(P\mathrm{diag}(\lambda))^2 = \sum_{i,j=1}^{d} \lambda_i\lambda_j P_{ii}P_{jj}.$$

A further consequence of the orthogonal invariance of $P$ is that its second order moments satisfy $\mathbb{E}[P_{ii}^2] = \mathbb{E}[P_{11}^2]$, $\mathbb{E}[P_{ij}^2] = \mathbb{E}[P_{12}^2]$ and $\mathbb{E}[P_{ii}P_{jj}] = \mathbb{E}[P_{11}P_{22}]$ for all $1 \le i \ne j \le d$, and so the expectations can be simplified as follows:

$$\mathbb{E}\,\mathrm{tr}(PSPS) = \mathbb{E}[P_{11}^2]\sum_i \lambda_i^2 + \mathbb{E}[P_{12}^2]\sum_{i\ne j}\lambda_i\lambda_j$$

$$= \left(\mathbb{E}[P_{11}^2] - \mathbb{E}[P_{12}^2]\right)\mathrm{tr}(S^2) + \mathbb{E}[P_{12}^2]\,\mathrm{tr}(S)^2 \tag{16}$$

$$\mathbb{E}\,\mathrm{tr}(PS)^2 = \mathbb{E}[P_{11}^2]\sum_i \lambda_i^2 + \mathbb{E}[P_{11}P_{22}]\sum_{i\ne j}\lambda_i\lambda_j$$

$$= \left(\mathbb{E}[P_{11}^2] - \mathbb{E}[P_{11}P_{22}]\right)\mathrm{tr}(S^2) + \mathbb{E}[P_{11}P_{22}]\,\mathrm{tr}(S)^2 \tag{17}$$

Finally, we can determine coefficients in these expressions by evaluating (16) and (17) for special choices of $S$. Recall that $P$ has $k$ nonzero eigenvalues all of which are equal to one. Therefore, if $S = I$, then $\mathrm{tr}(SP) = k$ and $\mathrm{tr}(SPSP) = k^2$ a.s., and in view of (16) and (17), we obtain

$$k = d\mathbb{E}[P_{11}^2], + d(d-1)\mathbb{E}[P_{12}^2] \qquad k^2 = d\mathbb{E}[P_{11}^2] + d(d-1)\mathbb{E}[P_{11}P_{22}].$$

Alternatively, if $S = e_1 e_2^\mathsf{T} + e_2 e_1^\mathsf{T}$ then $\mathbb{E}\,\mathrm{tr}(SP)^2 = \mathbb{E}[(P_{12} + P_{21})^2] = 4\mathbb{E}[P_{12}^2]$ and so (17) implies that

$$2\mathbb{E}[P_{12}] = \mathbb{E}[P_{11}^2] - \mathbb{E}[P_{11}P_{22}].$$

Solving these linear equations yields

$$\mathbb{E}[P_{11}^2] = \frac{k(k+2)}{d(d+2)}, \qquad \mathbb{E}[P_{12}^2] = \frac{k-d}{d(d-1)(d+2)}, \qquad \mathbb{E}[P_{11}P_{22}] = \frac{k(kd+k-2)}{d(d-1)(d+2)}.$$

Combining these expressions with (16) and (17) gives the desired result. $\qquad\square$

**Lemma 8.** *Let* $P = \Sigma^{1/2}A(A^\mathsf{T}\Sigma A)^{-1}A^\mathsf{T}\Sigma^{1/2}$ *where* $\Sigma$ *is an deterministic* $d \times d$ *positive definite matrix with spectral decomposition* $\Sigma = \sum_i \lambda_i u_i u_i^\mathsf{T}$ *and* $A$ *is distributed uniformly on* $\mathrm{St}(k, d)$. *Then, the mean of* $P$ *is given by* $\mathbb{E}P = \sum_i \eta_i u_i u_i^\mathsf{T}$ *where*

$$\eta_i = \mathbb{E}\left[\frac{\lambda_i Z_i^\mathsf{T} W_i^{-1} Z_i}{1 + \lambda_i Z_i^\mathsf{T} W_i^{-1} Z_i}\right]$$

*with* $Z_1, \ldots, Z_d$ *independent* $\mathcal{N}(0, I_k)$ *variables and* $W_i = \sum_{j\ne i} \lambda_j Z_j Z_j^\mathsf{T}$.

*Proof.* It is straightforward to show (see e.g., [44, Theorem 3.2]) that the distribution of the $n \times k$ matrix $\Sigma^{1/2}A(A^\mathsf{T}\Sigma A)^{-1/2}$ is unchanged if the random matrix $A$ is replaced by Gaussian matrix $Z = [Z_1, \ldots, Z_n]^\mathsf{T}$ whose rows are independent $\mathcal{N}(0, I_k)$ variables. Thus, letting $U = [u_1, \ldots, u_n]$ and $\Lambda = \mathrm{diag}(\lambda_1, \ldots, \lambda_n)$ be the be the eigenvectors and eigenvalues of $\Sigma$ we have

$$U^\mathsf{T}PU \stackrel{d}{=} \Lambda^{1/2}Z(Z^\mathsf{T}\Lambda Z)^{-1}Z^\mathsf{T}\Lambda^{1/2}.$$

In view of the above decomposition, we see that the $ij$-th entry of $\mathrm{U}^\mathsf{T}\mathrm{P}\mathrm{U}$ is equal in distribution to $\lambda_i^{1/2}\lambda_j^{1/2}\mathrm{Z}_i^\mathsf{T}(\lambda_i\mathrm{Z}_i\mathrm{Z}_i^\mathsf{T}+\mathrm{W}_i)^{-1}\mathrm{Z}_j$. For the off-diagonal entries, note that the distribution of $(\mathrm{Z}_1,\ldots,\mathrm{Z}_n)$ is equal to the distribution of $(\mathrm{Z}_1,\ldots,\mathrm{Z}_{i-1},S\mathrm{Z}_i,\mathrm{Z}_{i+1},\ldots,\mathrm{Z}_n)$ where $S$ is an independent random variable distributed uniformly on $\{-1,1\}$. Making this substitution and then taking the expectation with respect to $S$ we see that the off-diagonal entries have mean zero. The expression for the diagonal follows from applying the matrix inversion lemma to $\lambda_i\mathrm{Z}_i^\mathsf{T}(\lambda_i\mathrm{Z}_i\mathrm{Z}_i^\mathsf{T}+\mathrm{W}_i)^{-1}\mathrm{Z}_i$. $\qquad\square$

**Lemma 9.** *Consider the setting of Lemma 8. There exists an absolute positive constant $C$ such that if*

$$\frac{(2+k)\|\Sigma\|_{\mathrm{op}}}{\mathrm{tr}(\Sigma)-\|\Sigma\|_{\mathrm{op}}}\le\epsilon,\qquad C\frac{\|\Sigma\|_{\mathrm{op}}}{\mathrm{tr}(\Sigma)}\left(k+\sqrt{kd}+\log\left(\frac{2\mathrm{tr}(\Sigma)\|\Sigma^{-1}\|_{\mathrm{op}}}{k\epsilon}\right)\right)\le\frac{\epsilon}{2+\epsilon}$$

*for some $\epsilon>0$, then*

$$\left|\eta_i\cdot\frac{\mathrm{tr}(\Sigma)}{k\lambda_i}-1\right|\le\epsilon$$

*for all $1\le i\le d$.*

*Proof.* We begin with a lower bound on $\eta_i$. For any nonzero vector $v\in\mathbb{R}^k$, the mapping $\mathrm{M}\mapsto(v^\mathsf{T}\mathrm{M}^{-1}v)/(1+v^\mathsf{T}\mathrm{M}^{-1}v)$ is convex over the cone of $k\times k$ positive semidefinite matrices. By Jensen's inequality, the independence of $\mathrm{W}_i$ and $\mathrm{Z}_i$, and the fact that $\mathbb{E}[\mathrm{W}_i]=\tau_i\mathrm{I}_k$ where $\tau_i:=\sum_{j\ne i}\lambda_i=\mathrm{tr}(\Sigma)-\lambda_i$, we have

$$\eta_i\ge\mathbb{E}\left[\frac{\lambda_i\mathrm{Z}_i(\mathbb{E}[\mathrm{W}_i])^{-1}\mathrm{Z}_i}{1+\lambda_i\mathrm{Z}_i(\mathbb{E}[\mathrm{W}_i])^{-1}\mathrm{Z}_i}\right]=\mathbb{E}\left[\frac{\lambda_i\|\mathrm{Z}_i\|^2}{\tau_j+\lambda_i\|\mathrm{Z}_i\|^2}\right].$$

To remove remove the expectation with respect to $\|Z_i\|^2$, we bound the RHS from below using

$$\mathbb{E}\left[\frac{\lambda_i\|\mathrm{Z}_i\|^2}{\tau_j+\lambda_i\|\mathrm{Z}_i\|^2}\right]=\frac{k\lambda_i}{\mathrm{tr}(\Sigma)}-\frac{\lambda_i^2}{\mathrm{tr}(\Sigma)}\mathbb{E}\left[\frac{\|\mathrm{Z}_i\|^2(\|\mathrm{Z}_i\|^2-1)}{(\tau_j+\lambda_i\|\mathrm{Z}_i\|^2)}\right]\ge\frac{k\lambda_i}{\mathrm{tr}(\Sigma)}-\frac{k(2+k)\lambda_i^2}{\mathrm{tr}(\Sigma)(\mathrm{tr}(\Sigma)-\|\Sigma\|_{\mathrm{op}})},$$

where the second step follows from $\mathbb{E}\|\mathrm{Z}_i\|^2=k$ and $\mathbb{E}\|\mathrm{Z}_i\|^4=k(k+2)$.

Next we consider an upper bound. If we let $L_i:=\min\{u^\mathsf{T}Wu:u\in\mathbb{S}^{k-2}\}$ be the minimum eigenvalue of $k\times k$ symmetric matrix $W_i$ and then we can write

$$\eta_i\le\mathbb{E}\left[\frac{\lambda_iL_i^{-1}\|\mathrm{Z}_i\|^2}{1+\lambda_iL_i^{-1}\|\mathrm{Z}_i\|^2}\right]\le\mathbb{E}\left[\frac{k\lambda_i}{L_i+k\lambda_i}\right]$$

where the second step follows from the Jensen's inequality and the independence of $Z_i$ and $L_i$. By concentration of Lipschitz functions of Gaussian measure, one finds that that $L_i^{1/2}$ is sub-Gaussian with variance proxy $\max_{j\ne i}\lambda_j\le\|\Sigma\|_{\mathrm{op}}$ and this implies a sub-exponential tail bound for $L_i$ of the form

$$\mathbb{P}\Big(L_i\le\mathbb{E}[L_i]-C'\|\Sigma\|_{\mathrm{op}}t\Big)\le2e^{-t}.$$

for some absolute constant $C'>0$. To obtain a lower bound on the expectation of $L_i$, recall that $\mathbb{E}[W_i]=\tau_i\mathrm{I}_k$ where $\tau_i=\sum_{j\ne i}\lambda_j$. Noting that

$$\tau_i-L_i\le|L_i-\tau_i|\le-\|W_i-\mathbb{E}[W_i]\|_{\mathrm{op}},$$

and then taking the expectation of both sides leads to $\mathbb{E}[L_i]\ge\tau_i-\mathbb{E}\|W_i-\mathbb{E}[W_i]\|_{\mathrm{op}}$. At this point, we can apply Theorem 3.13 in [45], which gives

$$\mathbb{E}\big\|W_i-\mathbb{E}[W_i]\big\|_{\mathrm{op}}\lesssim\sqrt{k\sum_{j\ne i}\lambda_j^2}+k\max_{j\ne i}\lambda_j\le\|\Sigma\|_{\mathrm{op}}(k+1+\sqrt{dk}),$$

Combining these bounds and recalling that $\tau_i=\mathrm{tr}(\Sigma)-\lambda_i$ yields

$$\mathbb{E}L_i\ge\mathrm{tr}(\Sigma)-C''\|\Sigma\|_{\mathrm{op}}(k+\sqrt{dk}),$$

for some absolute constant $C'' > 0$. Putting the pieces together, we have for all $t > 0$,

$$\eta_i \leq \mathbb{E}\left[\frac{k\lambda_i}{L_i + k\lambda_i}\mathbb{1}_{\{L_i \geq \mathbb{E}[L_i] + C'\|\Sigma\|_{\mathrm{op}}t\}}\right] + \mathbb{E}\left[\frac{k\lambda_i}{L_i + k\lambda_i}\mathbb{1}_{\{L_i < \mathbb{E}[L_i] + C'\|\Sigma\|_{\mathrm{op}}t\}}\right]$$

$$\leq \frac{k\lambda_i}{\mathbb{E}[L_i] - C'\|\Sigma\|_{\mathrm{op}}t + k\lambda_i} + 2e^{-t}$$

$$\leq \frac{k\lambda_i}{\mathrm{tr}(\Sigma) - C''\|\Sigma\|_{\mathrm{op}}(k + \sqrt{dk}) - C'\|\Sigma\|_{\mathrm{op}}t + k\lambda_i} + 2e^{-t}.$$

where the last two lines hold provided that the denominator is strictly positive. Hence, if $t = \log(2\mathrm{tr}(\Sigma)\|\Sigma^{-1}\|_{\mathrm{op}}/(k\epsilon))$ and

$$\mathrm{tr}(\Sigma) - C''\|\Sigma\|_{\mathrm{op}}(k + \sqrt{dk}) - C'\|\Sigma\|_{\mathrm{op}}t \geq \frac{\mathrm{tr}(\Sigma)}{1 + \epsilon/2},$$

then

$$\eta_i \leq (1 + \epsilon/2)\frac{k\lambda_i}{\mathrm{tr}(\Sigma)} + \frac{\epsilon k}{2\mathrm{tr}(\Sigma)\|\Sigma^{-1}\|_{\mathrm{op}}} \leq \frac{k\lambda_i}{\mathrm{tr}(\Sigma)}(1 + \epsilon)$$

Simplifying the conditions leads to the stated bound. □

### A.7 Proof of Decomposition in Equation (4)

Fix $\theta := (\theta_1, \theta_2) = (\mathrm{A}, \mathrm{B}) \in \mathrm{St}(k, d_x) \times \mathrm{St}(k, d_y)$ and let $f_\theta$, $f_{\theta_1}$, and $f_{\theta_2}$ denote, respectively, the densities of $(\mathrm{A}^\mathsf{T}X, \mathrm{B}^\mathsf{T}Y)$, $\mathrm{A}^\mathsf{T}Z$, and $\mathrm{B}^\mathsf{T}Y$, where $(X, Y) \sim \mu_{XY}$. Similarly, we use $\varphi_\theta$, $\varphi_{\theta_1}$, and $\varphi_{\theta_2}$, for the densities when $(X, Y)$ are replaced with their Gaussian approximation $(X^*, Y^*) \sim \gamma_{XY}$. We may now decompose

$$\mathsf{I}(\mathrm{A}^\mathsf{T}X; \mathrm{B}^\mathsf{T}Y) = \int f_\theta(s, t) \log\left(\frac{\varphi_\theta(s, t)}{\varphi_{\theta_1}(s)\varphi_{\theta_2}(t)}\frac{f_\theta(s, t)}{\varphi_\theta(s, t)}\frac{\varphi_{\theta_1}(s)\varphi_{\theta_2}(t)}{f_{\theta_1}(s)f_{\theta_2}(t)}\right) ds\, dt$$

$$= \mathbb{E}_{\mu_{XY}}\left[\log\left(\frac{\varphi_\theta}{\varphi_{\theta_1}\varphi_{\theta_2}}\right)\right] + \mathsf{D}\big((\mathfrak{p}^\mathrm{A}, \mathfrak{p}^\mathrm{B})_\sharp\mu_{XY}\big\|(\mathfrak{p}^\mathrm{A}, \mathfrak{p}^\mathrm{B})_\sharp\gamma_{XY}\big)$$

$$- \mathsf{D}\big((\mathfrak{p}^\mathrm{A}, \mathfrak{p}^\mathrm{B})_\sharp\mu_X \otimes \mu_Y\big\|(\mathfrak{p}^\mathrm{A}, \mathfrak{p}^\mathrm{B})_\sharp\gamma_X \otimes \gamma_Y\big).$$

Observing that $\log\left(\frac{\varphi_\theta}{\varphi_{\theta_1}\varphi_{\theta_2}}\right)$ depends only on the 2nd moment on the random variables and since the Gaussian approximation $(X^*, Y^*) \sim \gamma_{XY}$ was chosen to have the same covariance matrix as $(X, Y) \sim \mu_{XY}$, we may replace the distribution $\mu_{XY}$ w.r.t. which the expectation is taken with $\gamma_{XY}$. Doing so and taking an average over $(\mathrm{A}, \mathrm{B}) \in \mathrm{St}(k, d_x) \times \mathrm{St}k, d_y)$, we obtain

$$\mathsf{SI}_k(X; Y) = \mathsf{SI}_k(X^*; Y^*) + \mathbb{E}\big[\delta(A, B)\big]$$

where $\delta(A, B)$ is as defined under Equation (4) in the main text. □

## B Bounds on Residual Term from Equation (4)

Throughout this appendix we interchangeably denote information measures in terms of probability distribution or the corresponding random variables. For instance, we write $\mathsf{J}(X)$ or $\mathsf{J}(\mu)$ for the Fisher information of $X \sim \mu$, and $\mathsf{W}_2(X, Y)$ or $\mathsf{W}_2(\mu, \nu)$ for the 2-Wasserstein distance between $X \sim \mu$ and $Y \sim \nu$. We also define $\alpha(X) := \frac{1}{d_x}\mathbb{E}\big[\big|\|X\|^2 - \mathbb{E}\|X\|^2\big|\big]$ and $\beta_r(X) := \frac{1}{d_x}\big(\mathbb{E}\big[\big|\langle X_1, X_2\rangle\big|^r\big]\big)^{1/r}$, for $r = 1, 2$, where $X_1$ and $X_2$ are independent copies of $X \sim \mu_X$. The quantities $\alpha(Y)$ and $\beta_r(Y)$ are defined analogously. Note that $\beta_1(X) \leq \beta_2(X) = \frac{1}{d_x}\|\Sigma_X\|_\mathrm{F}$.

Due to translation invariance of $k$-SMI we may assume that $X$ and $Y$ are centered. Define the shorthand notation $\Theta = \mathrm{A} \oplus \mathrm{B}$ and $Z = (X^\mathsf{T}\, Y^\mathsf{T})^\mathsf{T}$. Accordingly, $\mu_Z = \mu_{XY}$ and we set $\gamma_Z = \gamma_{XY} = \mathcal{N}(0, \Sigma_{XY})$ for the corresponding Gaussian; the Gaussian vector with distribution $\gamma_Z$ is denoted by $Z_* = (X_*^\mathsf{T}\, Y_*^\mathsf{T})^\mathsf{T}$. Slightly abusing notation we define $\mathfrak{p}^\Theta(z) = \Theta^\mathsf{T}z = (x^\mathsf{T}\mathrm{A}\, y^\mathsf{T}\mathrm{B})^\mathsf{T}$.

To control the residual from (4), we first bound it in terms of a certain MI term. Let $\mathrm{A}_*$ and $\mathrm{B}_*$ be matrices of dimension $d_x \times k$ and $d_y \times k$ with entries i.i.d. according to $\mathcal{N}(0, 1/d_x)$ and $\mathcal{N}(0, 1/d_y)$,

respectively. Define $\Theta_* = A_* \oplus B_*$ and let $W = \Theta_*^{\mathsf{T}} Z + \sqrt{t} N$, where $N \sim \mathcal{N}(0, I_{2k})$. The following bound controls the residual in terms of $I(\Theta_*; W)$, plus a term that vanishes when $t$ is small and $d_x, d_y$ are large. The proof is deferred to Appendix B.1.

**Lemma 10** (Residual bound via noisy MI). *Under the above model with $d_x \wedge d_y > k+1$ and for any $t > 0$, we have*

$$\mathbb{E}\big[D(\mathfrak{p}_\sharp^\Theta \mu_Z \| \mathfrak{p}^\Theta \gamma_Z)\big] \leq I(\Theta_*; W)$$
$$+ k\sqrt{2\|J_F(\mu_Z)\|_{\mathrm{op}}} \left( \sqrt{t \left( \frac{d_x}{d_x - k + 1} + \frac{d_y}{d_y - k + 1} \right)} + \sqrt{\alpha(X) + \alpha(Y) + \frac{2 d_x \beta_2^2(X)}{\mathrm{tr}(\Sigma_X)} + \frac{2 d_y \beta_2^2(Y)}{\mathrm{tr}(\Sigma_Y)}} \right).$$

Next, we bound the noisy MI term $I(\Theta_*; W)$. Let $\lambda_x = \frac{1}{d_x} \mathbb{E}\|X\|^2$ and $\lambda_y = \frac{1}{d_y} \mathbb{E}\|Y\|^2$, and for simplicity of presentation, henceforth assume that $\lambda = \lambda_x = \lambda_y$. This is without loss of generality since $k$-SMI is scale invariant.[6] Note that if $\lambda \in (0, \infty)$, then we have $0 \leq \alpha(X) \leq 2\lambda$ and $\lambda/\sqrt{d_x} \leq \beta_2(X) \leq \lambda$ (cf. [32]). Lastly, set $\bar{\alpha} = \max\{\alpha(X), \alpha(Y)\}$ and $\bar{\beta}_r = \max\{\beta_r(X), \beta_r(Y)\}$, for $r = 1, 2$. We prove the following result in Appendix B.2.

**Lemma 11** (Noisy MI bound). *For any $t > 0$ and $\epsilon \in (0, 1]$, we have*

$$I(\Theta_*; W) \leq Ck \log\left(1 + \frac{\lambda}{t}\right) \frac{\bar{\alpha}}{\epsilon \lambda} + C \left(\frac{1 + \epsilon}{1 - \epsilon}\right)^{\frac{k}{4}} \left( k^{\frac{3}{4}} \sqrt{\frac{\bar{\beta}_1}{\lambda}} + k^{\frac{1}{4}} \left(1 + \frac{2(1 + \epsilon)\lambda}{t}\right)^{\frac{k}{2}} \frac{\bar{\beta}_2}{\lambda} \right).$$

*where $C$ is an absolute constant (in particular, $C = 3$ is sufficient).*

Combining Lemmas 10 and 11, yields a bound on $\mathbb{E}\big[D(\mathfrak{p}_\sharp^\Theta \mu_Z \| \mathfrak{p}^\Theta \gamma_Z)\big]$ in terms $k$, $d_x$, $d_y$, $\lambda$, $\bar{\alpha}, \bar{\beta}$ and (arbitrary) $t > 0$ and $\epsilon \in (0, 1]$. To further simplify the subsequent expressions, suppose that $(\bar{\beta}_2/\lambda)^{2/(k+1)} \leq \frac{1}{2}$, and set[7]

$$t^* = 2(1 + \epsilon)\lambda \left( \left(\frac{\bar{\beta}_2}{\lambda}\right)^{-\frac{2}{k+1}} - 1 \right)^{-1} \leq 4(1 + \epsilon)\lambda \left(\frac{\bar{\beta}_2}{\lambda}\right)^{\frac{2}{k+1}}.$$

Inserting into the said bound, we obtain

$$\mathbb{E}\big[D(\mathfrak{p}_\sharp^\Theta \mu_Z \| \mathfrak{p}^\Theta \gamma_Z)\big]$$
$$\leq Ck \log\left(1 + \frac{1}{2(1 + \epsilon)} \left( \left(\frac{\bar{\beta}_2}{\lambda}\right)^{-\frac{2}{k+1}} - 1 \right) \right) \frac{\bar{\alpha}}{\epsilon \lambda}$$
$$+ C \left(\frac{1 + \epsilon}{1 - \epsilon}\right)^{\frac{k}{4}} \left( k^{\frac{3}{4}} \sqrt{\frac{\bar{\beta}_1}{\lambda}} + k^{\frac{1}{4}} \left(\frac{\bar{\beta}_2}{\lambda}\right)^{\frac{1}{k+1}} \right)$$
$$+ k\sqrt{2\|J_F(\mu_Z)\|_{\mathrm{op}}} \left( \left(\frac{\bar{\beta}_2}{\lambda}\right)^{\frac{1}{k+1}} \sqrt{4(1 + \epsilon)\lambda \left( \frac{d_x}{d_x - k + 1} + \frac{d_y}{d_y - k + 1} \right)} \right.$$
$$\left. + \sqrt{2\bar{\alpha} + \frac{2 d_x \beta_2^2(X)}{\mathrm{tr}(\Sigma_X)} + \frac{2 d_x \beta_2^2(Y)}{\mathrm{tr}(\Sigma_Y)}} \right).$$

We can now complete the bound on the residual term $\mathbb{E}[\delta_{XY}(A, B)]$ from (4). Recall the definition of $\Theta = A \oplus B$ and $Z = (X^{\mathsf{T}} Y^{\mathsf{T}})^{\mathsf{T}}$, we have

$$\mathbb{E}\big[\delta_{XY}(A, B)\big] \leq \mathbb{E}\left[D\big((\mathfrak{p}^A, \mathfrak{p}^B)_\sharp \mu_{XY} \| (\mathfrak{p}^A, \mathfrak{p}^B)_\sharp \gamma_{XY}\big)\right]$$

---

[6]This scaling does affect the $\alpha, \beta$ factors in the lemma but will not change their convergence properties so long as $\lambda_x$ and $\lambda_y$ scale at the same rate.

[7]Our bounds only need $t^*$ to be strictly positive, which is always the case under the considered setting. Indeed, by the the Cauchy-Schwartz inequality $\bar{\beta}_2 \leq \lambda$, with equality having probability zero since two independent copies of a continuous random variables are a.s. not linearly aligned.

$$\leq Ck \log \left( 1 + \frac{1}{2(1+\epsilon)} \left( \left( \frac{\bar{\beta}_2}{\lambda} \right)^{-\frac{2}{k+1}} - 1 \right) \right) \frac{\bar{\alpha}}{\epsilon \lambda}$$

$$+ C \left( \frac{1+\epsilon}{1-\epsilon} \right)^{\frac{k}{4}} \left( k^{\frac{3}{4}} \sqrt{\frac{\bar{\beta}_1}{\lambda}} + k^{\frac{1}{4}} \left( \frac{\bar{\beta}_2}{\lambda} \right)^{\frac{1}{k+1}} \right)$$

$$+ k \sqrt{2 \| J_F(\mu_Z) \|_{op}} \left( \left( \frac{\bar{\beta}_2}{\lambda} \right)^{\frac{1}{k+1}} \sqrt{4(1+\epsilon)\lambda \left( \frac{d_x}{d_x - k + 1} + \frac{d_y}{d_y - k + 1} \right)} \right.$$

$$\left. + \sqrt{2\bar{\alpha} + \frac{2 d_x \beta_2^2(X)}{\text{tr}(\Sigma_X)} + \frac{2 d_x \beta_2^2(Y)}{\text{tr}(\Sigma_Y)}} \right).$$

Observe that this will typically converge to zero with increasing $d_x$, $d_y$. To better instantiate this regime, we revisit the concept of weak dependence, i.e. random vectors with weakly dependent entries [32] (essentially, a notion of approximate isotropy). The following proposition, whose proof is straightforward and hence omitted, provides explicit convergence rate for the residual subject to the weak dependence assumption.

**Proposition 4** (Convergence rate under weak dependence). *Suppose that $\lambda$, $k$, $\| J_F(\mu_Z) \|_{op}$, $\frac{d_x}{\text{tr}(\Sigma_X)}$, $\frac{d_y}{\text{tr}(\Sigma_Y)}$ are $O(1)$ with respect to $d_x$ and $d_y$, and that there exists an absolute $C < 0$ such that*

$$\frac{\alpha(X)}{\lambda} \leq \frac{C}{\sqrt{d_x}} \quad , \quad \frac{\alpha(Y)}{\lambda} \leq \frac{C}{\sqrt{d_y}} \quad , \quad \frac{\beta_2(X)}{\lambda} \leq \frac{C}{\sqrt{d_x}} \quad , \quad \frac{\beta_2(Y)}{\lambda} \leq \frac{C}{\sqrt{d_y}}.$$

*Then, up to log factors,*

$$\mathbb{E}[\delta_{XY}(A, B)] \lesssim \frac{k}{\epsilon} \left( d_x^{-\frac{1}{2}} + d_y^{-\frac{1}{2}} \right) + \left( \frac{1+\epsilon}{1-\epsilon} \right)^{\frac{k}{4}} \left( k^{\frac{3}{4}} \left( d_x^{-\frac{1}{4}} + d_y^{-\frac{1}{4}} \right) + k^{\frac{1}{4}} \left( d_x^{-\frac{1}{2(k+1)}} + d_y^{-\frac{1}{2(k+1)}} \right) \right)$$

$$+ k \left( d_x^{-\frac{1}{2(k+1)}} + d_y^{-\frac{1}{2(k+1)}} \right) + k \left( d_x^{-\frac{1}{4}} + d_y^{-\frac{1}{4}} \right),$$

*which, for $d_x = d_y = d$ increasing, decays to zero as $\tilde{O}\left( d^{-\frac{1}{4}} + d^{-\frac{1}{2(k+1)}} \right)$.*

## B.1 Proof of Lemma 10

We can represent the residual term from (4), as

$$\mathbb{E}\left[ D(\mathfrak{p}_\sharp^\Theta \mu_Z \| \mathfrak{p}^\Theta \gamma_Z) \right] = I(\Theta; \Theta^\mathsf{T} Z) + h(\Theta^\mathsf{T} Z^* | \Theta) - h(\Theta^\mathsf{T} Z)$$
$$\leq I(\Theta; \Theta^\mathsf{T} Z) + h(\Theta^\mathsf{T} Z^*) - h(\Theta^\mathsf{T} Z) \tag{18}$$

For the latter entropy difference we use the Wasserstein continuity result from Lemma 1, to obtain

$$h(\Theta^\mathsf{T} Z^*) - h(\Theta^\mathsf{T} Z) \leq \sqrt{J(\Theta^\mathsf{T} Z)} W_2(\Theta^\mathsf{T} Z, \Theta^\mathsf{T} Z). \tag{19}$$

For the Fisher information term we use the data processing inequality [41, Proposition 5][8] and the fact that $\Theta$ is an orthogonal matrix (i.e., $\Theta^\mathsf{T} \Theta = I_{2k}$) to obtain

$$J(\Theta^\mathsf{T} Z) \leq \int J(\theta^\mathsf{T} Z) d(\sigma_{k,d_x} \otimes \sigma_{k,d_y})(\theta)$$

$$\leq \int \text{tr}\left( \theta J_F(Z) \theta^\mathsf{T} \right) d(\sigma_{k,d_x} \otimes \sigma_{k,d_y})(\theta)$$

---

[8]The Fisher information $J(W)$ is related to the parametric Fisher information $J_\vartheta(W) := \text{Var}\left( \frac{\partial}{\partial \vartheta} \log p_\vartheta(W) \right)$ as follows: if $\vartheta \in \mathbb{R}^d$ is a location parameter, i.e., $p_\vartheta(w) = p(w + \vartheta)$, then $J(W) = J_\vartheta(W - \vartheta)$. Proposition 5 of [41] states that if $\vartheta \leftrightarrow W \leftrightarrow \widetilde{W}$ form a Markov chain, then $J_\vartheta(\widetilde{W}) \leq J_\vartheta(W)$. Take $W = (\vartheta + \Theta^\mathsf{T} Z, \Theta)$ and $\widetilde{W} = \vartheta + \Theta^\mathsf{T} Z$, which clearly satisfy the said Markov chain, and invoke that result to obtain $J(\Theta^\mathsf{T} Z) = J_\vartheta(\widetilde{W}) \leq J_\vartheta(W) = \int J(\theta^\mathsf{T} Z) d(\sigma_{k,d_x} \otimes \sigma_{k,d_y})(\theta)$. The latter equality is since $\frac{\partial}{\partial \vartheta} \log p_{\vartheta + \Theta^\mathsf{T} Z, \Theta}(\cdot, \cdot) = \frac{\partial}{\partial \vartheta} \log p_{\vartheta + \Theta^\mathsf{T} Z | \Theta}(\cdot | \cdot).$

$$\leq 2k\|\mathsf{J}_\mathrm{F}(Z)\|_\mathrm{op}. \tag{20}$$

To treat the 2-Wasserstein distance, note that by orthogonal invariance of the projections, we see that the (unconditional) distribution of $\Theta^\intercal Z$ satisfies

$$\Theta^\intercal Z = \begin{pmatrix} \mathrm{A}^\intercal X \\ \mathrm{B}^\intercal Y \end{pmatrix} \stackrel{d}{=} \begin{pmatrix} \mathrm{A}_1\|X\| \\ \mathrm{B}_1\|Y\| \end{pmatrix}$$

where $\mathrm{A}_1$ and $\mathrm{B}_1$ are the first rows of $\mathrm{A}$ and $\mathrm{B}$, respectively. The same decomposition holds for $\Theta^\intercal Z^*$. Hence,

$$\mathsf{W}_2(\Theta^\intercal Z, \Theta^\intercal Z^*) = \mathsf{W}_2\left( \begin{pmatrix} \mathrm{A}_1\|X\| \\ \mathrm{B}_1\|Y\| \end{pmatrix}, \begin{pmatrix} \mathrm{A}_1\|X^*\| \\ \mathrm{B}_1\|Y^*\| \end{pmatrix} \right) \leq \sqrt{k}\,\mathsf{W}_2\left( \begin{pmatrix} \|X\|/\sqrt{d_x} \\ \|Y\|/\sqrt{d_y} \end{pmatrix}, \begin{pmatrix} \|X^*\|/\sqrt{d_x} \\ \|Y^*\|/\sqrt{d_y} \end{pmatrix} \right),$$

where the inequality follows from restricting to a coupling with the same $(\mathrm{A}_1, \mathrm{B}_1)$ and recalling that the entries of $\mathrm{A}_1$ and $\mathrm{B}_1$ have second moments of $1/d_x$ and $1/d_y$, respectively.

For any coupling of $(X, Y)$ and $(X^*, Y^*)$, we have

$$\mathbb{E}\left\| \begin{pmatrix} \|X\|/\sqrt{d_x} \\ \|Y\|/\sqrt{d_y} \end{pmatrix} - \begin{pmatrix} \|X^*\|/\sqrt{d_x} \\ \|Y^*\|/\sqrt{d_y} \end{pmatrix} \right\|^2$$

$$\leq \frac{2}{d_x}\mathbb{E}\left[ \left| \|X\| - \sqrt{\mathrm{tr}(\Sigma_X)} \right|^2 \right] + \frac{2}{d_x}\mathbb{E}\left[ \left| \|X^*\| - \sqrt{\mathrm{tr}(\Sigma_X)} \right|^2 \right]$$

$$+ \frac{2}{d_x}\mathbb{E}\left[ \left| \|Y\| - \sqrt{\mathrm{tr}(\Sigma_Y)} \right|^2 \right] + \frac{2}{d_y}\mathbb{E}\left[ \left| \|Y^*\| - \sqrt{\mathrm{tr}(\Sigma_Y)} \right|^2 \right]$$

where we have used the inequality $(a+b)^2 \leq 2a^2 + 2b^2$. Note that for any random positive random variable $W$ we have

$$\mathbb{E}\left[ \left| W - \sqrt{\mathbb{E}W^2} \right|^2 \right] \leq \mathbb{E}\left[ \left| W - \sqrt{\mathbb{E}W^2} \right|^2 \left( 1 + \frac{W}{\sqrt{\mathbb{E}W^2}} \right)^2 \right] = \frac{\mathrm{Var}(W^2)}{\mathbb{E}W^2}$$

Since $(X^*, Y^*)$ are Gaussian, their squared Euclidean norms can be expressed as the weighted sum of independent chi-squared variables, and one finds that $\mathbb{E}[\|X^*\|^2] = \mathrm{tr}(\Sigma_X)$ and $\mathrm{Var}(\|X^*\|^2) = 2\|\Sigma_X\|_\mathrm{F}^2$, and similarly for $Y^*$. Putting everything together, we obtain

$$\mathsf{W}_2^2(\Theta^\intercal Z, \Theta^\intercal Z^*) \leq 2k\left( \alpha(X) + \alpha(Y) + \frac{2d_x\beta_2^2(X)}{\mathrm{tr}(\Sigma_X)} + \frac{2d_y\beta_2^2(X)}{\mathrm{tr}(\Sigma_Y)} \right) \tag{21}$$

where $\alpha(X)$ and $\alpha(Y)$ are defined in Lemma 10.

It remains to transform the MI term $\mathsf{I}(\Theta; \Theta^\intercal Z)$ in (18) into $\mathsf{I}(\Theta_*; \Theta_*^\intercal Z + \sqrt{t}N)$, where $N \sim \mathcal{N}(0, \mathrm{I}_{2k})$ and $\Theta_* = \mathrm{A}_* \oplus \mathrm{B}_*$ with $\mathrm{A}_*$ and $\mathrm{B}_*$ matrices of dimension $d_x \times k$ and $d_y \times k$ and entries i.i.d. according to $\mathcal{N}(0, 1/d_x)$ and $\mathcal{N}(0, 1/d_y)$, respectively. Using the polar decomposition of Gaussian matrices, we know that $\mathrm{A}_* \stackrel{d}{=} \mathrm{A}(\mathrm{A}_*^\intercal\mathrm{A}_*)^{1/2}$, where $\mathrm{A} \sim \sigma_{k,d_x}$, i.e., it is uniformly distributed over $\mathrm{St}(k, d_x)$. A similar claim holds for $\mathrm{B}_*$. By invariance of MI to invertible transformations ($\mathrm{A}_*^\intercal\mathrm{A}_*$ and $\mathrm{B}_*^\intercal\mathrm{B}_*$ are a.s. invertible), we have

$$\mathsf{I}(\Theta; \Theta^\intercal Z) = \mathsf{I}(\mathrm{A}, \mathrm{B}; \mathrm{A}^\intercal X, \mathrm{B}^\intercal Y) = \mathsf{I}(\mathrm{A}_*, \mathrm{B}_*; \mathrm{A}_*^\intercal X, \mathrm{B}_*^\intercal Y) = \mathsf{I}(\Theta_*; \Theta_*^\intercal Z).$$

Next, we introduce the noise into the latter MI as follows. Denote the distribution of $\Theta_*$ by $\gamma$ and consider

$$\mathsf{I}(\Theta_*; \Theta_*^\intercal Z) - \mathsf{I}(\Theta_*; W) = \mathsf{h}(\Theta_*^\intercal Z) - \mathsf{h}(W) + \mathsf{h}(W|\Theta_*) - \mathsf{h}(\Theta_*^\intercal Z|\Theta_*)$$

$$\leq \int \mathsf{h}(W|\Theta_* = \theta) - \mathsf{h}(\Theta_*^\intercal Z|\Theta_* = \theta)d\gamma(\theta)$$

$$\leq \int \sqrt{\mathsf{J}(\theta^\intercal Z)}\mathsf{W}_2\left( \theta^\intercal Z, \theta^\intercal Z + \sqrt{t}N \right)d\gamma(\theta)$$

$$\leq \sqrt{2kt\int \mathsf{J}(\theta^\intercal Z)d\gamma(\theta)}, \tag{22}$$

where the first inequality follow since $h(W) \geq h(W|N) = h(\Theta_*^\intercal Z)$ (as conditioning cannot increase differential entropy), the second inequality follows from Lemma 1, while the last step upper bounds the 2-Wasserstein distance by $\left(\mathbb{E}[\|\sqrt{t}N\|^2]\right)^{1/2} = \sqrt{2kt}$ and applies Jensen's inequality.

To bound the expected Fisher information, let $\theta^\dagger$ denote the pseudo-inverse of $\theta$. Using the data processing inequality once more, we have

$$\int \mathsf{J}(\theta^\intercal Z)d\gamma(\theta) \leq \int \operatorname{tr}\big(\theta^\dagger \mathsf{J}_{\mathrm{F}}(Z)(\theta^\dagger)^\intercal\big)d\gamma(\theta)$$

$$= \int \operatorname{tr}\big(\mathsf{J}_{\mathrm{F}}(Z)(\theta\theta^\intercal)^{-1}\big)d\gamma(\theta)$$

$$= \|\mathsf{J}_{\mathrm{F}}(Z)\|_{\mathrm{op}} \,\mathbb{E}\big[\operatorname{tr}\big((\Theta_*^\intercal \Theta_*)^{-1}\big)\big]$$

$$= \|\mathsf{J}_{\mathrm{F}}(Z)\|_{\mathrm{op}} \left(d_x\mathbb{E}\left[\operatorname{tr}\big((\tilde{\mathsf{A}}_*^\intercal\tilde{\mathsf{A}}_*)^{-1}\big)\right] + d_y\mathbb{E}\left[\operatorname{tr}\big((\tilde{\mathsf{B}}_*^\intercal\tilde{\mathsf{B}}_*)^{-1}\big)\right]\right),$$

where $\tilde{\mathsf{A}}_*$ and $\tilde{\mathsf{B}}_*$ are random Gaussian matrices of dimensions $d_x \times k$ and $d_y \times k$, respectively, with i.i.d. $\mathcal{N}(0,1)$ entries. Consequently, note that $\tilde{\mathsf{A}}_*^\intercal\tilde{\mathsf{A}}_*$ and $\tilde{\mathsf{B}}_*^\intercal\tilde{\mathsf{B}}_*$ follow the $k \times k$ Wishart distribution with $d_x$ and $d_y$ degrees of freedom, respectively. For $d_x > k+1$ the mean of the inverse is $\mathbb{E}\big[(\mathsf{A}_*^\intercal\mathsf{A}_*)^{-1}\big] = \frac{1}{d_x-k-1}I_{2k}$ and so $\mathbb{E}\big[\operatorname{tr}\big((\mathsf{A}_*^\intercal\mathsf{A}_*)^{-1}\big)\big] = \frac{k}{d_x-k-1}$; cf. e.g., [46] (and similarly for $\tilde{\mathsf{B}}_*$). Inserting this into (22) and combining with (19) and (21) yields the result. $\qquad\square$

## B.2  Proof of Lemma 11

The following bounds follow by the exact same argument of Lemmas 4 and 5 from [32], respectively.

**Lemma 12.** *We have*

$$\mathsf{I}(\Theta_*; W) \leq \kappa \int_{\mathbb{R}^{2k}} \sqrt{\operatorname{Var}(p_{W|\Theta_*}(w|\Theta_*))}dz$$

*where* $\kappa = \sup_{x\in(0,\infty)} \log(1+x)/\sqrt{x} \approx 0.80474$.

**Lemma 13.** *Let* $f : \mathbb{R}^d \to \mathbb{R}_{\geq 0}$ *be a non-negative integrable function and denote its $p$th moment by* $\eta_p[f] := \int \|z\|^p f(z)dz$. *If* $\eta_{d-1}[f]$, $\eta_{d+1}[f] < \infty$, *then*

$$\int \sqrt{f(z)}dz \leq \sqrt{\frac{2\pi^{\frac{d}{2}+1}}{\Gamma\left(\frac{d}{2}\right)}} \left(\eta_{d-1}[f]\eta_{d+1}[f]\right)^{\frac{1}{4}},$$

*where* $\Gamma(z)$ *is the Gamma function.*

Let $\varphi_t$ denote the density of $\mathcal{N}(0, tI_d)$; the dimension is suppressed and should be understood from the context, while the subscript is omitted when $t = 1$. Define the following quantities:

$$m_p(W, \Theta_*) := \frac{\int_{\mathbb{R}^{2k}} \|w\|^p \operatorname{Var}\big(p_{W|\Theta_*}(w|\Theta_*)\big)dw}{\left(\int_{\mathbb{R}^k} \varphi^2(w)dw\right)\left(\int_{\mathbb{R}^k} \|w\|^p \varphi^2(w)dw\right)}$$

$$M(W, \Theta_*) := \sqrt{m_{2k-1}(W, \Theta_*)m_{2k+1}(W, \Theta_*)}.$$

The following lemma is adapted from Lemma 6 of [32] to accommodate our $M(W, \Theta_*)$, definition which slightly differs from theirs.

**Lemma 14.** *If the conditional distribution of $W$ given $\Theta_*$, $p_{W|\Theta_*}$, is absolutely continuous w.r.t. Leb and $M(W, \Theta_*) < \infty$, we have*

$$\mathsf{I}(W; \Theta_*) \leq \kappa \left(\frac{3\pi k}{8}\right)^{\frac{1}{4}} \sqrt{M(W, \Theta_*)},$$

*where* $\kappa$ *is as defined in Lemma 12.*

*Proof.* Lemmas 12 and 13 together imply

$$\mathsf{I}(W; \Theta_*) \leq \kappa\sqrt{\frac{2\pi^{k+1}}{\Gamma(k)}} \left(\eta_{2k-1}\big[\operatorname{Var}\big(p_{W|\Theta_*}(w|\Theta_*)\big)\big] \cdot \eta_{2k+1}\big[\operatorname{Var}\big(p_{W|\Theta_*}(w|\Theta_*)\big)\big]\right)^{\frac{1}{4}},$$

$$= \kappa \sqrt{\frac{2\pi^{k+1}}{\Gamma(k)}} \sqrt{M(W, \Theta_*) \int_{\mathbb{R}^k} \varphi^2(w) dw}$$

$$\times \left( \int_{\mathbb{R}^k} \|w\|^{2k-1} \varphi^2(w) dw \int_{\mathbb{R}^k} \|w\|^{2k+1} \varphi^2(w) dw \right)^{\frac{1}{4}},$$

$$= \kappa \sqrt{M(W, \Theta_*)} \sqrt{2^{1-2k}\pi} \left( \frac{\Gamma(\frac{3k-1}{2})\Gamma(\frac{3k+1}{2})}{\Gamma^2(\frac{k}{2})\Gamma^2(k)} \right)^{\frac{1}{4}},$$

$$\leq \kappa \sqrt{M(W, \Theta_*)} (6k\pi^2)^{\frac{1}{4}} \left( 2^{-2k} \frac{\Gamma(\frac{3k-1}{2})}{\Gamma(\frac{k}{2})\Gamma(k)} \right)^{\frac{1}{2}},$$

$$\leq \frac{\kappa}{2} \sqrt{M(W, \Theta_*)} (6k\pi)^{\frac{1}{4}},$$

where we have substituted in $M(W, \Theta_*)$ as defined above, have noted that $\int \|w\|^p \varphi^2(w) dw = (4\pi)^{-\frac{k}{2}} \Gamma(\frac{k+p}{2})/\Gamma(\frac{k}{2})$. The last step observes that $2^{-2k} \frac{\Gamma(\frac{3k-1}{2})}{\Gamma(\frac{k}{2})\Gamma(k)}$ is a decreasing in $k \geq 1$. This can be verified by by using the fact that that as $\Gamma(z+1) = z\Gamma(z)$, increasing $k$ by 2 will decrease $2^{-2k} \frac{\Gamma(\frac{3k-1}{2})}{\Gamma(\frac{k}{2})\Gamma(k)}$. $\square$

Given the bound in Lemma 14, we next bound the moment $M(W, \Theta_*)$. To that end, we control $\eta_p(W, \Theta_*)$. For convenience of notation, we set $Z = (Z_1^\mathsf{T} \ Z_2^\mathsf{T})^\mathsf{T}$, i.e., $Z_1 = X$, and $Z_2 = Y$, and $d_1 = d_x$, $d_2 = d_y$. We consider two independent copies of $Z$, denoted by $Z^{(1)}$ and $Z^{(2)}$. With this notation, we have the following lemma.

**Lemma 15.** *For any $p \geq 0$, we have*

$$m_p(W, \Theta_*) = \mathbb{E} \left[ (V_{a,1} - R_1)^{-\frac{1}{2}} (V_{a,2} - R_2)^{-\frac{1}{2}} \left( \left( \frac{V_{g,1}^2 - R_1^2}{V_{a,1} - R_1} \right)^{\frac{p}{2}} + \left( \frac{V_{g,2}^2 - R_2^2}{V_{a,2} - R_2} \right)^{\frac{p}{2}} \right) \right]$$

$$- \mathbb{E} \left[ V_{a,1}^{-\frac{1}{2}} V_{a,2}^{-\frac{1}{2}} \left( \left( \frac{V_{g,1}^2}{V_{a,1}} \right)^{\frac{p}{2}} + \left( \frac{V_{g,2}^2}{V_{a,2}} \right)^{\frac{p}{2}} \right) \right],$$

*where*

$$V_{a,i} = t + \frac{1}{2d_i} \|Z_i^{(1)}\|^2 + \frac{1}{2d_i} \|Z_i^{(2)}\|^2,$$

$$V_{g,i} = \sqrt{\left( t + \frac{1}{d_i} \|Z_i^{(1)}\|^2 \right) \left( t + \frac{1}{d_i} \|Z_i^{(2)}\|^2 \right)},$$

$$R_i = \frac{1}{d_i} \langle Z_i^{(1)}, Z_i^{(2)} \rangle.$$

*Proof.* By the definition of $W$, we have $p_{W|\Theta_*}(w|\theta) = \mathbb{E}[\varphi_t(w - \theta^\mathsf{T} Z)]$, whereby

$$p_{W|\Theta_*}^2(w|\theta) = \mathbb{E}[\varphi_t(w - \theta^\mathsf{T} Z^{(1)}) \varphi_t(w - \theta^\mathsf{T} Z^{(2)})].$$

Taking the expectation over the distribution of $\Theta_*$ and swapping the order of expectation yields

$$\mathbb{E}[p_{W|\Theta_*}^2(w|\Theta_*)] = \mathbb{E}[\nu(w, Z^{(1)}, Z^{(2)})]$$

where $\nu(y, z^{(1)}, z^{(2)}) = \mathbb{E}[\varphi_t(w - \Theta_*^\mathsf{T} z^{(1)}) \varphi_t(w - \Theta_*^\mathsf{T} z^{(2)})]$. Note that since $z^{(1)}, z^{(2)}$ are fixed,

$$\begin{bmatrix} \Theta_*^\mathsf{T} z^{(1)} \\ \Theta_*^\mathsf{T} z^{(2)} \end{bmatrix} \sim \mathcal{N}(0, \Sigma)$$

where

$$\Sigma = \begin{bmatrix} \|z_1^{(1)}\|^2/d_1 & 0 & \frac{\langle z_1^{(1)}, z_1^{(2)}\rangle}{d_1} & 0 \\ 0 & \|z_2^{(1)}\|^2/d_2 & 0 & \frac{\langle z_2^{(1)}, z_2^{(2)}\rangle}{d_2} \\ \frac{\langle z_1^{(1)}, z_1^{(2)}\rangle}{d_1} & 0 & \|z_1^{(2)}\|^2/d_1 & 0 \\ 0 & \frac{\langle z_2^{(1)}, z_2^{(2)}\rangle}{d_2} & 0 & \|z_2^{(2)}\|^2/d_2 \end{bmatrix} \otimes \mathrm{I}_k.$$

The proof of Lemma 7 from [32] shows that

$$\nu(w, z^{(1)}, z^{(2)}) = (2\pi)^{2k} \left|\Sigma + t\,\mathrm{I}_{4k}\right|^{-\frac{1}{2}} \exp\left(-\frac{1}{2}\left\|(\Sigma + t\,\mathrm{I}_{4k})^{-\frac{1}{2}}\begin{bmatrix} w \\ w \end{bmatrix}\right\|^2\right). \qquad (23)$$

It is convenient to transform $\Sigma$ into a block-diagonal form. To that end, let us consider the (orthonormal) permutation matrix

$$\mathrm{P} = \begin{bmatrix} 1 & 0 & 0 & 0 \\ 0 & 0 & 1 & 0 \\ 0 & 1 & 0 & 0 \\ 0 & 0 & 0 & 1 \end{bmatrix},$$

and set $\Sigma' = (\mathrm{P}\otimes \mathrm{I}_k)\Sigma(\mathrm{P}\otimes \mathrm{I}_k)^{\mathsf{T}}$. This gives

$$\Sigma' = \begin{bmatrix} \|z_1^{(1)}\|^2/d_1 & \frac{\langle z_1^{(1)}, z_1^{(2)}\rangle}{d_1} & 0 & 0 \\ \frac{\langle z_1^{(1)}, z_1^{(2)}\rangle}{d_1} & \|z_1^{(2)}\|^2/d_1 & 0 & 0 \\ 0 & 0 & \|z_2^{(1)}\|^2/d_2 & \frac{\langle z_2^{(1)}, z_2^{(2)}\rangle}{d_2} \\ 0 & 0 & \frac{\langle z_2^{(1)}, z_2^{(2)}\rangle}{d_2} & \|z_2^{(2)}\|^2/d_2 \end{bmatrix} \otimes \mathrm{I}_k.$$

Note that since P is a permutation matrix, the eigenvalues of $\Sigma$ and $\tilde{\Sigma}$ are equal, hence $|\Sigma + t\mathrm{I}| = \left|\tilde{\Sigma} + t\mathrm{I}\right| = (v_{g,1}^2 - r_1^2)^k(v_{g,2}^2 - r_2^2)^k$. We also obtain

$$\frac{1}{2}\left\|(\Sigma + t\,\mathrm{I}_{4k})^{-\frac{1}{2}}\begin{bmatrix} w \\ w \end{bmatrix}\right\|^2 = \frac{1}{2}\left\|(\tilde{\Sigma} + t\,\mathrm{I}_{4k})^{-\frac{1}{2}}\begin{bmatrix} w_1 \\ w_1 \\ w_2 \\ w_2 \end{bmatrix}\right\|^2 = \left(\frac{v_{a,1} - r_1}{v_{g,1}^2 - r_1^2}\right)\|w_1\|^2 + \left(\frac{v_{a,2} - r_2}{v_{g,2}^2 - r_2^2}\right)\|w_2\|^2,$$

where $(v_{a,i}, v_{g,i}, r_i)$ are as defined in the lemma statement and we have observed that $\mathrm{P}^{\mathsf{T}}\mathrm{P} = \mathrm{I}_4$ and

$(\mathrm{P}\otimes \mathrm{I}_k)\begin{bmatrix} w \\ w \end{bmatrix} = \begin{bmatrix} w_1 \\ w_1 \\ w_2 \\ w_2 \end{bmatrix}$. Substituting into (23) and simplifying yields

$$\nu(w, Z^{(1)}, Z^{(2)}) = (V_{a,1} - R_1)^{-\frac{k}{2}}(V_{a,2} - R_2)^{-\frac{k}{2}} U_1^{\frac{k}{2}} U_2^{\frac{k}{2}} \varphi_{k,1}^2(U_1^{-\frac{1}{2}}y_1)\varphi_{k,1}^2(U_2^{-\frac{1}{2}}y_2)$$

where $U_i = (V_{g,i}^2 - R_i^2)/(V_{a,i} - R_i)$. Then the $p$th moment of $\nu$ with respect to $w$ is (using change of variables) is give by

$$\eta_p\left[\mathbb{E}\left[p_{W|\Theta_*}^2(w|\Theta_*)\right]\right]$$

$$= \mathbb{E}\left[\int \|w\|^p \nu(w, Z^{(1)}, Z^{(2)})dw\right]$$

$$= \mathbb{E}\left[(V_{a,1} - R_1)^{-\frac{k}{2}}(V_{a,2} - R_2)^{-\frac{k}{2}}\int w_1^p U_1^{\frac{k}{2}} U_2^{\frac{k}{2}} \varphi_1^2(U_1^{-\frac{1}{2}}w_1)\varphi_1^2(U_2^{-\frac{1}{2}}z_2)dw\right.$$

$$\left. + (V_{a,1} - R_1)^{-\frac{k}{2}}(V_{a,2} - R_2)^{-\frac{k}{2}}\int w_2^p U_1^{\frac{k}{2}} U_2^{\frac{k}{2}} \varphi_{k,1}^2(U_1^{-\frac{1}{2}}w_1)\varphi_{k,1}^2(U_2^{-\frac{1}{2}}w_2)dw\right]$$

$$= \eta_0[\varphi^2]\eta_p[\varphi^2]\mathbb{E}\left[(V_{a,1} - R_1)^{-\frac{k}{2}}(V_{a,2} - R_2)^{-\frac{k}{2}}\left(\left(\frac{V_{g,1}^2 - R_1^2}{V_{a,1} - R_1}\right)^{\frac{p}{2}} + \left(\frac{V_{g,2}^2 - R_2^2}{V_{a,2} - R_2}\right)^{\frac{p}{2}}\right)\right].$$

$$(24)$$

Next, we find the $p$th moment of the unconditional squared density $p_W^2$. First, as in the conditional case, we write $p_W^2(w) = \mathbb{E}[\tilde{\nu}(w, z^{(1)}, z^{(2)})]$, where

$$\tilde{\nu}(w) = \mathbb{E}\big[\varphi_t(w - \Theta_*^{(1)} z^{(1)})\varphi_t(w - \Theta_*^{(2)} z^{(2)})\big]$$

with $\Theta_*^{(i)}$, for $i = 1.2$, being independent copies of $\Theta_*$. This independence in turn decorrelates $\Theta_*^{(1)} z^{(1)}$ and $\Theta_*^{(2)} z^{(2)}$, i.e.,

$$\begin{bmatrix} \Theta_*^{(1)} z^{(1)} \\ \Theta_*^{(2)} z^{(2)} \end{bmatrix} \sim \mathcal{N}\big(0, \operatorname{diag}(\Sigma)\big).$$

Proceeding as in the correlated case above yields

$$\eta_p\big[\mathbb{E}[p_W^2(w)]\big] = \eta_0[\varphi^2]\eta_p[\varphi^2] = \mathbb{E}\left[V_{a,1}^{-\frac{k}{2}} V_{a,2}^{-\frac{k}{2}}\left(\left(\frac{V_{g,1}^2}{V_{a,1}}\right)^{\frac{p}{2}} + \left(\frac{V_{g,2}^2}{V_{a,2}}\right)^{\frac{p}{2}}\right)\right],$$

and combining this with (24) gives

$$\begin{aligned}
m_p(W, \Theta_*) &= \frac{\eta_p\big[\operatorname{Var}\big(p_{W|\Theta_*}^2(w|\Theta_*)\big)\big]}{\eta_0[\varphi^2]\eta_p[\varphi^2]} \\
&= \mathbb{E}\left[(V_{a,1} - R_1)^{-\frac{k}{2}}(V_{a,2} - R_2)^{-\frac{k}{2}}\left(\left(\frac{V_{g,1}^2 - R_1^2}{V_{a,1} - R_1}\right)^{\frac{p}{2}} + \left(\frac{V_{g,2}^2 - R_2^2}{V_{a,2} - R_2}\right)^{\frac{p}{2}}\right)\right] \\
&\qquad - \mathbb{E}\left[V_{a,1}^{-\frac{k}{2}} V_{a,2}^{-\frac{k}{2}}\left(\left(\frac{V_{g,1}^2}{V_{a,1}}\right)^{\frac{p}{2}} + \left(\frac{V_{g,2}^2}{V_{a,2}}\right)^{\frac{p}{2}}\right)\right].
\end{aligned}$$

$\square$

It remains to bound the expectation in Lemma 15. We start with the following bound.

**Lemma 16.** *For any $p \geq 0$, we have*

$$m_p(W, \Theta_*) \leq \mathbb{E}\left[V_{a,2}^{-\frac{k}{2}} V_{a,1}^{-\frac{k-p}{2}} g_p\left(\frac{R_1}{V_{a,1}}, \frac{R_2}{V_{a,2}}\right) + V_{a,1}^{-\frac{k}{2}} V_{a,2}^{-\frac{k-p}{2}} g_p\left(\frac{R_2}{V_{a,2}}, \frac{R_1}{V_{a,1}}\right)\right],$$

*where $g_{k,p} : (-1, 1) \to \mathbb{R}$ is given by $g_{k,p}(u, v) := (1 - v)^{-\frac{k}{2}}(1 - u)^{-\frac{k}{2}}(1 + u)^{\frac{p}{2}} - 1$.*

*Proof.* Note that for $i, j = 1, 2$ with $i \neq j$, we have

$$\begin{aligned}
&(V_{a,j} - R_j)^{-\frac{k}{2}}(V_{a,i} - R_i)^{-\frac{k}{2}}\left(\frac{V_{g,i}^2 - R_i^2}{V_{a,i} - R_i}\right)^{\frac{p}{2}} - V_{a,i}^{-\frac{k}{2}} V_{a,j}^{-\frac{k}{2}}\left(\frac{V_{g,i}^2}{V_{a,i}}\right)^{\frac{p}{2}} \\
&= V_{a,i}^{-\frac{k+p}{2}} V_{a,j}^{-\frac{k}{2}} V_{g,i}^p\left[\left(1 - \frac{R_j}{V_{a,j}}\right)^{-\frac{k}{2}}\left(1 - \frac{R_i}{V_{a,i}}\right)^{-\frac{k}{2}}\left(1 + \frac{R_i}{V_{g,i}}\right)^{\frac{p}{2}} - 1\right] \\
&= g_p\left(\frac{R_i}{V_{a,i}}, \frac{R_j}{V_{a,j}}\right) V_{a,j}^{-\frac{k}{2}} V_{a,i}^{-\frac{k+p}{2}} V_{g,i}^p \\
&\leq g_p\left(\frac{R_i}{V_{a,i}}, \frac{R_j}{V_{a,j}}\right) V_{a,j}^{-\frac{k}{2}} V_{a,i}^{-\frac{k-p}{2}},
\end{aligned}$$

where we have noted that $V_{g,i} \leq V_{a,i}$ since the geometric mean is upper bounded by the arithmetic mean. Substituting into Lemma 15 completes the proof. $\square$

To make the expectation of $g_p$ tractable we next upper bound it by a quadratic function.

**Lemma 17.** *For any $t > 0$ and $(r_1, r_2)$ such that $|r_1| \leq c_1, |r_2| \leq c_2$, for some $c_1, c_2$, we have*

$$g_p\left(\frac{r_i}{t + c_i}, \frac{r_j}{t + c_j}\right) \leq \frac{k + p}{2}\frac{r_i}{t + c_i} + \frac{k}{2}\frac{r_j}{t + c_j} + t^{-k}(t + 2c_i)^{\frac{p}{2}}(t + c_i)^{\frac{k-p}{2}}(t + c_j)^{\frac{k}{2}}\left(\frac{r_i^2 + r_j^2}{c_i^2 \wedge c_j^2}\right),$$

*where $i, j = 1, 2$ with $i \neq j$.*

*Proof.* Since $g_p(0,0) = 0$, we decompose

$$g_p(u,v) = \left(\nabla g_p(0,0)\right)^\top \begin{bmatrix} u \\ v \end{bmatrix} + h_p(u,v)(u^2 + v^2),$$

where $h_p(u,v) = \left(g_p(u,v) - \nabla g_p(0,0)^\top \begin{bmatrix} u \\ v \end{bmatrix}\right)/(u^2 + v^2)$. It can be verified that $h_p$ is non-negative and nondecreasing in both arguments. Hence, for all $-1 < u \le z_u < 1, -1 < v \le z_v < 1$, we have

$$g_p(u,v) \le \nabla g_p(0,0)^\top \begin{bmatrix} u \\ v \end{bmatrix} + h_p(z_u, z_v)(u^2 + v^2).$$

Furthermore, for $z_u, z_v > 0$,

$$h_p(z_u, z_v) \le \frac{g_p(z_u, z_v)}{z_u^2 + z_v^2} = \frac{1}{z_u^2 + z_v^2}(1 - z_v)^{-\frac{k}{2}}(1 - z_u)^{-\frac{k}{2}}(1 + z_u)^{\frac{p}{2}}.$$

Using $z_u = c_i/(t + c_i), z_v = c_j/(t + c_j)$, we obtain

$$h_p(z_u, z_v) \le \frac{1}{2\min(z_u^2, z_v^2)}t^{-k}(t + c_j)^{\frac{k}{2}}(t + c_i)^{\frac{k-p}{2}}(t + 2c_i)^{\frac{p}{2}},$$

from which the result follows.

$\square$

Next, we provide a bound on $M(W, \Theta_*)$ subject to a.s. boundedness assumption on the squared norms of the random variables. The subsequently presented Lemma 19 then relaxes this assumption to a bound on the MI term of interest.

**Lemma 18.** *Suppose that* $\lambda_{\min} \le \frac{\|\mathbf{X}\|^2}{d_x} \wedge \frac{\|\mathbf{Y}\|^2}{d_y} \le \frac{\|\mathbf{X}\|^2}{d_x} \vee \frac{\|\mathbf{Y}\|^2}{d_y} \le \lambda_{\max}$ *a.s. Then*

$$M(W, \Theta_*) \le 2^{\frac{1}{4}}\left(\frac{\lambda_{\max}}{\lambda_{\min}}\right)^{\frac{k}{2}}\left[4k\frac{\beta_1(Z_1) + \beta_1(Z_2)}{\lambda_{\min}} + 2\left(1 + \frac{2\lambda_{\max}}{t}\right)^k\frac{\beta_2^2(Z_1) + \beta_2^2(Z_2)}{\lambda_{\min}^2}\right].$$

*Proof.* Using Lemma 17 and the definitions of $\beta_r^r$, $R_i$, and $V_{a,i}$, for $p = 2k - 1$, we have

$$\mathbb{E}\left[V_{a,j}^{-\frac{k}{2}}V_{a,i}^{-\frac{k-p}{2}}g_p\left(\frac{R_i}{V_{a,i}}, \frac{R_j}{V_{a,j}}\right)\right] \le (t + \lambda_{\min})^{-\frac{k}{2}}(t+\lambda_{\max})^{\frac{k-1}{2}}\left(\frac{3k-1}{2}\frac{\beta_1(Z_i)}{\lambda_{\min}} + \frac{k}{2}\frac{\beta_1(Z_j)}{\lambda_{\min}}\right)$$
$$+ t^{-k}(t + 2\lambda_{\max})^{\frac{2k-1}{2}}\frac{\beta_2^2(Z_i) + \beta_2^2(Z_j)}{\lambda_{\min}^2}.$$

By Lemma 16, this yields

$$m_{2k-1}(W, \Theta_*) \le (t + \lambda_{\min})^{-\frac{k}{2}}(t + \lambda_{\max})^{\frac{k-1}{2}}(4k - 1)\frac{\beta_1(Z_1) + \beta_1(Z_2)}{\lambda_{\min}}$$
$$+ 2t^{-k}(t + 2\lambda_{\max})^{\frac{2k-1}{2}}\frac{\beta_2^2(Z_1) + \beta_2^2(Z_2)}{\lambda_{\min}^2}.$$

Similarly for $p = 2k + 1$,

$$m_{2k+1}(W, \Theta_*) \le (t + \lambda_{\min})^{-\frac{k}{2}}(t + \lambda_{\max})^{\frac{k+1}{2}}(4k + 1)\frac{\beta_1(Z_1) + \beta_1(Z_2)}{\lambda_{\min}}$$
$$+ 2t^{-k}(t + 2\lambda_{\max})^{\frac{2k+1}{2}}\frac{\beta_2^2(Z_1) + \beta_2^2(Z_2)}{\lambda_{\min}^2}.$$

By the definition of $M(W, \Theta_*) = \sqrt{m_{2k-1}(W, \Theta_*)m_{2k+1}(W, \Theta_*)}$, we obtain

$$M(W, \Theta_*) \le 2^{\frac{1}{4}}\left(\frac{\lambda_{\max}}{\lambda_{\min}}\right)^{\frac{k}{2}}\left[4k\frac{\beta_1(Z_1) + \beta_1(Z_2)}{\lambda_{\min}} + 2\left(1 + \frac{2\lambda_{\max}}{t}\right)^k\frac{\beta_2^2(Z_1) + \beta_2^2(Z_2)}{\lambda_{\min}^2}\right].$$

where we have used the fact that the geometric mean is upper bounded by the arithmetic mean.

$\square$

The derivation is concluded by adapting Lemma 12 of [32] to our notation and setting.

**Lemma 19.** *Let $\mathcal{E} \subseteq \mathbb{R}^{d_x+d_y}$ be measurable. Then*

$$\mathsf{I}(W;\Theta_*) \leq \frac{k}{2}\log\left(1+\frac{\lambda}{t}\right)\left(\mu_Z(\mathcal{E}^c)+\frac{\alpha(X)+\alpha(Y)}{\lambda}\right)+\mu_Z(\mathcal{E})\mathsf{I}(W;\Theta_*|Z\in\mathcal{E}).$$

*Proof.* Letting $U = \mathbb{1}_{\mathcal{E}}(Z)$, the MI chain rule gives

$$\mathsf{I}(W,U;\Theta_*) = \mathsf{I}(W;\Theta_*)+\mathsf{I}(U;\Theta_*|W) = \mathsf{I}(W;\Theta_*|U)+\mathsf{I}(U;\Theta_*).$$

Since $\mathsf{I}(U;\Theta_*)=0$, we have $\mathsf{I}(W;\Theta_*) \leq \mathsf{I}(W;\Theta_*|U)$, and expanding the conditioning yields

$$\mathsf{I}(W;\Theta_*) \leq \mu_Z(\mathcal{E}^c)\mathsf{I}(W;\Theta_*|Z\notin\mathcal{E})+\mu_Z(\mathcal{E})\mathsf{I}(W;\Theta_*|Z\in\mathcal{E}).$$

Recall that $W = \Theta_*^\intercal Z + \sqrt{t}N$, where $\Theta_*$ and $N$ are independent Gaussians. Therefore, conditioned on $Z$, $(W,\Theta_*)$ is jointly Gaussian and we have

$$\mathsf{I}(W;\Theta_*|Z\notin\mathcal{E})$$
$$\leq \mathsf{I}(W;\Theta_*|Z,Z\notin\mathcal{E})$$
$$= \mathsf{h}(W|Z,Z\notin\mathcal{E})-\mathsf{h}(W|\Theta_*,Z,Z\notin\mathcal{E})$$
$$= \frac{k}{2}\mathbb{E}\left[\log\left(\frac{t+\frac{1}{d_x}\|X\|^2}{t}\right)+\log\left(\frac{t+\frac{1}{d_y}\|Y\|^2}{t}\right)\Bigg|Z\notin\mathcal{E}\right]$$
$$= \frac{k}{2\mu_Z(\mathcal{E}^c)}\left(\mathbb{E}\left[\left(\log\left(\frac{t+\frac{1}{d_x}\|X\|^2}{t+\lambda}\right)+\log\left(\frac{t+\frac{1}{d_y}\|Y\|^2}{t+\lambda}\right)\right)\mathbb{1}_{\mathcal{E}^c}(Z)\right]+\log\left(1+\frac{\lambda}{t}\right)\mu_Z(\mathcal{E}^c)\right)$$
$$\leq \frac{k}{2\mu_Z(\mathcal{E}^c)}\log\left(1+\frac{\lambda}{t}\right)\left(\mu_Z(\mathcal{E}^c)+\frac{\alpha(X)+\alpha(Y)}{\lambda}\right),$$

where the last inequality follows from Lemma 19 of [32]. Combining expressions yields the lemma. $\qquad\square$

To use the bound on $M(W,\Theta_*)$ from Lemma 18, we therefore let

$$\mathcal{E}_i = \left\{w_i\in\mathbb{R}^{d_i}:\left|\frac{1}{d_i}\|z_i\|^2-\lambda\right|\leq\frac{\epsilon}{2}\lambda\right\},$$

where $\epsilon\in(0,1]$. Markov's inequality implies $\mu_{Z_i}(\mathcal{E}_i^c)\leq\frac{2}{\epsilon\lambda}\alpha(Z_i)$. Define $\mathcal{E}=\mathcal{E}_1\times\mathcal{E}_2$, so that by the union bound $\mu_Z(\mathcal{E}^c)\leq\mu_{Z_1}(\mathcal{E}_1^c)+\mu_{Z_2}(\mathcal{E}_2^c)\leq\frac{4}{\epsilon\lambda}\alpha(Z_1)\vee\alpha(Z_2)$. Let $Z'$ be drawn according to the conditional distribution of $Z$ given $Z\in\mathcal{E}$, and set $W'=\Theta_*^\intercal Z'+\sqrt{t}N$. By Lemma 18, we have

$$M(W',\Theta_*)\leq 2^{\frac{1}{4}}\left(\frac{1+\epsilon}{1-\epsilon}\right)^{\frac{k}{2}}\left[4k\frac{\beta_1(Z_1')+\beta_1(Z_2')}{\lambda}+2\left(1+\frac{2(1+\epsilon)\lambda}{t}\right)^k\frac{\beta_2^2(Z_1')+\beta_2^2(Z_2')}{\lambda^2}\right].$$

Hence

$$\mathsf{I}(W;\Theta_*|Z\in\mathcal{E})\leq\frac{\kappa}{2}(6\sqrt{2}k\pi)^{\frac{1}{4}}\left(\frac{1+\epsilon}{1-\epsilon}\right)^{\frac{k}{4}}\left[2\sqrt{2k}\sqrt{\frac{\beta_1(Z_1')\vee\beta_1(Z_2')}{\lambda}}\right.$$
$$\left.+2\left(1+\frac{2(1+\epsilon)\lambda}{t}\right)^{\frac{k}{2}}\frac{\beta_2(Z_1')\vee\beta_2(Z_2')}{\lambda}\right],$$

and applying Lemma 19, while noting that $\beta_r^r(Z_i')\leq\frac{\beta_r^r(Z_i)}{\mu_{Z_i}^2(\mathcal{E}_i)}\leq\frac{\beta_r^r(Z_i)}{\mu_Z^2(\mathcal{E})}$, for $i=1,2$ and $r=1,2$, yields the result of Lemma 11.