# OpenReview forum: "$k$-Sliced Mutual Information: A Quantitative Study of Scalability with Dimension"
_NeurIPS.cc/2022/Conference — NeurIPS 2022 Accept_

### Official Review · Reviewer_p5hU · 2022-06-27

**Rating:** 6
**Confidence:** 3
**Soundness:** 4 excellent
**Presentation:** 3 good
**Contribution:** 3 good

**Summary:**

The paper proposes $k$-sliced Mutual Information (k-SMI) which is an extension of sliced Mutual Information (SMI). The difference between SMI and k-SMI is that k-SMI projects original measures to k-dimensional subspaces instead of unidimensional subspaces. Similar to SMI, the authors derive structural properties for k-SMI including Identification of independence, Bounds, Relative entropy and variational form, Entropy decomposition, Chain rule, and Tensorization. The authors derive bounds on the error of Monte Carlo (MC)-based estimates of k-SMI using the continuity of differential entropy in the 2-Wasserstein metric. Moreover, the author explores asymptotics of the population k-SMI, Gaussian approximation results, and neural estimation of SMI. On the application side, k-SMI is applied to sliced InfoGAN for disentangled representation learning.

**Questions:**

* Why we should consider $k$-SMI instead of 1-SMI in practice?
* What is the computational complexity of the projection step in k-SMI?

**Limitations:**

This is fundamental work, hence, there is no foreseen negative social impact.

**Strengths And Weaknesses:**

# Strengths

## Originality
* This is the first work that proposes a variant of mutual information from $k$-dimensional subspace projections. The corresponding theoretical analysis is also original.

## Quality and Significance
* The investigation of the paper is important for scalable and no curse of dimensionality mutual information estimation

## Clarity
* The paper is well-written and easy to follow.

# Weaknesses
* Sampling uniformly from Stiefel manifold requires using QR decomposition that makes Monte Carlo estimation of k-SMI slower than SMI. Therefore, some discussion and empirical investigation are needed.
* It is hard to see the benefits of k-SMI over 1-SMI from the empirical experiments. Also, the comparison of Info-GAN and sliced Info-GAN does not show the superior of k-SMI over k-SMI.

---

> ### Author Response · Authors · 2022-07-30
> **We thank the reviewer for feedback and comments, which we address below:**
>
> We appreciate the reviewer's positive feedback. For the specific comments brought up:
>
> 1. **Sampling from Stiefel manifold/"complexity of projection":**
> Note that the reduced (or ``thin") QR decomposition of a  $d \times k$ matrix can be computed in $O(d k^2)$ operations (e.g., via the Schwarz-Rutishauser algorithm) and this is only a factor $k$ removed from the complexity of matrix multiplication.
> Since for us $k = O(1)$ (indeed, going beyond $k=10$ would be already problematic due to the $n^{-1/k}$ sample complexity bound) and $d \gg k$, this QR step is linear in $d$. Once these samples from the Stielfel manifold are obtained, the actual projection step is simply a matrix multiplication operation of complexity $O(d k n)$ (for projecting all the $n$ data points), making the overall complexity of sampling a projection and projecting the data onto it $O(d k^2 + dkn)$. Note that since $n \gg k$, the contribution of the QR step to the complexity is negligible. We will add a discussion of computational complexity to the supplement.
>
> 2. **$k$-SMI vs SMI:**
> As mentioned in the 1st response to Reviewer 1, $k$-SMI is not proposed as a uniformly better alternative to SMI, but rather as a natural extension thereof. The main contributions of the work are the theoretical properties/guarantees derived for $k$-SMI (and hence for SMI), that address critical shortcoming of the previously available theory from [17]. Generally, projections to more than a single dimension may prove beneficial in situations where the data distribution is increasingly complex. We briefly demonstrated this point in the independence testing experiment (Section 5), but will further highlight it and support it with additional empirical results in the revision. In particular, we plan to run the independence testing experiment with more settings where the common signal is of dimension larger than 2. The goal is to show that $k$-SMI with $k>1$ is able to recover those better and, in turn, offer improved testing capabilities. Sliced InfoGAN experimenters with larger $k$ values will also be added to the revision, along with a discussion of potential observed gains.

---

> > ### Comment · Reviewer_p5hU · 2022-08-03
> > **Response to authors**
> >
> > Thank your for you responses,
> >
> > All my previous questions are addressed.
> >
> > I have two additional questions. The first question is about the stability of k-SMI compared to SMI. When the dimension increases, I guess that using QR decomposition could lead to numerical issues. Hence, k-SMI might not be as stable as SMI in high-dimension. Could authors verify this by some toy experiments?
> >
> > The second question is about the implementation of k-SMI. Can k-SMI be implemented as efficiently as SMI? In more detail, k-SMI needs to sample a Gaussian tensor of size $L \times d \times k$, then QR composition is applied on $L$ matrices of size $d \times k$. After that, $L$ matrix multiplications are computed. I wonder if the whole process can be computed in the vectorization (tensorization) way on some high-level languages such as R and Python or if we need to use low-level packages to parallelize the computation on $L$ independent projections.
> >
> > Best regards,

---

> > > ### Author Response · Authors · 2022-08-04
> > > **Response to additional questions**
> > >
> > > **“When the dimension increases, I guess that using QR decomposition could lead to numerical issues.“**
> > > In fact, numerical issues are easily avoidable due to the simplicity of the procedure. To easily see this, note that a $(d \times k)$ matrix $\mathrm{M}$ can be orthonormalized by computing the SVD $(\mathrm{U S V}^\intercal)$ of the $(k \times k)$ matrix $(\mathrm{M}^\intercal \mathrm{M})$ and multiplying the resulting $(k \times k)$ matrix $\mathrm{U S}^{-1/2}$ by the original $(d \times k )$ matrix, forming $\mathrm{M U S}^{-1/2}$. Since $k$ is very small, the only numerical issues that could arise in this implementation are due to a pair of matrix multiplications. However, on modern hardware this should not be an issue, even for extremely high dimensions. Furthermore, note that when $d$ is large (in practice $d > 20$ or 50), a $(d \times k)$ matrix with i.i.d. $\mathcal{N}(0, 1/d)$ Gaussian entries is arbitrarily close to orthonormal with overwhelming probability. Hence, the QR decomposition may not even be needed in high dimensions!
> > >
> > >
> > >
> > > **”k-SMI needs to sample a Gaussian tensor of size L x d x k, then QR decomposition is applied on L matrices of size d x k. After that, L matrix multiplications are computed. I wonder if the whole process can be computed in the vectorization (tensorization) way on some high-level languages such as R and Python or if we need to use low-level packages to parallelize the computation on L independent projections.”**
> > >
> > > This is a good question. We did not run into scaling issues with the following implementation:
> > > * Create $L$ random seeds, but do not yet use them to generate the $L$ slices.
> > > * Choose some value $\ell < L$ that is moderate (typically 20-50 is sufficient).
> > > * As the neural network is training, it loads in batches of data. For each new batch, randomly choose $\ell$ of the $L$ seeds. If the QR decomposition already exists for those seeds, load them and project the batch’s data using them, creating a set of $\ell$ projected batches. If the QR decomposition for any of the $\ell$ slices do not yet exist, compute those QR decompositions, store them, and project the data.
> > > * Note that if $\ell  k < d$, these $\ell$ projected batches actually require less memory than the original batch.
> > > * Python allows for vectorization of all operations.
> > > * Update the $\ell$ chosen slice’s neural networks.
> > > This subsampling of $L$ at each batch is valid since we seek stochastic gradient updates. As a result of this procedure, everything is computed on the fly and $\ell$ is small enough that regardless of $k$, the computational cost at each batch does not scale with $L$ and the overall cost is comparable to running a batch of a non-sliced neural MI estimator, e.g. MINE. We are happy to add a short clarification of the above to the revision.
> > >
> > > Again note that if $d$ is large and QR decomposition were to become a computational problem, it can be skipped due to the concentration properties of the iid Gaussian matrix.
> > >
> > > **”All my previous questions are addressed.”**
> > > We are glad to hear your questions were addressed, we hope the new answers continue to clarify our work and that you might consider raising your score accordingly. Thank you!

---

> > > > ### Comment · Reviewer_p5hU · 2022-08-08
> > > > **Response to authors**
> > > >
> > > > Thank you for your response.
> > > >
> > > > I appreciate the effort of the authors in clarifying the paper. All my questions are addressed. However, I need time to reevaluate the paper. I will adjust my score after the discussion with other reviewers.
> > > >
> > > > Best regards,

---

### Official Review · Reviewer_MwzW · 2022-07-07

**Rating:** 4
**Confidence:** 4
**Soundness:** 3 good
**Presentation:** 3 good
**Contribution:** 2 fair

**Summary:**

The paper proposes k-SMI (k-sliced mutual information) as a measure of statistical dependence defined by averaging MI (mutual information) terms between k-dimensional projections of high dimensional random variables, which is an extension of SMI (sliced mutual information). Although SMI avoids the curse of dimensionality in mutual information estimation by averaging
over the one-dimensional slices, the bounds rely on high-level assumptions that are hard to verify in practice and do not explicitly capture dependence on the ambient dimension, which is crucial for understanding scalability. The authors extend this theory to k-dimensional
projections and provide justifications with explicit dependence on k and the dimensions of data via numerical experiments. Additionally, they provide an application to sliced infoGAN, showing that k-SMI can successfully replace classic MI even in applications with more intricate
underlying structure.

**Questions:**

Significance
I recognize that estimating MI in high dimensions is a hard problem. This approach of k-SMI is interesting in that regard. However, it seems to me that one could just as easily perform some kind of dimensionality reduction first (e.g. using diffusion maps, or even PCA) and then estimating the MI on the reduced dimensions. This would likely be computationally cheaper as the MC step in k-SMI is unlikely to be computationally fast. Also, this would allow for nonlinear projections if the dimensionality reduction method is nonlinear. In contrast, k-SMI only performs linear projections. What advantage, if any, does k-SMI have over this approach?

Experiments
* In line 312 — the result of the independence test, It seems the sinusoidal model (a) and the common signal model (b) perform differently. In (a), it only works when k = 1 when d is large; In particular, when d = 20, the AUC is pretty small for 2-SMI. On the other hand, both 1-SMI and 2-SMI work pretty well in (b) even when d is large enough. Although the difference of the two models is clear, how they could make such a difference in result is not discussed.
* Again in the independence test, it is shown that there is potential gain of higher k, but I am worrying about whether the simulated data is too artificial, since Z and V are both simulated from multivariate Gaussian with identity covariance matrices (line 282); clearly model (a) does not provide similar results. So I think experimenting with the theory on more models would be preferred.
* In line 318 — the experiment of sliced InfoGAN, it is only shown that the k-SMI successfully disentangles the latent factors with k = 1. Are there any other results from other k values? In addition, it would be nice if some quantitative measure of performance could be given for this experiment.

**Limitations:**

Limitations were generally not discussed. Are there circumstances where the authors expect k-SMI to fail in some way?

**Strengths And Weaknesses:**

Strengths:
* The originality is good. While the approach is a straightforward extension of SMI, the authors provide significantly improved theoretical results that also apply to SMI.
* The theoretical results are good as the authors provide a comprehensive quantitative study of dependence on dimension of k-SMI, encompassing the MC error, performance bounds for neural estimators, and asymptotics of the population k-SMI as dimension increases.
* The paper is generally clear and well-written.

Weaknesses
* I'm not entirely convinced of the significance of this work. See questions below.
* The experiments are decent but could use some improvement. See questions below.

---

> ### Author Response · Authors · 2022-07-30
> **We thank the reviewer for feedback and comments, which we address below:**
>
> 1. **Significance:** We agree that a wide variety of dimensionality reduction techniques can and should be considered within the broad context of estimating dependence from data. Indeed, these methods may be computationally efficient and have sufficient discriminative power for certain models.
>  However, the focus of this paper is on dependence measures (such as mutual information and SMI) that are interpretable and possess certain mathematical properties (e.g., nullify if and only if the underlying variables are independent). Many of these properties no longer hold if one first applies a non-invertible dimensionality reduction to the data and then measures the dependence between the resulting low-dimensional representations. A crucial aspect of $k$-SMI is that it depends on the ensemble of low-dimensional projections indexed by the Stiefel manifold (i.e., it is not just the mutual information in a single projection direction). This is the reason why it possesses the 'Identification of Independence' property stated in Proposition 2.
> A method based on analyzing a low-dimensional function of the data, on the other hand, is fundamentally limited since any information lost in the processing of the data cannot be recovered.
> With this in mind, the choice of linear projections is perhaps the most natural starting point. The idea that a high-dimension distribution can be represented by the marginals of its low-dimensional projection has a long history that can be traced back to work of Cramer and Wald in the 1930s. It also underpins the data exploration technique of projection pursuit used in statistics since the 1970s. More generally, one may go beyond linear projections and consider averaging (or maximizing) over a different class of test functions (e.g., a $k$-SMI variant that applies non-linear feature maps to the data). We think the theory established in our paper provides the foundations for exploring these directions in future work. We will mention this appealing research direction in the summary section of the revision.
> To conclude, our paper belongs to an active line of work on surrogates of mutual information that are both interpretable and have desirable mathematical properties. In this context, our results are the first to establish non-asymptotic estimation error bounds with explicit dependence on the problem dimension. We will add language to the revision that further clarifies the motivation to study $k$-SMI, in accordance to the above.
>
> 2. **Experiments:** We address the three points raised by the reviewer below:
>
>     a. **Independence testing -- dependence on $k$:** It is important to note that in Figure 2, settings (a) and (b) are unrelated and should not be compared. In particular, they have different signal-to-noise ratios and thus are of differing difficulty. Instead, one should compare the relative performance of the various $k$. We agree with the reviewer that more emphasis on the effect of $k$ is warranted and will provide it in the revision. First, we explain the difference in performance observed in Figure 1. In Figure 1(a), the shared signal is 1-dimensional. Consequently, $k=1$ outperforms $k=2$ since increasing the projection dimension to $k=2$ does not extract additional information but does worsen the sample complexity. In contrast, in Figure 1(b), the shared single is 2-dimensional and $k=2$ outperforms $k=1$. This happens since the increased sample complexity of $k=2$ is made up for by the fact that the 2-SMI is able to extract additional meaningful information beyond what 1-SMI can capture. This results in a crisper dependence profile and, in turn, improved independence testing performance. The conclusion is that for more complex data structures, it may be beneficial to pay the extra sample complexity in order to gain in expressivity and overall performance. There is a tradeoff involved in the optimal choice of $k$, but $k$-SMI provides the flexibility to adapt the projection dimension.
>
>     In the revision, we will expand on the independence testing experiments to consider common signals of rank greater than 2, and also examine nonlinear low-rank manifolds. The goal is to show the potential benefits of larger $k$ values, but to also surface the aforementioned tradeoff with sample complexity and illustrate it empirically.
>
>    b. **Sliced InfoGAN:** Sliced InfoGAN experiments with larger $k$ values will be added to the revision, along with a discussion of observed gains. We did not initially include these due to space limitations and since our focus was not as much on the extension to $k > 1$, but rather on theoretically establishing the scalability of SMI/$k$-SMI to high dimensions.

---

> > ### Author Response · Authors · 2022-08-08
> > **Follow-up**
> >
> > Dear Reviewer,
> >
> > We understand the time window to interact with reviewers is very short this year, and only lasts up to this Tuesday. Nevertheless, could you confirm that you have had a chance to look at the new elements we have provided in the rebuttal?
> >
> > We do hope that the rebuttal has resolved all of the concerns raised in your review. In particular, we have tried to further highlight our main focus and contributions, so as to make sure that they are not overlooked. We believe that the discussion period is particularly useful in cases like this and are happy to address any further questions/concerns you might have. We look forward to further hearing from you.
> >
> > Thank you again for your time.

---

> > ### Comment · Reviewer_MwzW · 2022-08-08
> > **Unconvinced of significance**
> >
> > Thank you for the response. I still remain largely unconvinced of the significance here. While it's true that in theory, applying dimensionality reduction beforehand may nullify certain properties, in practice this is unlikely to be an issue in my opinion, especially in settings where the manifold assumption is reasonable. To go with the authors' example of the independence property, it is true that applying a noninvertible reduction could make it so that the MI is zero even if the original variables are not independent. But if we assume we're applying a good nonlinear method that captures the majority of the variability in the data (e.g. diffusion maps), this would only occur in a low SNR setting. While k-SMI may be able to theoretically detect the nonzero MI in this case, does it work in practice? I think I would need to see some comparisons before being convinced on this point.
> >
> > In summary, it would be helpful if the authors could create some experiments where using a nonlinear dimensionality reduction method followed by MI estimation falsely estimates a (close to) zero MI while k-SMI does not.

---

> > > ### Author Response · Authors · 2022-08-09
> > > **Response**
> > >
> > > Thanks for providing more feedback. At the risk of oversimplifying, is it fair to say that your main concern can be summarized as follows: The SMI estimator does not seem to improve upon other MI estimation methods (e.g., by first applying dimensionality reduction techniques and then estimating MI from the transformed data) for the purposes of estimating MI.
> > >
> > > First, as was stressed several times in the original SMI paper https://arxiv.org/pdf/2110.05279.pdf (referred to henceforth as [A]), SMI (or $k$-SMI) is not a proxy of MI but rather as a measure of dependence in its own right. The goal of the theoretical exploration of SMI properties (Section 3.1 in [A] and the current submission) is to justify it as a meaningful such measure. There are pair of random variables $(X,Y)$ for which the gap between SMI and MI is unbounded; cf. the example from Section 3.2 of [A] with $a=1$, for which MI is infinite but SMI is finite. Thus, using SMI or $k$-SMI to approximate MI is ill-advised and should not be done.
> > >
> > > Focusing on MI estimation, we note that in high- or even moderate-dimensional settings, accurate estimation of MI is impossible due to the curse of dimensionality. Indeed, existing sample complexity bounds for minimax optimal MI estimators (e.g., based on kNN or KDE techniques) scale as $n^{-1/(d_x+d_y)}$, which is vacuous already when d is larger than, say, 7. Even under the manifold hypothesis, the intrinsic dimension in most ML applications is too large for MI estimation to be possible; cf. https://openreview.net/pdf?id=XJk19XzGq2J where common image datasets are shown to have intrinsic dimension ranging between 20-40.
> > >
> > > With this in mind, attempting to estimate high-dimensional MI (using dimensionality reduction techniques, as the reviewer suggested, or otherwise) is not reasonable. Indeed, if one obtains an estimate of the MI that is small, then they may (falsely) conclude that the MI is actually small. But this could be a consequence of large error (because of the curse of dimensionality), which highlights the importance of non-vacuous formal guarantees for estimating the object of interest. Unfortunately, MI estimators are routinely applied to high-dimensional data without properly acknowledging the fundamental limitations of MI estimation. While the quantities the estimators produce can be useful for certain applications, it would be incorrect (and possibly dangerous) to believe that they are actually estimating the underlying MI.
> > >
> > > SMI or $k$-SMI, on the other hand, are meaningful and interpretable dependence measures that can be reliably estimated at the parametric $n^{-1/2}$ rate, in arbitrary dimension. In this case, if one obtains an estimate of the SMI that is small, then they (correctly) conclude that the SMI is small and perform principle and correct inference accordingly. The virtue of SMI and $k$-SMI is that they are both meaningful dependence measures with desirable theoretical properties **and** that they can be efficiently estimated in practice. The heuristic approach described by the reviewer can also be applied in practice, but what is it actually measuring? What understanding do we have of the resulting value? How robust is it w.r.t. the chosen ("good") transformation of the data or is the observation an artifact? All these are crucial questions for principled inference, but unlike for SMI or $k$-SMI, it is unclear how to account for them for the approach proposed by the reviewer.
> > >
> > > More importantly, we reemphasize that **SMI is not a contribution of this work**. SMI was previously proposed in a published NeurIPS paper that received a spotlight presentation [A]. Our work closes several important theoretical gaps left in the first paper, by providing a quantitative, dimension-dependent, sample complexity analysis of SMI and $k$-SMI estimation, as well as additional structural properties as summarized in our first rebuttal text. Given the above, **we kindly urge the reviewer to address the theoretical contributions of our work, rather than criticize SMI as an object of interest, given the published SMI paper [A].**

---

### Official Review · Reviewer_2P49 · 2022-07-11

**Rating:** 3
**Confidence:** 4
**Soundness:** 2 fair
**Presentation:** 3 good
**Contribution:** 2 fair

**Summary:**

Sliced mutual information (SMI) serves as a surrogate measure of dependence to classic MI, but a quantitative characterization of how its estimation rates depend on the dimension remains obscure.
This paper naturally extends the original SMI definition to k-SMI and provides a comprehensive analysis on how k-SMI's estimation rates depend on dimension.
The authors utilize the neural estimation framework to estimate k-SMI, with optimal convergence rates.
They discuss numerical considerations and perform a thorough simulation study and real data analysis, showing the favorable performance of their method.

**Questions:**

* For the proposed measure k-SMI, there seems to be a little lack of innovation.
It is a natural extension of SMI.
And I don't find the advantages of k-SMI($k>1$) over SMI.
Furthermore, the computational cost of k-SMI is larger than SMI.

* There is not any discussion about the complexity of the proposed method.
Though the proposed method can be estimated by a neural network with optimal convergence rates, the computational cost is also important in practice.
The proposed method need to solve an optimization problem by neural network, which may be a disadvantage compared with other dependence measurements.
Some discussion and empirical results are needed.

* There seem to be many existing methods that measure the dependence between random variables without using a neural network.
The authors may compare their algorithm with these methods and clarify the advantages and disadvantages.

**Limitations:**



**Strengths And Weaknesses:**

* The authors provide a comprehensive analysis of the quantitative account of the scalability question of k-SMI.
They provide ample theoretical analysis on it.

* The authors apply a neural estimation framework to estimate k-SMI.
What's more, they provide this estimation with theoretical guarantees, which claims optimal convergence rates.

* The simulation study and real data analysis are very sufficient.
The authors illustrate the effectiveness from many aspects.

---

> ### Author Response · Authors · 2022-07-30
> **We thank the reviewer for feedback and comments, which we address below:**
>
> 1. **Significance and main contributions:** We agree with the reviewer that k-SMI is a natural extension of SMI from [17], and its introduction is not a particularly significant contribution as such. However, we also stress that the definition of k-SMI is not the main contribution or theme of this work. Rather, the theory provided in [17] for SMI ($k=1$) was lacking on several key aspects, and the main goal of the current paper is to resolve those issues. Namely, the formal accuracy guarantees provide in [17] for the SMI estimator (i) relied on high-level assumptions, which are hard to verify in practice; (ii)hide dimension-dependent constants whose characterization is crucial for understanding scalability in dimension; and (iii) did not cover the neural estimator, which was mentioned in [17] but was left unanalyzed. In the current submission we provide a full account of all three items so as to close those gaps and facilitate principled usage of SMI/k-SMI in practice.
>
>      Providing the said theory involved overcoming several technical challenges, which brought us to develop the differential entropy Lipschitz continuity results from Lemma 1 and Proposition 1. We believe that these technical tools are of independent interest and could be valuable for analysis in various other settings. We also view the Gaussianization of k-SMI (Section 4.3) as another significant contribution that sheds light on what both SMI and k-SMI quantify and how they can be approximated using simpler properties of the distributions (e.g., covariance matrices). Recall that SMI and k-SMI are proposed as new, mutual-information-like dependence measures (not necessarily proxies of classic mutual information); as such, understanding the type of dependence being quantified is pivotal for appropriate application thereof. In sum, we feel that faulting the submission on the fact that k-SMI offers little innovation of top of SMI from [17] overlooks our main contributions and does not do justice to our work.
>
>     Chronologically, we first focused on the SMI ($k=1$) from [17], but then realized that the theory can be adapted to cover any $1\leq k\leq d_x\wedge d_y$. We decided to update presentation to account for the more general k-SMI functional, but as mentioned at the start, this is more of a natural extension than the core contribution. That said, projections to more than a single dimension may prove beneficial in situations where the data distribution is increasingly complex. We briefly demonstrated this point in the independence testing experiment (Section 5), but will further highlight it and support it with additional empirical results in the revision. In particular, we plan to run the independence testing experiment with more settings where the common signal is of dimension larger than 2. The goal is to show that k-SMI with $k>1$ is able to recover those better and, in turn, offer improved testing capabilities. Sliced InfoGAN experimenters with larger $k$ values will also be added to the revision, along with a discussion of observed gains.
>
> 2. **Measuring dependence via neural nets:** k-SMI is not inherently tied to neural networks and can be estimated without their use. Indeed, the estimator given in Equation (2) allows for any mutual information estimator between k-dimensional variables for $\hat{\mathsf{I}}(\cdot,\cdot)$; the $\delta_k(n)$ term from Theorem 1 would then be replaced with an error bound for that estimator. This modularity is one of the advantages of the proposed approach and the bound from Theorem 1. In [17], the classic Kozachenko-Leonenko kNN estimator as well as a KDE-based estimator were used and analyzed (albeit still hiding dependence on dimension in their bounds as previously mentioned). In the current submission we derived formal guarantees for the neural estimator since this is a highly popular method these days but no theory for k-SMI (or SMI) neural estimation was available prior to this work. Generally, neural estimators are appealing due to their computational scalability and compatibility with gradient-based optimization routines. We will add a remark to the text to discusses and clarify the above.
>
> 3. **Computational complexity:** In light of the previous response, applying the (non-NN) Kozachenko-Leonenko entropy estimator to k-SMI would yield a computational complexity of $O\big(nkm(d+\log n)\big)$, i.e. linear (up to a log factor in $n$) in all arguments. The complexity of optimizing over NNs is harder to characterize, but this fact applies to any of the numerous applications of NNs. Notwithstanding, neural estimators have seen huge empirical success in high-dimensional estimation settings, including for mutual information (see the MINE estimator for example, among many others). Combined with the compatibility of neural estimators with gradient-based optimization, we believe that the NN estimator (and variants of it) have many potential uses in machine learning applications.

---

> > ### Author Response · Authors · 2022-08-08
> > **Follow-up**
> >
> > Dear Reviewer,
> >
> > We understand the time window to interact with reviewers is very short this year, and only lasts up to this Tuesday. Nevertheless, could you confirm that you have had a chance to look at the new elements we have provided in the rebuttal?
> >
> > We do hope that the rebuttal has resolved all of the concerns raised in your review. In particular, we have tried to further highlight our main focus and contributions, so as to make sure that they are not overlooked. We believe that the discussion period is particularly useful in cases like this and are happy to address any further questions/concerns you might have. We look forward to further hearing from you.
> >
> > Thank you again for your time.

---

### Meta-Review · Area_Chair_De6v · 2022-08-26

**Recommendation:** Accept
**Confidence:** Less certain

**Metareview:**

While the reviewers have raised concerns, mainly about the significance of k-SMI, they also acknowledged that the theoretical contributions of the paper are solid. On the other hand, I do agree with the authors on the fact that the main contribution of the paper is the theoretical analysis that can be trivially applied to SMI as well, rather than the introduction of k-SMI itself. Hence, I will go ahead and recommend a borderline acceptance for the paper. However, I strongly suggest the authors to take into account the reviewers' concerns, and reword their contributions to emphasize their theoretical contributions, which also clarify the behavior of SMI.

**Award:**

No

---

### Decision · Program_Chairs · 2022-09-14

Accept